# A disinhibitory nigra-parafascicular pathway amplifies seizure in temporal lobe epilepsy

Bin Chen [1,4], Cenglin Xu [1,4], Yi Wang [1,2,4 ✉], Wenkai Lin [1,4], Ying Wang [1], Liying Chen[1], Heming Cheng [1], Lingyu Xu[1], Tingting Hu [1], Junli Zhao [1], Ping Dong [1], Yi Guo [2], Shihong Zhang [1], Shuang Wang[2], Yudong Zhou [1], Weiwei Hu [1], Shuming Duan[1] & Zhong Chen [1,2,3 ✉]

The precise circuit of the substantia nigra pars reticulata (SNr) involved in temporal lobe epilepsy (TLE) is still unclear. Here we found that optogenetic or chemogenetic activation of SNr parvalbumin$^+$ (PV) GABAergic neurons amplifies seizure activities in kindling- and kainic acid-induced TLE models, whereas selective inhibition of these neurons alleviates seizure activities. The severity of seizures is bidirectionally regulated by optogenetic manipulation of SNr PV fibers projecting to the parafascicular nucleus (PF). Electrophysiology combined with rabies virus-assisted circuit mapping shows that SNr PV neurons directly project to and functionally inhibit posterior PF GABAergic neurons. Activity of these neurons also regulates seizure activity. Collectively, our results reveal that a long-range SNr-PF disinhibitory circuit participates in regulating seizure in TLE and inactivation of this circuit can alleviate severity of epileptic seizures. These findings provide a better understanding of pathological changes from a circuit perspective and suggest a possibility to precisely control epilepsy.

[1] Key Laboratory of Medical Neurobiology of the Ministry of Health of China, Department of Pharmacology, School of Basic Medical Sciences, College of Pharmaceutical Sciences, Zhejiang University, Hangzhou, China. [2] Epilepsy Center, Department of Neurology, Second Affiliated Hospital, School of Medicine, Zhejiang University, Hangzhou, China. [3] College of Pharmaceutical Science, Zhejiang Chinese Medical University, Hangzhou, China. [4] These authors contributed equally: Bin Chen, Cenglin Xu, Yi Wang, Wenkai Lin. ✉email: wang-yi@zju.edu.cn; chenzhong@zju.edu.cn

Epilepsy is one of the most common neurological disorders, affecting ~1% of the population[1]. Epilepsy is considered to be a circuit-level syndrome pathologically characterized by excessive or hypersynchronous discharges with enhanced neuronal excitability due to excitatory–inhibitory imbalance[2]. Temporal lobe epilepsy (TLE) is particularly challenging from a therapeutic standpoint because of frequent resistance to anti-epileptic drugs (AEDs)[3]. In addition, surgical resection of epileptic foci within the temporal lobe might still fail to control seizures[4]. The main factor in these challenges is probably that a complex epileptogenic network has formed in the brain of TLE patients. Structural and metabolic imaging from both clinical and experimental studies demonstrate that abnormal pathological changes in TLE are associated with not only the neighboring epileptogenic structures but also with remote brain regions, extending from subcortical limbic structures to cortical and other remote structures[5,6]. Thus, identification of the neuronal circuitry involved in seizure of TLE is necessary for developing precise and safe interventions to control TLE.

Basal ganglia circuits are closely involved in the modulation, propagation, and cessation of seizure in different types of epilepsies, including TLE[7–9]. The substantia nigra pars reticulata (SNr), a region that mainly contains GABAergic neurons, controls the activity of both corticothalamic and limbic networks through its primary GABAergic output, acting like the "choke point" of the basal ganglia[10,11]. Clinical reports demonstrate that TLE patients have smaller structural volumes of substantia nigra and altered iron concentration in the substantia nigra compared with healthy controls[12,13]. Experimental studies have also reported structural and functional changes of SNr neurons among different types of epileptic models[14–18]. In addition to those findings, lesion[19], pharmacological interference[20–24], or deep brain stimulation[25,26] targeting the SNr can modulate the intensity of epileptic seizures, suggesting that the SNr plays a key role in seizure control. However, conflicting data have been reported from studies investigating the role of the SNr in epilepsy due to the lack of specificity in the approaches to intervening neuronal activity[27]. Particularly, the cell types and neural circuits responsible for the SNr controlling of epilepsy are elusive. In this study, by using optogenetics[28] and chemogenetics[29], we investigated whether and how SNr parvalbumin (PV) neurons, the main subtype of GABAergic neurons in the SNr[30], regulate seizures in TLE. We identified a nigra-parafascicular disinhibitory circuit for regulation of seizure in TLE. We found that selective activation of SNr PV neurons amplifies seizure activities, whereas inhibition of those neurons alleviates the severity of epileptic seizures. Subsequent in vivo and in vitro electrophysiology combined with rabies virus-assisted circuit mapping revealed that SNr PV neurons form structural and functional connection with neurons in the parafascicular nucleus (PF), some of which form the nigra-parafascicular disinhibitory circuit that is involved in bidirectional modulation of seizures in TLE.

## Results

**SNr PV+ GABAergic neurons are activated during seizure.** To examine the function of SNr GABAergic neurons in relation to seizure in TLE, we recorded the firing activity of SNr putative GABAergic neurons and local field potentials (LFPs) in anesthetized mice using in vivo single-unit recording. These recordings were made during ventral hippocampal CA3 kindling stimulation (monophasic square-wave pulses, 20 Hz, 1 ms/pulse, 100 pulses) in the hippocampal kindling model that is one of the most commonly used animal models of TLE and mimics clinical focal epilepsy. The putative GABAergic neurons, which constitute the majority of SNr neurons, are characterized by high firing rates and narrow spike

waveforms (Supplementary Fig. 1a), as described in a previous study[31]. Immediately after the initiation of CA3 kindling, about 55% of GABAergic neurons in the ipsilateral SNr increased their firing rate during kindling-induced acute seizure (peak firing rate was 17.06 ± 1.01 Hz; out of 38 recorded neurons from 3 wild-type mice, 21 neurons were activated, 4 inhibited, 13 no response; no inter-animal statistical difference was detected, $\chi^2$-test, $p = 0.8927$; Supplementary Fig. 1b, c). These data indicated that most of the putative GABAergic neurons in the ipsilateral SNr were activated during hippocampal kindling seizures.

PV neurons are a major population of GABAergic neurons in the SNr[30]. We tried to further define the functional change of SNr PV neurons during seizure in TLE by using fiber photometry to record calcium signals from SNr PV neurons of freely moving mice. Following stereotaxic infusion of the Cre-dependent adeno-associated virus AAV-Ef1α-DIO-GCaMP6m-eYFP into the SNr of PV-Cre-recombinase mice (*PV-Cre*) mice (referred to as *PV-GCaMP6m$^{SNr}$* mice), the calcium indicator GCaMP6m was efficiently expressed in PV+ neurons (77.84 ± 1.38% of PV+ neurons expressed enhanced yellow fluorescent protein (eYFP) and 80.28 ± 6.11% of eYFP+ neurons were PV+ neurons; Fig. 1a). Then we implanted an optical fiber into the SNr for recordings of GCaMP fluorescence changes during hippocampal kindling-induced seizures. We detected an increase of calcium response (average signal peak $\Delta F/F$: 2.02 ± 0.15% for focal seizures (FSs) and 3.24 ± 0.54% for generalized seizures (GSs); Fig. 1b) in the ipsilateral SNr during hippocampal kindling-induced seizure (monophasic square-wave pulses, 20 Hz, 1 ms/pulse, 40 pulses). Calcium signals of SNr PV neurons gradually increased along with the development of seizure severity during a 3-day kindling acquisition (Supplementary Fig. 2). During hippocampal kindling seizures, the waveforms of increased calcium signals in SNr PV neurons were similar to the shape of curve showing the increased firing rate of electrophysiological signals in in vivo single-unit recordings (Supplementary Fig. 1b). The above results imply that SNr GABAergic neurons including PV neurons are activated during hippocampal seizure.

**Activation of SNr PV neurons amplifies seizure activities.** To investigate the role of SNr GABAergic neurons in seizure of TLE, we used an optogenetic approach to selectively stimulate Channelrhodopsin 2 (ChR2)-expressing GABAergic neurons in the SNr of *Vgat-ChR2-eYFP* mice. Histological data confirmed that eYFP-expressing neurons in the SNr co-expressed glutamate decarboxylase (GAD) 65/67 (95.35 ± 1.70% of GAD+ neurons expressed eYFP and 68.02 ± 0.56% of eYFP+ neurons were GAD+ from 3 *Vgat-ChR2-eYFP* mice; Supplementary Fig. 1d). The fewer number of GAD neurons compared with eYFP-labeled neurons was likely due to the detection limitation of GAD antibody. We designed opto-electrodes that allow optogenetic stimulation during in vivo single-unit recordings in the SNr of urethane-anesthetized mice[32]. Representative peri-event histograms confirmed that blue-light stimulation (473 nm, 20 Hz, 10 ms, 5 mW, 10 s on–off cycle) excited 15 out of 19 SNr GABAergic neurons from 3 *Vgat-ChR2-eYFP* mice, suggesting that SNr GABAergic neurons can be functionally activated by blue-light stimulation. Yellow-light stimulation (589 nm, 20 Hz, 10 ms, 5 mW, 10 s on–off cycle), serving as the control stimulation, had no effect on the same neurons (Supplementary Fig. 1f). In the hippocampal kindling model, we applied 30 s photostimulation in the ipsilateral SNr immediately after each hippocampal kindling stimulation (similar to the closed-loop stimulation pattern[33]; Supplementary Fig. 1e). We found that photo-activation (473 nm, 20 Hz, 5 mW, 10 ms, 30 s) of SNr GABAergic neurons accelerated the progression of seizure stage (Supplementary Fig. 1g, h-1),

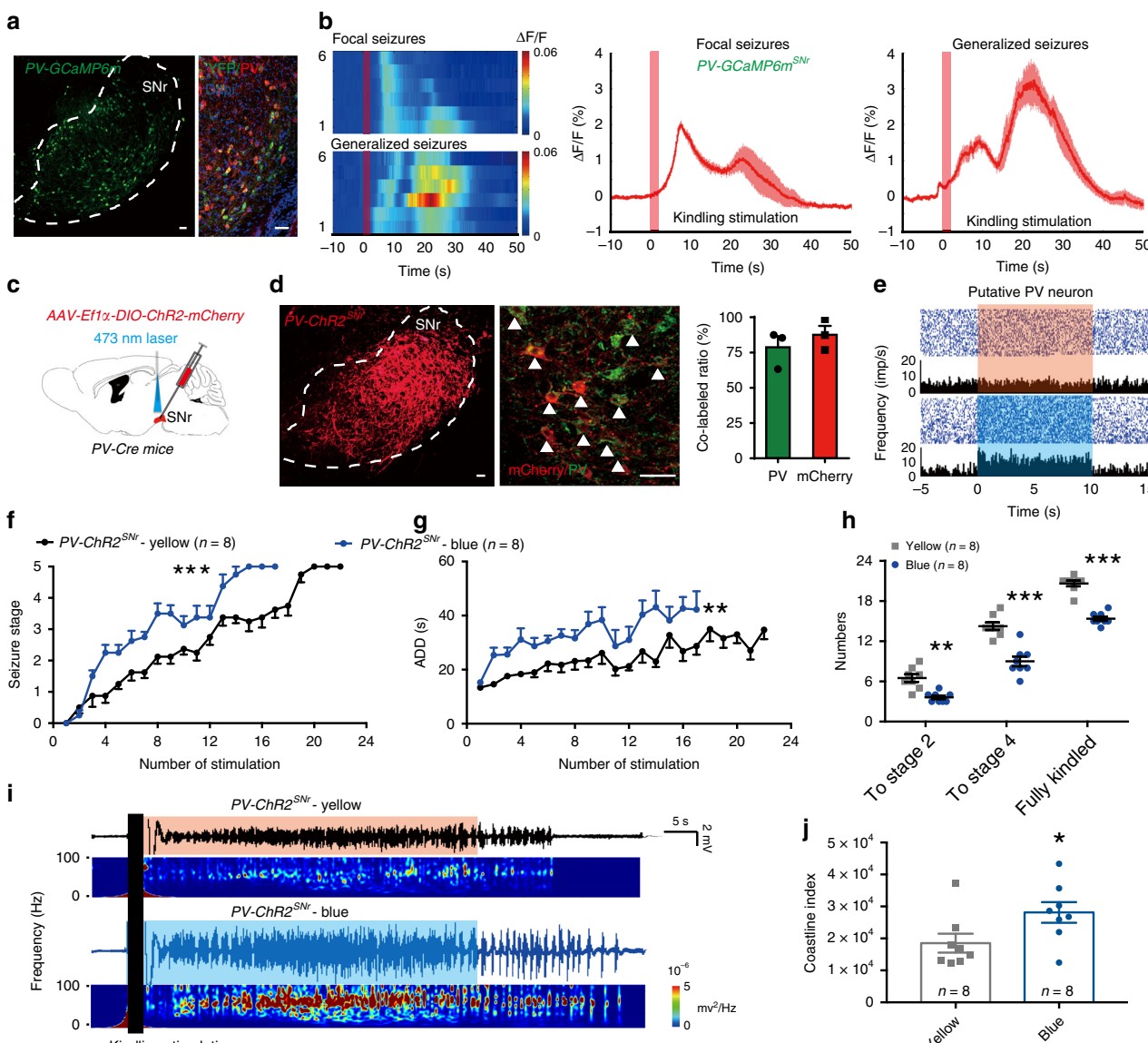

**Fig. 1 Optogenetic activation of SNr PV neurons amplifies kindling progression in hippocampal kindling model. a** Representative images of the SNr from a *PV-GCaMP6m*<sup>*SNr*</sup> mouse, showing the overlap of *GCaMP6m*-eYFP and PV-positive neurons. Scale bar, 50 μm. **b** Fiber photometry of SNr PV neural dynamics during hippocampal kindling seizures in *PV-GCaMP6m*<sup>*SNr*</sup> mice. Left, heatmap illustration of calcium signals aligned to the initiation of kindling stimulation. Each row represents the typical calcium signal of each mouse during focal seizures and generalized seizures, and a total of six mice are illustrated. Color scale indicates Δ*F/F* and warmer colors indicated higher fluorescence signal. Right, peri-event plots of the average calcium signals corresponding to the heatmaps. Thick lines indicated mean and shaded areas indicated SEM. The red rectangles represent kindling stimulation. **c** Scheme of experiment for viral injection and photostimulation in *PV-ChR2*<sup>*SNr*</sup> mice. **d** Representative images of the SNr from a *PV-ChR2*<sup>*SNr*</sup> mouse, showing the overlap in expression of ChR2-mCherry (red) and PV-positive neurons (green). Scale bar, 50 μm. Right, quantification of the percentage of ChR2-mCherry neurons that co-expressed PV neurons from three *PV-ChR2*<sup>*SNr*</sup> mice (repeated three times per mouse). **e** Representative peri-event raster histograms showing the firing rate of the same SNr PV neuron in response to blue-light and yellow-light stimulation. **f–h** Effects of optogenetic activation of SNr PV neurons on the development of seizure stage (**f**), afterdischarge duration (ADD, **g**), and number of stimulations needed to reach each seizure stage (**h**) in the hippocampal kindling model. Align-and-rank data for a nonparametric ANOVA was used for **f**, general linear model with repeated measures was used for **g**, and Mann–Whitney *U*-test was used for **h**; \*\**P* < 0.01, \*\*\**P* < 0.001 compared with yellow-light control. **i, j** Representative EEGs, corresponding EEG spectra power (**i**), and the quantification of EEG coastline index (**j**) recorded from the CA3 during the fully kindled state. The black rectangle represents kindling stimulation. Unpaired *t*-test was used for **j**; \**P* < 0.05, compared with yellow-light control. The number of mice used in each group is indicated in the figure. Data are presented as means ± SEM. Source data are provided as a Source Data file.

prolonged the afterdischarge duration (ADD; Supplementary Fig. 1h-2), and led to fewer stimulations being needed to reach stage 4 and the fully kindled state (Supplementary Fig. 1h-3) during kindling acquisition, compared with the yellow-light control. The pro-epileptic effect was only achieved when the photostimulation was delivered immediately after kindling

stimulation, suggesting a seizure state-dependent property (Supplementary Fig. 1h). The photo-activation of SNr GABAergic neurons did not change the initial afterdischarge threshold (ADT; Supplementary Fig. 1i), suggesting that photostimulation did not change the epileptic sensitivity, but mainly facilitated the process of kindling progression. These results indicate that activation of

SNr GABAergic neurons is sufficient to amplify seizure activities in hippocampal kindling model.

To further investigate the role of SNr PV neurons in seizure of TLE, we introduced AAV-Ef1α-DIO-ChR2-mCherry into the SNr of *PV-Cre* mice (referred to as *PV-ChR2$^{SNr}$* mice; Fig. 1c). Histological data confirmed that ChR2-mCherry was mainly expressed in the SNr PV$^+$ neurons ($78.71 \pm 7.76\%$ of PV$^+$ neurons expressed mCherry and $87.58 \pm 6.30\%$ of mCherry$^+$ neurons were PV$^+$ from 3 *PV-ChR2$^{SNr}$* mice; Fig. 1d). In vivo single-unit recordings confirmed that blue-light stimulation excited putative PV neurons (10 out of 22 neurons from 3 *PV-ChR2$^{SNr}$* mice), whereas yellow-light stimulation had no effect on the same neurons (Fig. 1e). In the hippocampal kindling model, photo-activation of SNr PV neurons in *PV-ChR2$^{SNr}$* mice accelerated the development of seizure stage (Fig. 1f), prolonged ADD (Fig. 1g), and led to fewer stimulations being needed to reach stage 2, stage 4, and the fully kindled state (Fig. 1h) during kindling acquisition, compared with the yellow-light control. The photo-activation of SNr PV neurons significantly increased the

coastline index of EEG during seizures (Fig. 1i–j). In addition, we also explored the role of SNr nitric oxide synthase (NOS) subtype GABAergic neurons in TLE, which is the second largest population of GABAergic neurons in the SNr[30]. Interestingly, we found that optogenetic activation of SNr NOS$^+$ neurons in *NOS-ChR2$^{SNr}$* mice had no effects on kindling progression in TLE (Supplementary Fig. 3). Thus, the above data demonstrate that driving SNr GABAergic neurons including PV neurons amplifies seizure activities in hippocampal kindling model.

**Inhibition of SNr PV neurons alleviates seizure activities.** To further test whether PV neurons are required for kindling progression in TLE, we introduced AAV-CAG-FLEX-ArchT-GFP into the SNr of *PV-Cre* mice (referred to as *PV-Arch$^{SNr}$* mice; Fig. 2a) to selectively inhibit SNr PV neurons. Histological data confirmed that Arch-GFP was mainly expressed in the SNr and colocalized with PV ($61.00 \pm 8.16\%$ of PV$^+$ neurons expressed green fluorescent protein (GFP) and $84.27 \pm 11.34\%$ of GFP$^+$ neurons were PV$^+$ from 3 *PV-Arch$^{SNr}$* mice; Fig. 2b). In vivo single-unit

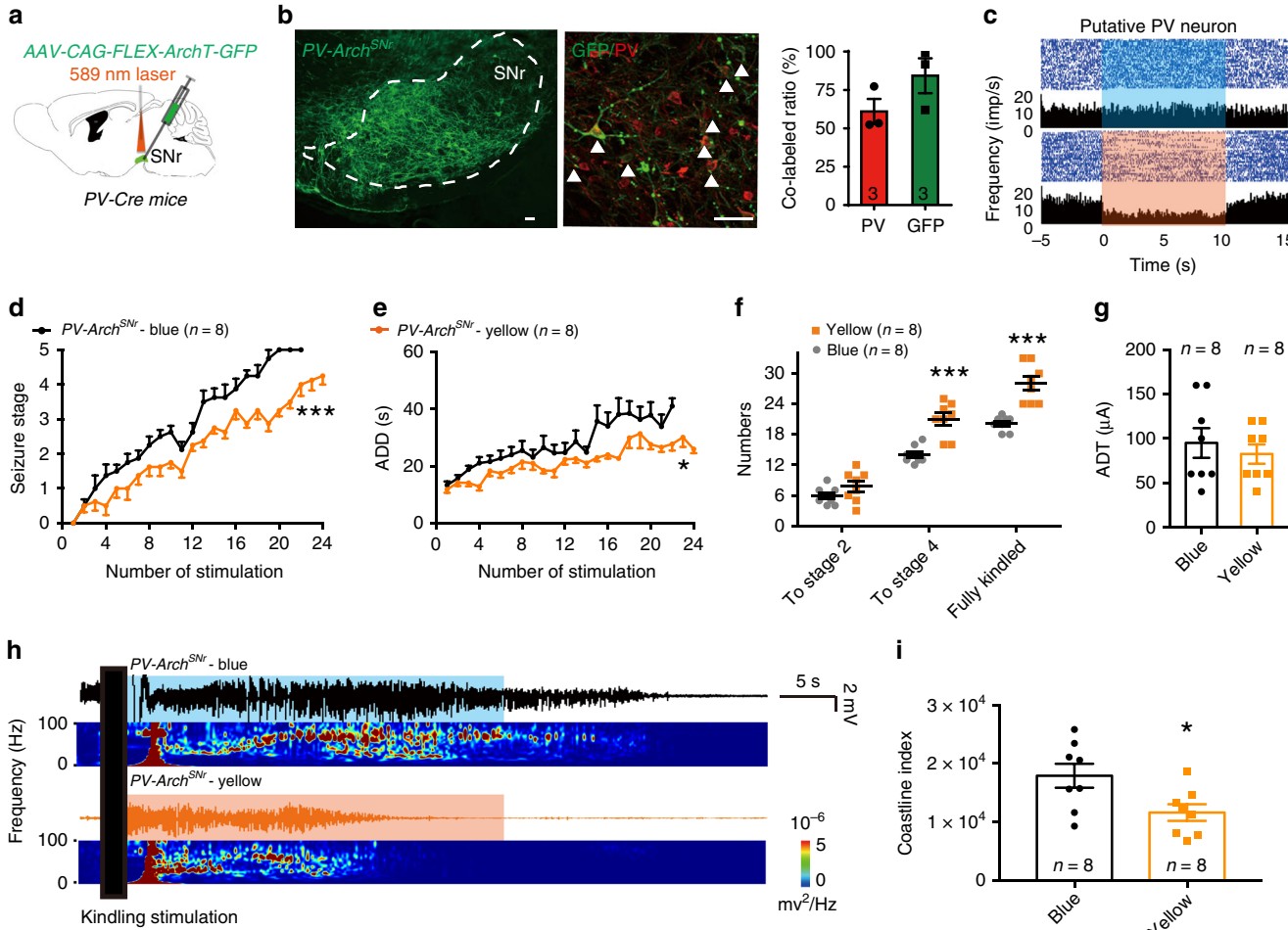

**Fig. 2 Optogenetic inhibition of SNr PV neurons retards kindling progression in hippocampal kindling model. a** Scheme of experiment for viral injection and photostimulation in *PV-Arch$^{SNr}$* mice. **b** Representative images of the SNr from a *PV-Arch$^{SNr}$* mouse, showing the overlap of GFP and PV neurons (red). Scale bar, 50 μm. Right, quantification of the percentage of GFP neurons that co-expressed PV neurons from three *PV-Arch$^{SNr}$* mice (repeated three times per mouse). **c** Representative peri-event raster histograms showing the firing rate of the same SNr PV neuron in response to yellow-light and blue-light stimulation. **d–g** Effects of optogenetic inhibition of SNr PV neurons on the development of seizure stage (**d**), afterdischarge duration (ADD, **e**), number of stimulations needed to each seizure stage (**f**), and afterdischarge threshold (ADT, **g**) in the hippocampal kindling model. Align-and-rank data for a nonparametric ANOVA was used for **d**, general linear model with repeated measures was used for **e**, and Mann–Whitney U-test was used for **f**; *$P < 0.05$, ***$P < 0.001$ compared with blue-light control. **h**, **i** Representative EEGs, corresponding EEG spectra power (**h**), and quantification of EEG coastline index (**i**) recorded from CA3 during the fully kindled state. The black rectangle represented the kindling stimulation. Unpaired t-test was used; *$P < 0.05$ compared with blue-light control. The number of mice used in each group is indicated in the figure. Data are presented as means ± SEM. Source data are provided as a Source Data file.

recordings confirmed that yellow-light stimulation (589 nm, continuous light, 5 mW, 10 s on–off cycle), but not blue-light stimulation (473 nm, continuous light, 5 mW, 10 s on–off cycle, the control light), inhibited the firing rate of putative PV neurons (7 out of 16 neurons from 3 $PV-Arch^{SNr}$ mice; Fig. 2c). We found that photo-inhibition of SNr PV neurons of $PV-Arch^{SNr}$ mice retarded the development of seizure stage (Fig. 2d), shortened ADD (Fig. 2e, h, i), and led to more stimulations being needed to reach stage 4 and the fully kindled state during kindling acquisition compared with the blue-light control (Fig. 2f). The number of stimulations needed to reach stage 2 was not affected here. This was in contrast to the reduced number of stimulations required to reach stage 2 in optogenetic activation experiments. Although we do not know the exact causes of this discrepancy yet, we speculate that there might be a different efficacy between optogenetic activation by ChR2 and optogenetic inactivation by Arch, which possibly resulted from different viral infection rates in the SNr. The photo-inhibition of SNr PV neurons significantly decreased the coastline index of EEG during seizures (Fig. 2h, i), but it did not change the ADT (Fig. 2g). The finding that photo-inhibition of SNr PV neurons effectively reduced kindling progression in the hippocampal kindling model drove us to investigate whether inhibition of SNr PV neurons can also alleviate the severity of seizures in other TLE models.

Then we adopted the classical kainic acid (KA)-induced TLE model, in which status epilepticus (SE) is induced via intrahippocampal injection of KA and spontaneous recurrent seizures typically appear after several weeks[34]. In this model, we used a chemogenetic method to long-termly suppress the activity of SNr PV neurons. We injected AAV-Ef1α-DIO-hM4Di-mCherry virus in the SNr of $PV-Cre$ mice (referred to as $PV-hM4Di^{SNr}$ mice, 68.71 ± 2.88% of $PV^+$ neurons expressed mCherry, and 92.24 ± 2.92% of $mCherry^+$ neurons were $PV^+$; Fig. 3a). This virus expresses hM4Di, an engineered inhibitory G-protein-coupled receptor, which is sensitive to the orally bioavailable and normally inert metabolite of clozapine, clozapine-$N$-oxide (CNO), and thus was used to suppress neuronal activity[35]. First, we aimed to observe whether chemogenetic inhibition of SNr PV neurons can alleviate KA-induced acute seizures. $PV-hM4Di^{SNr}$ mice received CNO (intraperitoneal (i.p.) 1 mg/kg in saline vehicle) half an hour before KA injection (0.5 µg/µL, 0.6 µL) via the implanted cannula guide into the dorsal hippocampcal CA1. $PV-hM4Di^{SNr}$ mice treated with saline or $PV-Cre$ mice treated with CNO served as controls. Chemogenetic suppression of SNr PV neurons substantially retarded seizure progression, manifested by prolonging the latency to GS (Fig. 3b), prolonging the latency to SE (Fig. 3c), reducing the number of FSs (Fig. 3d) and GSs (Fig. 3e). EEG spectra power after KA injection quickly increased in the $PV-Cre$ with CNO group and $PV-hM4Di^{SNr}$ with saline group, but significantly reduced and delayed in the $PV-hM4Di^{SNr}$ with CNO group (Fig. 3h).

Second, we aimed to test whether chemogenetic inhibition of SNr PV neurons can alleviate KA-induced chronic recurrent spontaneous seizures (Fig. 3g, h). The 7-day treatment of CNO consistently reduced the number of spontaneous GSs and FSs, and shortened the total seizure duration of GSs and FSs in chronic epileptic $PV-hM4Di^{SNr}$ mice during a 10 h observation period each day (Fig. 3i, j). CNO itself did not change the number and seizure duration of both FSs and GSs in P$V-mCherry^{SNr}$ mice (Fig. 3k, l), suggesting anti-seizure effect of CNO in $PV-hM4Di^{SNr}$ mice may not be associated with off-target effects of CNO. Conversely, chemogenetic activation of SNr PV neurons in chronic epileptic $PV-hM3Dq^{SNr}$ mice (AAV-Ef1α-DIO-hM3Dq-mCherry virus was injected into the SNr of $PV-Cre$ mice) increased the number of FSs and GSs, and prolonged the total seizure duration of FSs and GSs (Fig. 3m–o). Even worse, two

$PV-hM3Dq^{SNr}$ mice died after CNO treatment. All of the above data show that SNr PV neurons bidirectionally regulate the severity of seizure in TLE models, activation of PV neurons amplifies seizure, whereas inhibition of PV neurons alleviates seizure.

**SNr-PF PV projections regulate seizure in TLE.** Next, we aimed to test how SNr PV neurons are involved in kindling progression of TLE. We injected the retrograde tracer cholera toxin B 488 (CTB-488) into either the hippocampal CA3 or the SNr of wild-type mice. No CTB-labeled neuron was observed either in the SNr (Supplementary Fig. 4a) or the hippocampus (Supplementary Fig. 4b), suggesting that there is no direct projection between the SNr and the hippocampus. As SNr PV-projecting neurons convey the primary efferents to thalamic nuclei[36–38], we suspected that inhibition of GABAergic nigral neurons in the SNr might cause disinhibition of thalamic neurons, which in turn influence seizure activity in TLE. In $PV-Arch^{SNr}$ mice, green GFP fluorescence was apparent at soma in the SNr and at the projections within the thalamus (Supplementary Fig. 4c). The prominent GFP-expressing terminals were visible in the ventromedial thalamic nucleus (VM), the PF, and the reticularis nucleus (RT), and there was sparse fiber labeling in the reuniens nucleus (RE) and some other regions. To assess which downstream pathway mediates the function of SNr PV neurons in TLE, we assessed the behavioral consequence of inhibiting SNr PV fibers innervating the VM, PF, RT, and RE in the hippocampal kindling model.

We implanted optical probes in the PF (referred to as $PV-Arch^{SNr-PF}$ mice, Fig. 4a), RT (referred to as $PV-Arch^{SNr-RT}$ mice; Supplementary Fig. 5a), RE (referred to as $PV-Arch^{SNr-RE}$ mice; Supplementary Fig. 5c), and VM (referred to as $PV-Arch^{SNr-VM}$ mice; Supplementary Fig. 5e) to photo-stimulate SNr PV-expressing fibers in each region. Photo-inhibition of the SNr-PF PV projections showed inhibitory effects on the kindling progression. These effects included retarding the development of seizure stage (Fig. 4c), shortened ADD (Fig. 4d), and led to more stimulations being needed to reach stage 2, stage 4, and the fully kindled state (Fig. 4e) during kindling acquisition. No effect was found when photo-inhibiting PV projections of the SNr-RT (Supplementary Fig. 5b), SNr-RE (Supplementary Fig. 5d), or SNr-VM (Supplementary Fig. 5f). As a positive control for effectiveness of optogenetic modulation, we found that the inhibition of SNr-VM circuit increased the clonic motor phase in seizure stage 3 of the kindling seizures (Supplementary Fig. 5g), which is consistent with previous studies that the VM is an important site for motor function[39].

Photo-activation of the SNr-PF PV projections (referred to as $PV-ChR2^{SNr-PF}$ mice; Fig. 4f) promoted kindling progression in TLE (Fig. 4g–i) compared with yellow-light control and this pro-epileptic effect of photo-activation was reversed by intra-PF application of the GABA$_A$ receptor antagonist bicuculline (Fig. 4g–i), suggesting that the pro-epileptic effect was mediated directly by the SNr-PF PV projections, but not by passing fibers. Thus, the above data revealed that the inhibitory output from SNr PV neurons to the thalamic PF, but not to the RT, RE, or VM, is involved in seizure of TLE.

**PF GABAergic neurons are innervated by SNr PV projections.** To understand how SNr PV neurons innervate PF neurons and thus participate in seizure of TLE, we examined the organization of SNr inhibitory circuitry in the PF. The PF is known to be composed of glutamatergic cells with long-range projections[40]. Interestingly, fluorescence in situ hybridization of a series of coronal sections found both $GAD^+$ and $Vglut2^+$ neurons in the PF. The proportion of GAD/Vglut2 was much higher in the

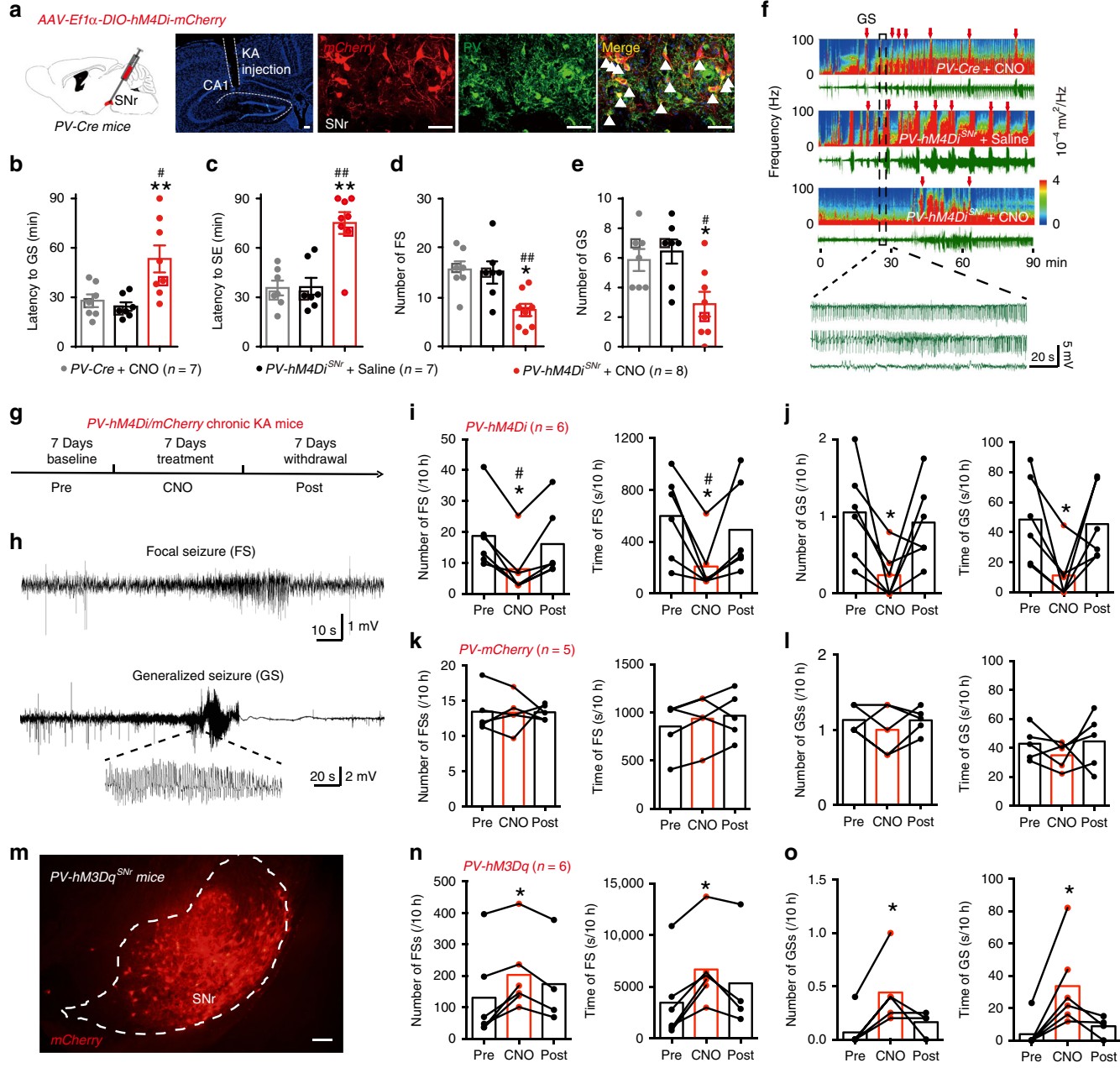

**Fig. 3 SNr PV neurons bidirectionally modulate the severity of epileptic seizures in kainic acid (KA)-induced spontaneous seizure model. a** Left, scheme of experiment for viral injection in *PV-hM4Di^SNr^* mice. Middle, coronal section showing the placement of guide cannula in dorsal hippocampal CA1 for kainic acid injection (Blue, DAPI). Right, representative images of the SNr from a *PV-hM4Di^SNr^* mouse, showing the overlap of mCherry (red) and PV neurons (green). Scale bar, 50 μm. **b–e** Effects of chemogenetic inhibition of SNr PV neurons on the latency to generalized seizure (GS, **b**), latency to status epilepticus (SE, **c**), the number of focal seizures (FSs, **d**), and GSs (**e**) in KA-induced acute seizure model. Kruskal–Wallis test with post-hoc Dunn's test for multiple comparisons; *$P < 0.05$, **$P < 0.01$ compared with *PV-hM4Di^SNr^* with saline; #$P < 0.05$, ##$P < 0.01$, compared with *PV-Cre* with CNO. **f** Representative EEG spectra power and raw EEG after KA injection; below are enlarged raw EEG segments corresponding to the dotted boxes. Red arrowheads indicate GS onsets. **g** Scheme of experiment for CNO treatment in KA-induced chronic epileptic *PV-hM4Di^SNr^* mice. **h** Typical spontaneous FS and GS in chronic epileptic *PV-hM4Di^SNr^* mice. **i, j** Effects of chemogenetic inhibition of SNr PV neurons on the number and duration of FSs (**i**) and GSs (**j**) in KA-induced chronic epileptic model. Wilcoxon's matched-pairs test, *$P < 0.05$ compared with Pre; #$P < 0.05$ compared with Post. **k, l** Effects of CNO treatment on the number and duration of FSs (**k**) and GSs (**l**) in KA-induced chronic epileptic *PV-mCherry^SNr^* mice. **m** Representative image of the SNr from a *PV-hM3Dq^SNr^* mouse showing the expression of mCherry (red). Scale bar, 50 μm. **n, o** Effects of chemogenetic activation of SNr PV neurons on the number and duration of FSs (**n**) and GSs (**o**) in KA-induced chronic epileptic *PV-hM3Dq^SNr^* mice. Wilcoxon's matched-pairs test, *$P < 0.05$ compared with Pre. The number of mice used in each group is indicated in the figure. Data are presented as means ± SEM. Source data are provided as a Source Data file.

posterior PF and the average proportion of GAD/Vglut2 in the whole PF was about 5% (Supplementary Fig. S6a, b), suggesting that there are at least two types of neurons in the PF, specially a few GABAergic neurons in the posterior PF. Meanwhile, we

compared the in vitro electrophysiological properties of these two cell populations in the posterior PF by transducing PF GABAergic neurons with control eYFP in Vgat-Cre-recombinase (*Vgat-Cre*) mice. GABAergic neurons were identified by the

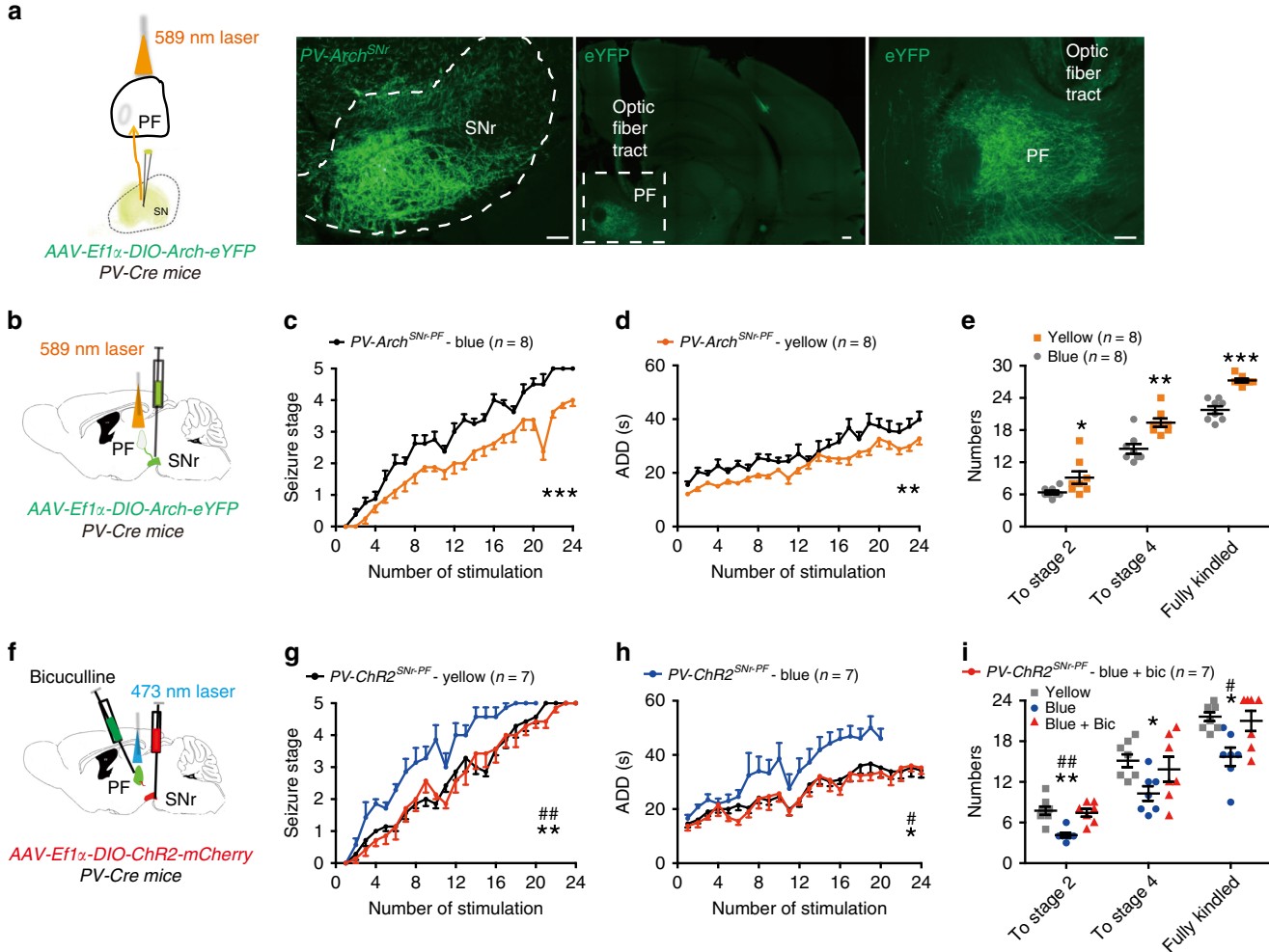

**Fig. 4 SNr-PF PV projections are involved in kindling progression. a** Representative images showing the expression of Arch-GFP soma in the SNr and Arch-GFP axon fibers within the PF from a *PV-Arch^SNr* mouse. Scale bar, 50 μm. **b** Scheme of experiment for optogenetic inhibition of the SNr-PF PV projections in *PV-Arch^SNr* mice. **c-e** Effects of optogenetic inhibition of SNr-PF PV projections on the development of seizure stage (**c**), afterdischarge duration (ADD, **d**), and number of stimulations needed to reach each seizure stage (**e**) in the hippocampal kindling model. Align-and-rank data for a nonparametric ANOVA was used for **c**, general linear model with repeated measures was used for **d**, and Mann–Whitney *U*-test was used for **e**; *$P < 0.05$, **$P < 0.01$, ***$P < 0.001$ compared with blue-light control. **f** Scheme of experiment for optogenetic activation of the SNr-PF PV projections in *PV-ChR2^SNr* mice. **g-i** Effects of optogenetic activation of SNr-PF PV projections on the development of seizure stage (**g**), ADD (**h**), and number of stimulations needed to reach each seizure stage (**i**) in the hippocampal kindling model, in the presence of GABA_A receptor antagonist bicuculline (Bic, intra-PF injection). Align-and-rank data for a nonparametric ANOVA with post-hoc Dunn's test for multiple comparisons was used in **g**, general linear model with post-hoc Dunn's test for multiple comparisons was used in **h**, Kruskal–Wallis test with post-hoc Dunn's test for multiple comparisons was used in **i**; *$P < 0.05$, **$P < 0.01$ compared with yellow-light control; #$P < 0.05$, ##$P < 0.01$ compared with blue-light with Bic. The number of mice used in each group is indicated in figure. Data are presented as means ± SEM. Source data are provided as a Source Data file.

expression of eYFP fluorescence, whereas putative glutamatergic neurons were identified by the feature of non-eYFP fluorescence. We found that action potentials of PF eYFP-expressing GABAergic neurons were much faster in kinetics than that of non-eYFP-expressing glutamatergic neurons (Supplementary Fig. 6c–h), suggesting two functionally different types of neurons in the posterior PF.

Then, to test the functional connection of the SNr and PF, we recorded the activity of PF neurons during blue-light activation of PV axons in the PF of *PV-ChR2^SNr* mice (Fig. 5a). We found that photo-activation of SNr-PF PV axons reliably and quickly inhibited the firing rate of 6/12 putative GABAergic neurons recorded in the PF (Fig. 5b). The firing rate of PF glutamatergic neurons in response to photo-activation of SNr-PF PV axons was heterogeneous: 5/26 neurons were decreased, 6/26 neurons were increased, and 15/26 neurons had no change from 6

*PV-ChR2^SNr-PF* mice (no inter-animal statistical difference, $\chi^2$-test, $p = 0.5875$; Fig. 5b). This data suggested that there might be direct and indirect neural connection between SNr PV neuron and PF glutamatergic neurons. In KA-induced chronic awake mice, although there is a minor reduce in the number of SNr PV neurons (Supplementary Fig. 7a, b), we found that optogenetic activation of SNr-PF PV projections can still inhibit 2/4 putative GABAergic neurons and activate a part of putative glutamatergic neurons (10/24 from 3 mice) reliably during the whole photostimulation period (Supplementary Fig. 7c–e). To clarify whether SNr PV neurons directly project to PF GABAergic neurons, we transduced SNr GABAergic neurons with ChR2-mCherry and PF GABAergic neurons with control eYFP in *Vgat-Cre* mice, and performed ChR2-assisted circuit mapping in acute brain slices containing PF using in vitro electrophysiology (Fig. 5c, d). Here we used *Vgat-cre* mice instead, as PV neurons

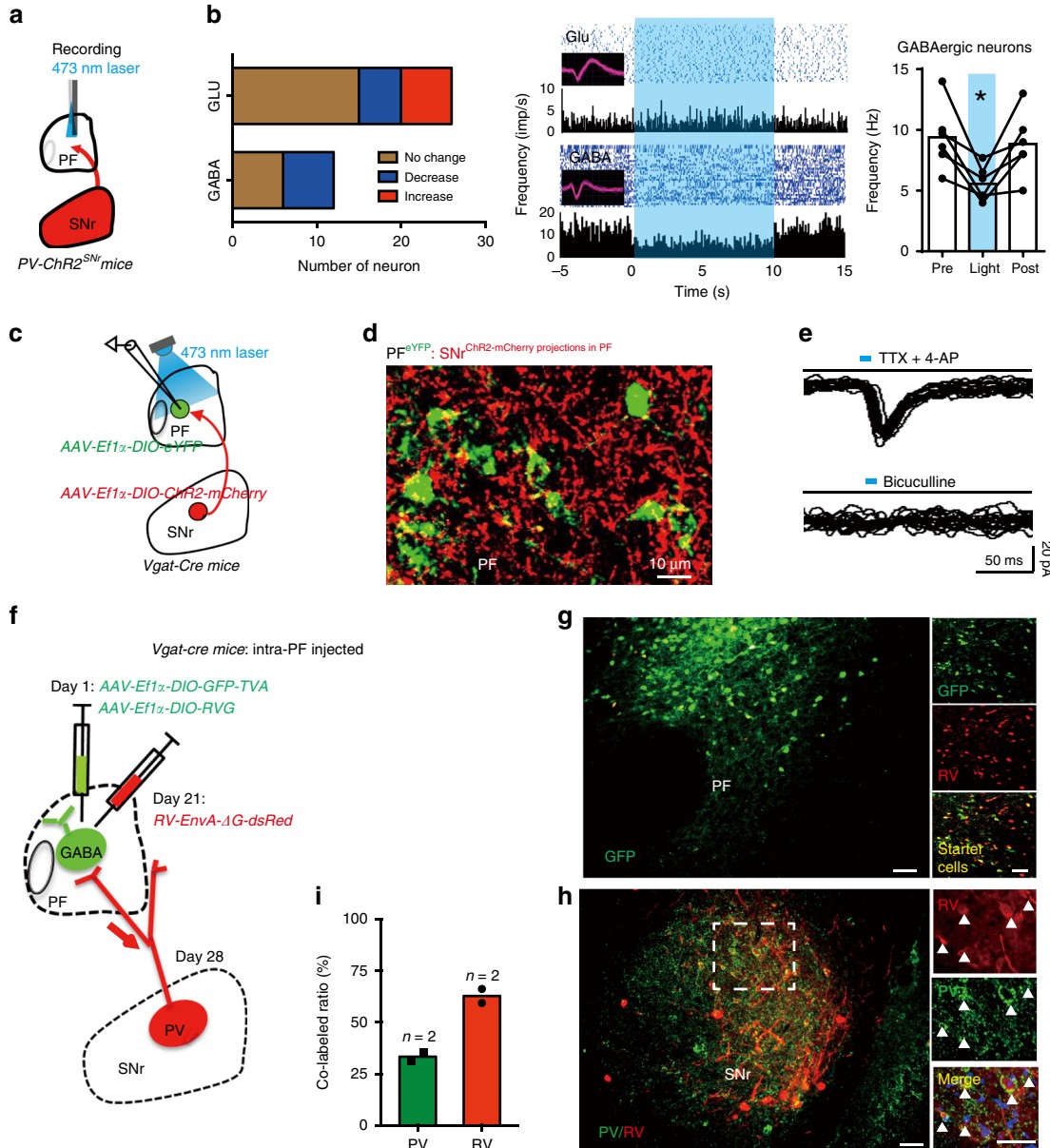

**Fig. 5 SNr PV neurons directly innervate PF GABAergic neurons. a** Scheme of experiment for in vivo single-unit recordings in the PF of *PV-ChR2^SNr* mice. **b** Left, quantification of the number of PF GABAergic and glutamatergic neurons in response to photo-activation of SNr-PF PV projections from six *PV-ChR2^SNr-PF* mice. Middle, representative peri-event raster histograms showing the firing rate of PF GABAergic and glutamatergic neurons in response to photostimulation. Right, blue-light stimulation reduced the firing rate of six responsive PF GABAergic neurons. Wilcoxon's matched-pairs test, *$P < 0.05$ compared with Pre. **c** Scheme of experiment for in vitro electrophysiology in the acute brain slices containing PF of *Vgat-Cre* mice. **d** Representative image showing SNr-PF GABAergic projections labeled with ChR2-mCherry (red) and PF GABAergic neurons labeled with eYFP (green). **e** ChR2-assisted circuit mapping. Light-evoked inhibitory postsynaptic currents (IPSCs) were recorded in PF GABAergic neurons (3 out of 17 neurons) during photostimulation (473 nm, 1 Hz, 10 ms, 10 pulses, 2 mW) of ChR2-expressing SNr-PF GABAergic projections in PF in the presence of tetrodotoxin (TTX, 1 µM) and 4-amynopyridine (4-AP, 100 µM). IPSCs were diminished in the presence of the GABA_A receptor antagonist bicuculline (20 µM). **f** Schematic of the experimental design for PF retrograde monosynaptic tracing in *Vgat-Cre* mice using a modified rabies virus system. **g** Representative images of the PF injected with the helper virus and rabies virus; right, higher-magnification images showing PF GABAergic neurons infected by the helper virus (GFP), rabies virus (mCherry), or both viruses. The double-infected (yellow) GABAergic cells represent the starter cells. Scale bar, 50 µm. **h** Representative images of the SNr from the same mouse as that in **g**, showing the labeling of neurons by the monosynaptic retrograde spread of rabies virus expressing mCherry; right, higher-magnification images showing the overlap expression of rabies virus (mCherry) and PV neurons (green). Scale bar, 50 µm. **i** Quantification of the percentage of rabies virus that co-expressed PV neurons in the SNr. The number of mice used in each group is indicated in figure. Data are presented as means ± SEM. Source data are provided as a Source Data file.

present a large proportion of SNr GABAergic neurons, and optogenetic activation of SNr GABAergic neurons and PV neurons produced similar effects on kindling progression. GABA-mediated synaptic currents were evoked by flashing blue light during whole-cell recording from PF GABAergic neurons labeled with eYFP (3 out of 17 neurons). Experiments performed in the presence of tetrodotoxin (TTX) and 4-amynopyridine (4-AP), which were used to block action potential-dependent

synaptic transmission in indirect circuit[41,42], demonstrated the direct monosynaptic input from the SNr to the PF. The synaptic currents were eliminated when the GABAergic receptor blocker bicuculline was applied (Fig. 5e), confirming the mediation of GABA. The low percentage of PF GABAergic neurons receiving projections from SNr may relate to the following three reasons: (1) SNr PV neurons may project to a small part of the posterior PF GABAergic neurons. (2) The efficacy of virus-mediated ChR2 expression at the presynaptic part is low and the light level is not able to induce GABAergic responses in postsynaptic PF GABAergic neurons. (3) Sub-optimal recording conditions: to isolate monosynaptic current, experiments were performed in the presence of TTX and 4-AP to block indirect circuit.

Further, we used retrograde tracing to confirm that posterior PF GABAergic neurons receive direct SNr PV innervation. Specifically, we used a modified rabies virus system that can trace monosynaptic inputs onto genetically identified neurons[43]. We injected AAV-Ef1α-DIO-GFP-TVA and AAV-Ef1α-DIO-RVG (1:1) on day 1 and dsRed-expressing rabies virus (RV-EnvA-ΔG-dsRed, RV) on day 21 into the same location of the PF in Vgat-Cre mice (Fig. 5f). On day 28, the GFP/mCherry double-positive cells observed in the PF represented the GABAergic starter cells in Vgat-Cre mice (Fig. 5g). Brain sections from the same mice showed trans-synaptic labeling of red fluorescent rabies–RV-infected cells in the ipsilateral SNr of Vgat-Cre mice (Fig. 5h). It is important to note that a large population of retrogradely labeled neurons in the SNr were PV+ neurons (62.83 ± 3.23% of RV in the SNr were PV+ neurons from 2 Vgat-Cre mice; Fig. 5i), which accounted for ~30% of all PV neurons in the SNr. In addition, we performed retrograde virus tracing from PF glutamatergic neurons of Vglut2-Cre mice to test whether SNr PV neurons directly project to the PF glutamatergic neurons. Quantity analysis showed the number of SNr PV neurons targeting PF glutamatergic neurons was much lower than that targeting PF GABAergic neurons (Supplementary Fig. 8). Together, these anatomical and functional evidence indicate that a portion of SNr PV neurons directly innervate the posterior PF GABAergic neurons.

**PF GABAergic neurons bidirectionally regulate seizures**. To further verify the regulation of seizure in TLE by the hypothetical "SNr PV-PF GABAergic neurons" circuit, we directly tested the involvement of posterior PF GABAergic neurons in seizure. We transduced either AAV-CAG-FLEX-ArchT-GFP (Fig. 6a) or AAV-Ef1α-DIO-ChR2-mCherry (Fig. 6g) into the posterior PF of Vgat-Cre mice. Histological data confirmed that either Arch-GFP or ChR2-mCherry was mainly expressed in the PF (Fig. 6b, h), suggesting that PF GABAergic neurons are local, but not projecting neurons. In vivo single-unit recordings confirmed that yellow-light stimulation inhibited PF GABAergic neurons in Vgat-Arch$^{PF}$ mice (8 out of 17 neurons from 3 Vgat-Arch$^{PF}$ mice; Fig. 6c) and blue-light stimulation excited PF GABAergic neurons in Vgat-ChR2$^{PF}$ mice (11 out of 25 neurons from 3 Vgat-ChR2$^{PF}$ mice; Fig. 6i). In the hippocampal kindling model, photoinhibition of PF GABAergic neurons in Vgat-Arch$^{PF}$ mice accelerated the development of seizure stage (Fig. 6d), prolonged ADD (Fig. 6e), and led to fewer stimulations being needed to reach stage 4, and the fully kindled state (Fig. 6f) during kindling acquisition. In contrast, photo-activation of PF GABAergic neurons in Vgat-ChR2$^{PF}$ mice retarded the development of seizure stage (Fig. 6j), decreased ADD (Fig. 6k), and led to more stimulations being needed to reach stage 2, stage 4, and the fully kindled state (Fig. 6l) during kindling acquisition. This antiepileptic effect of photo-activation was reversed by intra-PF application of the GABA$_A$ receptor antagonist bicuculline (Fig. 6j–l). Although bicuculline is not cell-type specific and may

affect both glutamatergic and GABAergic cells in the PF, the above data at least suggested there is a local GABAergic microcircuit within the PF regulating seizure. Meanwhile, chemogenetic activation of PF GABAergic neurons in Vgat-hM3Dq$^{PF}$ mice also alleviated the severity of seizure in the KA-induced acute seizure model (Fig. 6m-q). Furthermore, chemogenetic inhibition of SNr-PF PV projections by intra-PF injection of CNO or chemogenetic activation of PF GABAergic neurons similarly alleviated the severity of seizure in KA-induced chronic epileptic model (Supplementary Fig. 7f–i).

To further test whether the PF GABAergic neurons are required for the role of SNr in seizure modulation, we chemogenetically activated PF GABAergic neurons before optogenetically activating SNr GABAergic neurons (Supplementary Fig. 9a). Histological data confirmed that ChR2-eYFP was apparent at soma in the SNr (Supplementary Fig. 9b) with projections around the hM3Dq-mCherry-labeled GABAergic neurons in the PF (Supplementary Fig. 9c). We found that chemogenetic activation of PF GABAergic neuron reversed the pro-epileptic effect of optogenetic-activating SNr GABAergic neurons on kindling progression (Supplementary Fig. 9d). In addition, we found that chemogenetic modulation of "SNr PV-PF GABAergic neurons" circuit at tested intensity (1.0 mg/kg CNO) did not interfere with motor learning or motor functions evaluated by the rotarod test and open-field test (Supplementary Fig. 10). Chemogenetic inhibition of SNr PV neurons by a high dose of CNO (10 mg/kg) increased the number of entries into the center and had a tendency to increase the travel distance in the center (Supplementary Fig. 10d), but did not change motor coordination in the rotarod test (Supplementary Fig. 10c). These findings were similar to previous finding that activating the direct striatonigral pathway, which inhibits SNr neurons, can decrease freezing and increase locomotor initiations without changing the coordination of ambulation itself[44]. These data suggested that inhibition intensity that we used here to block seizure is at the lower range for SNr modulating motor function. Taken together, our results demonstrated that PF GABAergic neurons bidirectionally modulate seizure activities in hippocampal kindling model, which mimics the role of PF-projecting PV fibers from the SNr in this model. It indicates that the "SNr PV-posterior PF GABAergic neurons" circuit is closely involved in the regulation of seizure in TLE.

In addition, we tested the balance between PF GABAergic and glutamatergic neurons in TLE. First, we used in vivo single-unit recordings to analyze the firing rate of PF GABAergic and glutamatergic neurons during kindling seizures in anesthetized mice. After CA3 kindling stimulation, about 54.5% of GABAergic neurons in the ipsilateral PF decreased their firing rate during seizures (1/21 neuron was excited, 11/21 neurons were inhibited, and 9/21 neurons were no response). Whereas, 50% of glutamatergic neurons in the ipsilateral PF increased their firing rate during seizures (16/33 neurons were excited, 1/33 neuron was inhibited, 15/33 neurons had no response, Fig. 7a–c), suggesting PF glutamatergic neurons may be indirectly disinhibited by SNr PV neuron and thus involved in seizure. Further, we found that optogenetic activation of PF GABAergic neurons inhibited surrounding glutamatergic neurons locally in the Vgat-ChR2$^{PF}$ mice (Fig. 7d), and direct optogentically inhibition of PF glutamatergic neurons produced an antiepileptic effect on seizure development in the hippocampal kindling model (Fig. 7e–h). These results suggested that PF GABAergic neurons produce an antiepileptic effect by inhibiting glutamatergic neurons locally. Furthermore, glutamatergic neurons were sufficient to accelerate seizure development in hippocampal kindling model when they were activated (Fig. 7i–l), suggesting PF glutamatergic neurons were a part of the circuit regulating seizure activities of TLE in a bidirectional manner (see summary diagram in Supplementary Fig. 11).

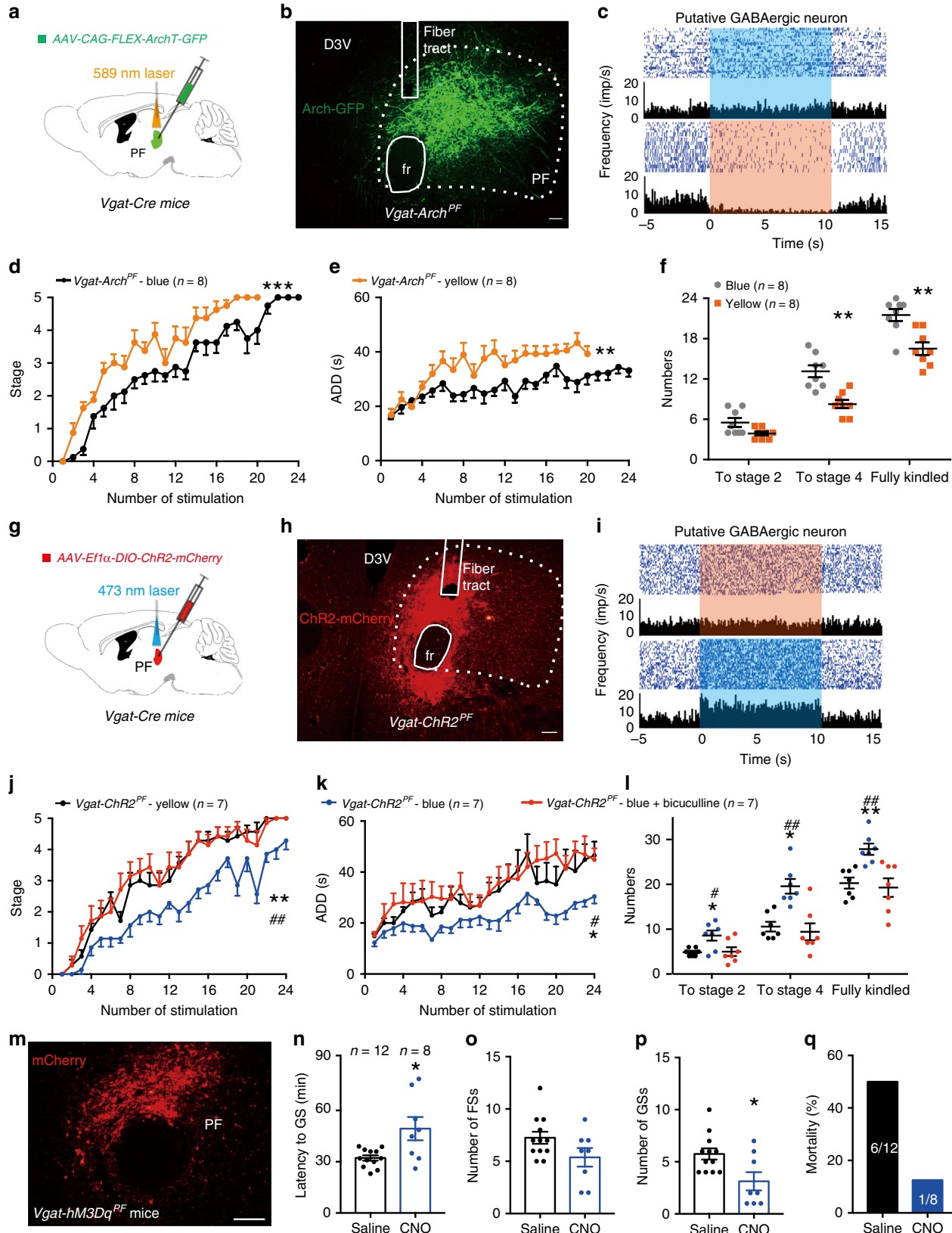

## Discussion

The abnormal network that arises during seizure activities in TLE has not yet been thoroughly investigated, especially the limbic brain regions that are distant from the seizure focus[45]. Our study aimed to establish the specific role of SNr GABAergic neurons and PV subtype neurons in TLE models. We provided two types

of evidence—in vivo real-time monitoring of cellular events using single-unit recordings and fiber photometry of calcium signals— showing that SNr GABAergic neurons, including PV neurons, were hyperactivated during hippocampal kindling seizures, indicating that SNr neurons are directly involved in seizure. Although previous reports have shown that the firing rate of SNr

**Fig. 6 PF GABAergic neurons bidirectionally regulate the severity of epileptic seizures. a, g** Schematic of experiment for viral injection and photostimulation in the PF of *Vgat-Arch*[PF] (**a**) and *Vgat-ChR2*[PF] (**g**) mice. **b, h** Representative images of the PF from *Vgat-Arch*[PF] (**b**) mouse showing the expression of Arch-GFP or *Vgat-ChR2*[PF] (**h**) mouse showing the expression of ChR2-mCherry. Scale bar, 50 μm. **c, i** Representative peri-event raster histograms showing the firing rate of a PF GABAergic neuron in response to yellow-light and blue-light stimulation in a *Vgat-Arch*[PF] (**c**) or *Vgat-ChR2*[PF] (**i**) mouse. **d–f** Effects of optogenetic inhibition of PF GABAergic neurons on the development of seizure stage (**d**), afterdischarge duration (ADD, **e**), and number of stimulations needed to reach each seizure stage (**f**) in hippocampal kindling model. Align-and-rank data for a nonparametric ANOVA was used for **d**, general linear model with repeated measures was used for **e**, and Mann–Whitney *U*-test was used for **f**; **$P < 0.01$, ***$P < 0.001$ compared with blue-light control. **j–l** Effects of optogenetic activation of PF GABAergic neurons on the development of seizure stage (**j**), ADD (**k**), and number of stimulations needed to reach each seizure stage (**l**) in the hippocampal kindling model, in the presence of GABA$_A$ receptor antagonist bicuculline (Bic, intra-PF injection). Align-and-rank data for a nonparametric ANOVA with post-hoc Dunn's test for multiple comparisons was used in **j**, general linear model with post-hoc Dunn's test for multiple comparisons was used in **k**, and an Kruskal–Wallis test with post-hoc Dunn's test for multiple comparisons was used in **l**; *$P < 0.05$, **$P < 0.01$ compared with yellow-light control and #$P < 0.05$, ##$P < 0.01$ compared with blue-light with bicuculline. **m** Representative image of the PF from a *Vgat-hM3Dq*[PF] mouse. Scale bar, 100 μm. **n–q** Effects of chemogenetic activation of PF GABAergic neurons on the latency to generalized seizure (GS, **n**), the number of focal seizures (FSs, **o**) and GSs (**p**), and mortality (**q**) in KA-induced acute seizure model. Mann–Whitney *U*-test, *$P < 0.05$ compared with saline. The number of mice used in each group is indicated in the figure. Data are presented as means ± SEM. Source data are provided as a Source Data file.

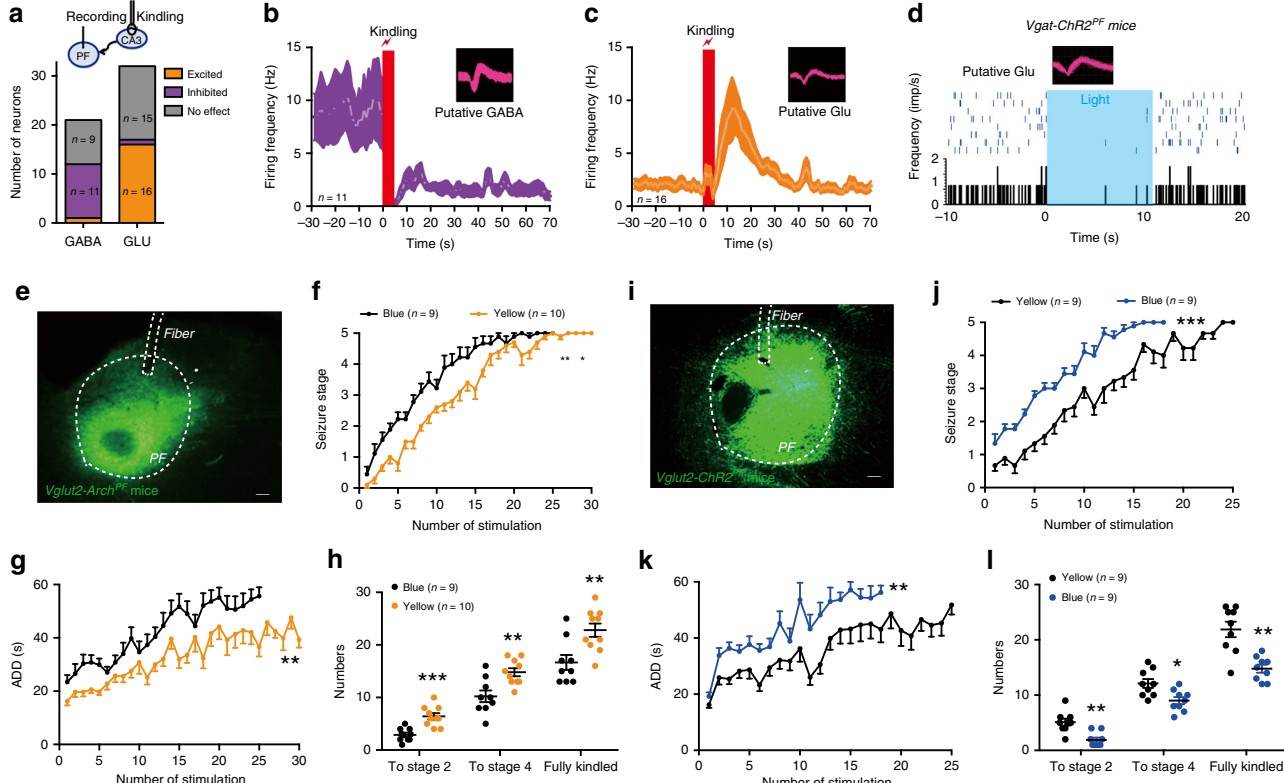

**Fig. 7 PF glutamatergic neurons bidirectionally modulate kindling progression in hippocampal kindling model. a** Above, scheme of experiment for kindling stimulation in the CA3 and in vivo single-unit recordings in the PF. Below, statistic of responses of recorded PF GABAergic and glutamatergic neurons during hippocampal kindling-induced seizures from three wild-type mice. **b, c** Average firing rates of responsive PF GABAergic (**b**) and glutamatergic neurons (**c**) during hippocampal kindling-induced seizures. **d** Representative peri-event raster histogram showing the firing of the PF glutamatergic neuron in response to blue-light stimulation in one *Vgat-ChR2*[PF] mouse. **e** Representative image of the PF from a *Vglut2-Arch*[PF] mouse. Scale bar, 100 μm. **f–h** Effects of optogenetic inhibition of PF glutamatergic neurons on the development of seizure stage (**f**), afterdischarge duration (ADD, **g**), and number of stimulations needed to each seizure stage (**h**) in the hippocampal kindling model. Align-and-rank data for a nonparametric ANOVA was used in **f**, general linear model with repeated measures was used in **g**, and Mann–Whitney *U*-test was used in **h**; **$P < 0.01$, ***$P < 0.001$ compared with blue-light control. **i** Representative image of the PF from a *Vglut2-ChR2*[PF] mouse. Scale bar, 100 μm. **j–l** Effects of optogenetic activation of PF glutamatergic neurons on the development of seizure stage (**j**), ADD (**k**), and number of stimulations needed to each seizure stage (**l**) in the hippocampal kindling model. Align-and-rank data for a nonparametric ANOVA was used in **j**, general linear model with repeated measures was used in **k**, and Mann–Whitney *U*-test was used in **l**; *$P < 0.05$, **$P < 0.01$, ***$P < 0.001$ compared with yellow-light control. The number of mice used in each group is indicated in the figure. All the data are presented as mean ± SEM. Source data are provided as a Source Data file.

GABAergic neurons significantly increased after a GS compared with naive controls[46], it is still unclear how this hyperactivated reaction of SNr GABAergic neurons influences seizure activities of TLE. Using optogenetics, we provided direct evidence that activation of SNr GABAergic neurons, including PV neurons, promoted kindling progression, whereas inhibition of these neurons alleviated the severity of epileptic seizures in the hippocampal kindling model. Furthermore, chemogenetic inhibition

of SNr PV neurons also produced an antiepileptic effect on chronic recurrent spontaneous seizures induced by intra-hippocampal KA injection, which is usually drug resistant to many current AEDs. These results indicate that the antiepileptic effect of inhibiting SNr PV neurons might be generally applicable to TLE. Previous reports have shown that AEDs, such as valproate, reduce the firing rate of SNr GABAergic neurons in TLE[17], which supports our finding that directly reducing the high firing rate of PV neurons during seizure can protect against epileptic seizures. It should be noted that not all regions of the SNr may be equivalent in their impact on seizure activity, as we can see appreciable variability in the response to optogenetic modulation in Figs. 1 and 2. Indeed, previous studies have reported that there is rostro-caudal differences of SNr in seizure control[47−49]. Therefore, to our knowledge, for the first time, our results clarified the causal role of SNr GABAergic neurons including PV neurons in seizure of TLE: activation of SNr GABAergic neurons including PV neurons amplified, whereas inhibition of SNr PV neurons retarded seizure activities.

We also investigated the underlying mechanism of SNr-related circuitry involved in seizure of kindling model. Although previous anatomical studies have shown that the SNr conveys primary outputs to thalamic nuclei[36−38], it is still unclear which specific thalamic nucleus or nuclei is (or are) involved in TLE. Our study provided the direct evidence that SNr PV neurons sent dense innervations into the RT, PF, and VM, but sent very sparse projections into the RE. We further examined the roles of SNr PV projection pathways in seizure in the above four regions (RT, PF, RE, and VM) and found that the SNr-PF PV projection, but not the SNr-RT, SNr-RE, and SNr-VM pathways, were involved in kindling progression. Surprisingly, although there is an indirect pathway from the SNr to the PF via the RT[50], we found that SNr-RT PV projections had no effect on kindling progression. These results indicated that the SNr PV projections were circuit-specific regulators of TLE. As a common issue in optogenetic and chemogenetic modulation, targeting hub regions, such as the SNr or PF here, may alter global brain dynamics. Therefore, apart from SNr-PF circuit, other potential global alterations may also be accounted for regulating seizure. For example, the superior colliculus, a primary target of SNr output, sends glutamatergic projections to the PF, which was previously reported to be involved in the control of absence seizures[51]. It is possible that the superior colliculus may also potentially have a role in seizure of TLE. Interestingly, SNr GABAergic neurons control absence epilepsy through the downstream VM thalamocortical circuit[9], whereas nigrothalamic/nigrostriatal projections do not appear to contribute to the electroshock convulsions[52], suggesting that basal ganglia play a role in various seizure disorders through distinct downstream circuits. Furthermore, using in vivo and in vitro electrophysiology combined with retrograde neural circuit tracing, we found that posterior PF GABAergic neurons received structural and functional inhibitory connections from SNr PV neurons. Only a small part of posterior PF GABAergic or glutamatergic neurons responded functionally to the modulation of the SNr, suggesting that specific subsets of PF GABAergic and glutamatergic neurons may mediate the bidirectional modulation kindling progression. In addition, our optogenetic experiments further indicate that photo-activation of PF GABAergic neurons suppressed kindling progression, whereas photo-inhibition of them promoted kindling progression; this mimics the effect of manipulating SNr-PF PV projections in seizure. It appears that this long-range inhibitory "SNr PV-posterior PF GABAergic neurons" circuitry is responsible for regulating seizure. It should be noted that the circuit we revealed here is only modulatory, as intervening this circuit did not abort or abolish seizures in both kindling and KA models. As SNr PV-projecting neurons convey

the primary inhibitory afferents to PF GABAergic neurons and promote kindling progression, the inhibition of SNr PV neurons would cause a disinhibition of nigral efferences in PF GABAergic neurons, which in turn retards kindling progression. Although the number of GABAergic neurons is much less than glutamatergic neurons in the PF, in the present study, we first reveal the function of these posterior PF GABAergic neurons in epilepsy.

As an important region of the intralaminar thalamic nuclei, the PF broadly connects with the cerebral cortex and limbic system[53−55], closely participating in the process of epileptic seizures[56,57]. In the present study, we found that modulation of the SNr-PF circuit did not change the ADT, an index of local neural excitability in the seizure focus, indicating this circuit may not affect seizure initiation but rather affect seizure spread. The SNr-PF circuit may act like an intrinsic feed-forward amplifier; it may respond to abnormal cortical information input through basal ganglia circuit during seizures (as demonstrated by our fiber photometry showing the increased SNr PV neural dynamics), and thus in turn regulate epileptic networks via thalamocortical-hippocampal circuits. Specifically, connections between the PF and the entorhinal cortex[55], the major source of cortical projections to the hippocampus[32,58], are likely involved in these modulatory effects. Although abnormalities in the PF still need to be clarified for a better understanding of the abnormal network in seizure propagation, here we identified inhibition of PF GABAergic neurons by SNr PV projections as an important mechanism allowing for precise regulation of TLE. Past findings have shown that high-frequency electrical stimulation of the PF interrupted ongoing hippocampal paroxysmal discharges[57], which may be due to the functional inhibition of the PF. Our findings provide a new viewpoint to understand the role of brain structures and neuronal circuits that are distant from the temporal lobe in TLE.

At present, AEDs that enhance GABA function, such as benzodiazepines, phenobarbital, and vigabatrin, are commonly used for the treatment of epilepsy[59]. Our results showed that activation of PF GABAergic neurons alleviated, whereas, paradoxically, activation of SNr GABAergic neurons aggravated the severity of seizure in TLE. Previous data have also shown that enhanced GABA inhibition promotes seizures[60−63], indicating that GABAergic activity can be excitatory, depending on the brain region and local circuitry (e.g., the PV synapses inhibited the inhibitory GABAergic cells in the nigral-parafascicular circuit, which resulted in a disinhibitory effect). The role of GABAergic neurons is region-specific and projection-specific in epilepsy. This may provide an explanation for the phenomenon that AEDs are ineffective in TLE patients and, in some cases, even promote seizures. Thus, from a clinical standpoint, targeted activation of GABAergic neurons in the posterior PF or other potential brain regions, such as seizure foci in the hippocampus, but not in the SNr, is a more attractive approach for epilepsy therapy.

In summary, our findings demonstrate that SNr PV neurons provide upstream inhibitory input to the posterior PF GABAergic neurons, and that this neural circuit functionally modulates seizure activities in TLE. More importantly, this disinhibitory neural circuit is distant from the epileptic foci, which provide a better understanding of the pathological network changes in and the precise spatiotemporal control of epilepsy.

## Methods
**Animals**. The *Vgat-Cre* mice (stock number: 016962), Vgat-ChR2-eYFP mice (transgenic *Vgat-ChR2-eYFP* mice; stock number: 014548), *PV-Cre* mice (stock number: 008069), NOS Cre-recombinase mice (*NOS-Cre* mice, stock number: 017526), and vesicular glutamate transporter (Vglut) Cre-recombinase mice (*Vglut2-Cre* mice, stock number: 016963) were genotyped according to the protocols provided by Jackson Laboratory. All mice were bred onto a C57BL/6J genetic background. Male mice 2–4 months of age were used in all experiments. All mice were group-housed (four to six per cage) prior to surgery under a 12 h light–dark

cycle (light on from 8:00 a.m. to 8:00 p.m.) with ad libitum access to food and water. The mice were individually housed after surgery for a better recovery and meanwhile reducing the failure rate of electrodes implantation, although singly housed animals showed stress and worse epilepsy phenotype[64]. All behavior experiments were performed each day between 9:00 a.m. and 7:00 p.m. The use and care of the mice were in accordance with the guidelines of the Animal Advisory Committee of Zhejiang University and the US National Institutes of Health Guidelines for the Care and Use of Laboratory Animals. All procedures were approved by the Animal Advisory Committee of Zhejiang University.

**Viral vectors**. AAV-Ef1α-DIO-GCaMP6m-eYFP (viral titers: $5.0 \times 10^{12}$ particles/mL), AAV-Ef1α-DIO-hChR2(H134R)-mCherry (viral titers: $1.7 \times 10^{13}$ particles/mL), AAV-Ef1α-DIO-hChR2(H134R)-eYFP (viral titers: $1.6 \times 10^{13}$ particles/mL), AAV-CAG-FLEX-ArchT-GFP (viral titers: $1.3 \times 10^{12}$ particles/mL), AAV-Ef1α-DIO-hM4Di-mCherry (viral titers: $1.0 \times 10^{13}$ particles/mL), and AAV-Ef1α-DIO-hM3Dq-mCherry (viral titers: $1.7 \times 10^{13}$ particles/mL) were purchased from Neuron Biotech Co., Ltd (Shanghai, China). The retrograde tracer Alexa Fluor-488-conjugated Cholera Toxin Subunit B (CTB-488, 2% in phosphate-buffered saline, PBS) was purchased from ThermoFisher Scientific (C22841, USA). AAV-Ef1α-DIO-GFP-TVA (viral titers: $3.8 \times 10^{12}$ particles/mL), AAV-Ef1α-DIO-RVG (viral titers: $3.4 \times 10^{12}$ particles/mL), and RV-EnvA-ΔG-dsRed (viral titers: $7.5 \times 10^{8}$ particles/mL) were purchased from BrainTVA Co., Ltd (Wuhan, China). All viral vectors were aliquoted and stored at −80 °C until use.

**Stereotactic surgery**. Mice were anesthetized with sodium pentobarbital (50 mg/kg, i.p., Sigma-Aldrich) and head-fixed in a stereotaxic apparatus (512600, Stoelting, USA). During the whole process of operation, the body temperature of anesthetized mice was kept constant at 37 °C using a heating pad. If the mice had a pain reflect in response a paw pinch, then additional 10% of initial dosage of sodium pentobarbital was given to guarantee a painless state. An incision was made to the mouse's head to expose the skull surface. After scraping the pericranium away, burr holes were stereotactically made on the skull.

For optogenetic or chemogenetic viral delivery, microinjection was performed using the following stereotaxic coordinates for the right SNr: antero-posterior (AP) −3.2 mm;, mediolateral (ML) −1.2 mm, and ventral (V) −4.5 mm; for the right PF: AP −2.4 mm, ML −0.8 mm, and V −3.5 mm; and for the right CA3: AP −2.9 mm, ML −3.2 mm, and V −3.5 mm, based on the mouse brain atlas[65]. Viruses were delivered via a gauge needle by an Ultra Micro Pump (160494 F10E, WPI) over a period of 10 min; the syringe was not removed until 15–20 min after the end of infusion, to allow diffusion of the viruses. All AAVs were injected at a total volume of ~0.4 μL and we waited ~4 weeks to allow for maximal viral expression. CTB-488 was injected at a total volume of ~0.2 μL and we waited ~2 weeks to allow the retrograde labeling of projection neurons.

For in vivo optogenetic manipulation in the epilepsy model of hippocampal CA3 kindling seizures, after 3 weeks of viral delivery, electrodes made of twisted Teflon-coated tungsten wires (795500; diameter, 0.125 mm; A.M. Systems) were implanted in the right ventral hippocampus (CA3: AP −2.9 mm, ML −3.2 mm, and V −3.5 mm) for kindling stimulation and EEG recording. The guide cannulas (62003, RWD Life Science Co., Ltd) were separately implanted in the following regions: right SNr: AP −3.2 mm, ML −1.2 mm, and V −4.5 mm; right PF: AP −2.4 mm, ML −0.8 mm, and V −3.5 mm; right RT: AP −0.8 mm, ML −1.6 mm, and V −3.2 mm; right RE: AP −0.6 mm, ML 0.0 mm, and V −4.0 mm; right VM: AP −1.2 mm, ML 0.7 mm, and V −4.0 mm. The guide cannulas were used to aid in inserting an optical fiber (diameter, 200 μm; Thorlabs) or the GABA_A receptor antagonist bicuculline injection during experiments (detailed below). Four screws were placed in the skull over the cortex to secure the dental cement, two of which were placed over the cerebellum to serve as the reference and ground electrodes. The start of kindling experiments was 1 week recovery after implantation surgery.

Following 3 weeks of viral delivery, KA (0.5 μg/μL, 0.6 μL, Abcam, ab120100) was injected via a guide cannula implanted into the right dorsal hippocampus (CA1: AP −2.0 mm, L −1.3 mm, and V −1.6 mm). Electrodes were implanted in the right ventral hippocampus (CA3: AP −2.9 mm, ML −3.2 mm, and V −3.5 mm) for EEG recording.

For monosynaptic tracing with pseudotyped rabies virus, Vgat-Cre or Vglut2-Cre mice were microinjected in the right PF (AP −2.4 mm, ML −0.8 mm, and V −3.5 mm) with 200 nl of a viral cocktail (1:1) containing AAV-Ef1α-DIO-GFP-TVA and AAV-Ef1α-DIO-RVG. AAV-Ef1α-DIO-GFP-TVA allows the initial infection of PF starter neurons and AAV-Ef1α-DIO-RVG codes for the rabies virus envelope glycoprotein and allows the trans-synaptic spread of virus. Three weeks later, 200 nl of the modified rabies virus RV-EnvA-ΔG-dsRed was microinjected in the same location. Complementation of the modified rabies virus with envelope glycoprotein in the TVA-expressing cells allows the generation of infectious particles, which then can trans-monosynaptically infect presynaptic neurons. One week after the last injection, mice were killed and brain sections were collected for confocal imaging.

Animals were kept on a heating pad throughout the entire surgical procedures and were brought back to their home cages after post-surgery recovery. The place of the electrode's implantation and the viral expression in all the mice were verified after behavioral tests. Only the mice with correct place of the electrodes

implantation and the viral expression were taken into analysis. These criteria were pre-established.

**Fiber photometry**. Following AAV-Ef1α-DIO-GCaMP6m-eYFP virus injection, an optical fiber (0.23 mm O.D., 0.37 mm numerical aperture (NA); Nanjing Thinkertech) was placed in a ceramic ferrule and inserted into the right SNr (AP −3.2 mm, ML −1.2 mm, and V −4.5 mm). Electrodes made of twisted Teflon-coated tungsten wires (795500; diameter, 0.125 mm; A.M. Systems) were implanted in the right ventral hippocampus (CA3: AP −2.9 mm, ML −3.2 mm, and V −3.5 mm) for kindling stimulation. The start of fiber photometry of calcium signals was 1 week after implantation surgery.

The fiber photometry system (Nanjing Thinkertech) used a 488 nm diode laser (OBIS 488LS; Coherent), reflected by a dichroic mirror (MD498, Thorlabs), and coupled into a 0.23 mm 0.37 NA optical fiber using a ×10 objective lens (Olympus) and fiber launch (Thorlabs). The laser intensity at the interface between the fiber tip and the animal ranged from 0.01 to 0.03 mW, to minimize bleaching. The GCaMP fluorescence was bandpass filtered (MF525-39, Thorlabs) and collected by a photomultiplier tube (R3896, Hamamatsu). An amplifier (C7319, Hamamatsu) was used to convert the photomultiplier tube current output to voltage signals, which was further filtered through a low-pass filter (40 Hz cut-off). The analog voltage signals were digitized at 500 Hz and recorded. Photometry data were exported as MATLAB Mat files for further analysis. We segmented the data based on individual trials of kindling stimulations and derived the values of fluorescence change ($\Delta F/F$) by calculating $(F - F_0)/F_0$. $\Delta F/F$ values were presented with heatmaps or average plots.

**Rapid hippocampal kindling model**. After 1 week of surgery recovery, the ADT of each mouse was determined (monophasic square-wave pulses, 20 Hz, 1 ms/pulse, 40 pulses) by a constant current stimulator (SEN-7203, SS-202J; Nihon Kohden, Japan) through a bipolar electrode in the right ventral hippocampal CA3 and the EEGs at the right ventral hippocampal CA3 were recorded with a digital amplifier (NuAmps, Neuroscan System, USA). The stimulation intensity began at 40 μA and was subsequently increased by 20 μA steps every 1 min until at least 5 s of ADD was elicited. This minimal intensity that produced ADD (≥5 s) was designated the ADT for that animal and was used for grouping thereafter. All mice received ten kindling stimulations daily (400 μA, monophasic square-wave pulses, 20 Hz, 1 ms/pulse, 40 pulses). The severity of behavioral seizures was scored according to Racine[66] using the following criteria: (1) facial movement; (2) head nodding; (3) unilateral forelimb clonus; (4) bilateral forelimb clonus and rearing; and (5) rearing and falling. Seizure stages 1–3 indicate FSs and stages 4–5 are GSs. When mice exhibited three consecutive stage-5 seizures, they were regarded as fully kindled. The seizure stage was scored by an investigator blinded to the group allocation.

**Photostimulation**. Blue (473 nm) or yellow (589 nm) laser light was delivered through a 200 μm diameter optic fiber (Thorlabs), which was connected to the laser (IKECOOL-Laser) by a Master-8 commutator. The optic fiber was cut flat and the laser power was adjusted to about 5 mW measured by a Power meter (Thorlabs). Immediately before placing a mouse in the chambers, the stylet was removed from the guide cannula and an optic fiber was inserted directly through the guide cannula. The optical fiber was secured to ensure no movement of the fiber occurred during the experiment. The blue-light (478 nm, 20 Hz, 10 ms/pulse and 600 pulses, 5 mW) or yellow-light (589 nm, continuous 30 s, 5 mW) stimulation was delivered immediately after kindling. To determine whether the pro-epileptic effect of SNr PV projections was mediated by GABA_A receptors, we injected the GABA_A receptor antagonist bicuculline (intra-PF injection, 2 μM, 0.5 μL) 10 min before the kindling stimulation.

**Intra-hippocampal KA model**. Acute seizure model: One week after the implant surgery, freely moving mice were intrahippocampally microinjected with KA (0.5 μg/μL, 0.6 μL) using an Ultra Micro Pump (160494 F10E, WPI) over a period of 5 min via the guide cannula. The EEG was recorded for 90 min after KA injection using a PowerLab system (AD Instruments, Australia) at a sampling rate of 1 kHz. Thirty minutes before KA injection, the groups received saline injection (i.p.) or CNO (1 mg/kg, i.p.). Seizure severity was classified according to a modified Racine scale using the following criteria: (1) facial movement; (2) head nodding; (3) unilateral forelimb clonus; (4) bilateral forelimb clonus and rearing; (5) rearing and falling; and (6) SE with continuous epileptiform activity or continual recurrent seizures (classes 3, 4, or 5). Seizure stages 1–3 indicate FSs and stages 4–5 are GSs. KA injection induced high-amplitude spike waves in the EEG and gradually elicited FS, GS, and SE within tens of minutes. The seizure stage was scored by an investigator blinded to the group allocation during a 1.5 h observation period. The EEG recorded in the ventral hippocampus was analyzed offline by a software package (Scan 4.5) in the Neuronscan System as in our previous study[67]. Briefly, raw EEG signals were sampled at 1000 Hz and recorded with bandpass filters spanning DC to 200 Hz. Spectral analysis was conducted using the fast Fourier transform or wavelet transform, which provided the total power between 0 and 100 Hz. The coastline index was determined as the sum of the absolute difference between successive points.

Chronic epileptic model SE is typically induced via intra-hippocampal injection of KA, which induces spontaneous recurrent seizures in the following several weeks.

First, we stereotaxically injected KA into the right dorsal hippocampus to induce SE in anesthetized mice. The mice were allowed to self-terminate and recov from the anesthesia state. Two months after KA injection, EEG was continuously recorded of freely moving mice using a PowerLab system (AD Instruments, Australia) at a sampling rate of 1 kHz, which was synchronized with video recording 10 h/day for 7 days as the baseline (Pre). In the present KA-induced chronic epilepsy model, the frequency of spontaneous seizures in KA mice was stable, so we recorded 10 h per day, to debase the workload. Most of discharges we recorded were characterized by bursts of high-voltage sharp waves (burst of spikes) and are not associated with any obvious behavioral alterations in the parallel video recordings, which usually were interpreted as paroxysmal discharges as previous studies[68,69]. Such paroxysmal events were defined as non-convulsive FSs when they lasted more than 10 s and had an average spike frequency $\geq 2\,\mathrm{Hz}$[70]. In addition to paroxysmal discharges, mice also exhibited infrequent tonic-clonic GSs that were associated with typical paroxysmal EEG activity with obvious post-seizure depression. Only the mice with detectable seizure-like events were further studied. Then, the mice were injected (i.p., 1 mg/kg) with CNO daily for 7 days (0.5 h before EEG recording each day) to test the effect of the chemogenetic modulation on chronic spontaneous seizures. EEG was recorded continuously during the CNO treatment and for an additional 7 days after CNO withdrawal (post treatment).

**Open-field test and rotarod test**. Locomotor behavior was tracked automatically via Open Field Video Tracking System. Half an hour after CNO injection (1 or 10 mg/kg, i.p.), mice were allowed to explore an open-field arena (45 cm × 45 cm × 45 cm, and the center zone line was 11.25 cm apart from the edge) for 5 min. The movement was recorded by a video camera mounted above the open-field arena, which was connected to a video recorder. Automated behavioral analysis was conducted using ANY-maze (ANY-maze Video Tracking System version 4.98, Stoelting Co., USA). The analysis produced a track record for each animal that contained a complete record of the animal's movement pattern in the open-field arena, including total travel distance, travel distance in the center, and number of entries into the center.

The rotarod test can be used to test the motor coordination ability of rodents. Half an hour after CNO injection (1 or 10 mg/kg, i.p.), the mice were placed on a rotating cylinder (6 cm × 3 cm, YLS-4C, Zhenghua Biologic, China). The cylinder accelerated from 5 r.p.m. to 40 r.p.m. in 2 min. Eight trials were tested per day for each mouse, at least 10 min interval between each trial and at least 1 h interval between every four trials. The latency to fall was recorded and the maximal observation time was 5 min. When each trial was finished, the apparatus was cleaned with 70% ethanol for the next use.

**In vivo single-unit recordings and analysis**. Extracellular single-unit and LFP recordings were conducted in urethane-anesthetized mice (1.4 g/kg i.p. with additional doses as needed). Body temperature was monitored and maintained by a heating pad. Craniotomies were made over the target coordinates, relative to the bregma for the SNr (AP −3.2 mm, ML −1.2 mm) or PF (AP −2.4 mm, ML −0.8 mm). Recordings were made with an 8-wire bundle of microelectrodes (12 μm, AM-systerm, USA) with an optical fiber lowered into the brain until they were just above the target structures (dorsoventral from the dura: SNr 4.2–4.8 mm; PF 3.0–3.5 mm) and advanced slowly using hydraulic microdrives (Narishige International USA, Inc., East Meadow, NY, USA) to isolate single cells. Neuronal activity was sampled using the Cerebus acquisition system (Version 6.04, Blackrock Microsystems; sample rate at 30 kHz, high-pass filtered at 250 Hz and sorted online) grounded to screws above the cerebellum and referenced online against a wire within the same brain area that exhibited a greater signal-to-noise ratio than 3:1. To improve recording signal-to-noise ratio, an online 50 Hz line noise-cancelation algorithm was also applied. Neuronal spike trains were recorded for at least 10 min with light on–off cycles. Recorded neuronal data were offline resorted by offline sorting software (version 3.3, Plexon, Inc., USA) to confirm the quality of the recorded cells. For cell-type classification in the in vivo single-unit recordings in the SNr and the PF, we used a criterion of 200 μs for full width half maximum, flat autocorrelogram (>10 ms), and high firing rate (>5 Hz) to define putative GABAergic neurons[31]. The width of spike waves was measured manually. Peri-event histograms and autocorrelation analysis were performed to analyze sorted neuronal data by neuroexplorer 4.0 software (Nex Technologies Int, Inc., USA). The criteria used to define an "excited" or "inhibited" neuronal response were as follows: firing rates were considered to be significantly different if they were >2 SDs, greater or less than baseline averages[71]. Briefly, the average firing rate during each 10 s-duration bin was calculated in 1 min baseline period. Then, the average firing rate of the seizure period or photostimulation was calculated and compared with that of the baseline period to test whether firing rate was >2 SDs, greater or less than the baseline average.

**In vitro electrophysiology**. To obtain acute PF slices, $Vgat\text{-}ChR2^{SNr}$ mice were decapitated and the brains were dissected rapidly and placed in ice-cold oxygenated (95% $O_2$ and 5% $CO_2$) solution containing the following (in mM): 110 choline chloride, 2.5 KCl, 1.3 $NaH_2PO_4$, 25 $NaHCO_3$, 0.5 $CaCl_2$, 7 $MgCl_2$, 20 D-glucose,

1.3 ascorbic acid, and 0.6 Na-pyruvate pH 7.3. Coronal brain slices (250 μm thickness) containing the PF were cut with a vibratome (VT1000S, Leica Instruments Ltd) and were maintained in an incubation chamber at 37 °C for further experiments. During experiments, an individual brain slice was transferred to a submersion-recording chamber and continuously perfused with recording solution containing the following (in mM): 125 NaCl, 2.5 KCl, 1.3 $NaH_2PO_4$, 25 $NaHCO_3$, 2 $CaCl_2$, 1.3 $MgCl_2$, 10 D-glucose, 1.3 ascorbic acid, and 0.6 Na-pyruvate, aerated with 95% $O_2$/5% $CO_2$ (~1–2 mL/min at room temperature). Whole-cell patch-clamp recordings were made by an EPC10 patch-clamp amplifier (HEKA Instruments), with a low-pass filter at 3 kHz and a sample rate of 1 kHz. The recording electrodes had tip resistances of ~4–8 MΩ. To record optical evoked GABAergic postsynaptic currents, recording pipettes were routinely filled with a solution containing the following (in mM): 140 CsCl, 5 NaCl, 10 HEPES, 0.2 EGTA, 2 Mg-ATP, 0.3 $Na_3GTP$, 5 QX314, 10 $Na_2$-phosphocreatine pH 7.2. Neurons were holding at −70 mV. The inhibitory postsynaptic currents were isolated by addition of 6-cyano-7-nitroquinoxaline-2,3-dione (20 μM) and (2 R)-amino-5-phosphono-valeric acid (20 μM) to the artificial cerebrospinal fluid to block excitatory transmission mediated by AMPA/Kainate and N-methyl-D-aspartate receptors, respectively. Experiments were performed in the presence of the sodium channel blocker TTX (1 μM) and the potassium channel blocker 4-AP (100 μM) to isolate the monosynaptic current while shining blue light on the surface of brain slices using an optic fiber connected to a blue laser power source (IKECOOL-Laser 473 nm). Light pulse (1 Hz, 10 ms, 10 pulses, 2 mW) was controlled with EPC10 patch-clamp amplifier (HEKA Instruments). To record the action potential firing properties of glutamatergic and GABAergic neurons in the PF, a series of hyperpolarizing and depolarizing current steps (−100 pA + 100 pA, 100 pA each a step, 500 ms) were applied to determine firing properties of PF neurons. A series of 300 ms depolarizing current pulses (increased in 5 pA steps) were applied to measure the threshold, amplitude, and the half-wave width of the action potentials. The pipette solution contained the following (in mM): 140 K-gluconate, 5 NaCl, 2 Mg-ATP, 0.2 EGTA, and 10 HEPES.

**Immunohistochemistry and imaging**. Mice were killed and transcardially perfused with saline, followed by 4% paraformaldehyde in 0.1 M phosphate buffer (PBS). The brain was then removed, fixed in 4% paraformaldehyde overnight, and then equilibrated in 30% sucrose at 4 °C. Coronal sections (30 μm) were cut on a freezing microtome (Leica CM3050, Germany). After rinsing with 0.5 % Triton X-100 (vol/vol) in 0.1 M PBS for 30 min and blocking with 10% (wt/vol) normal bovine serum for 2 h at room temperature, sections were incubated with the following primary antibodies diluted in PBS with 0.15% Triton X-100 overnight at 4 °C: anti-GAD65/67 (Millipore AB1511), anti-nNOS (ThermoFisher 61-7000), and anti-PV (Swant PV27). After rinsing with PBS, sections were incubated with Alexa-594 or Alexa-488-conjugated secondary fluorescent antibody (Jackson ImmunoResearch, 711-585-152 or 711-545-152) for 2 h at room temperature. After rinsing, the sections were mounted on slides using Vectashield Mounting Media (Vector Labs) and confocal images were captured (Olympus FV-1000). Fluorescent expression of optogenetic or chemogenetic virus was examined for all mice. If the optical fiber was located above (within 0.5 mm) the targeted region with the correct viral expression, the corresponding mouse was taken into statistical analysis.

**Statistics and reproducibility**. Data are presented as means ± SEM. Number of experimental replicates (n) is indicated in the figure legends and refers to the number of experimental subjects independently treated in each experimental condition. For all experiments, at least two times experiment was repeated independently with similar results. Statistical comparisons were performed using SPSS (version 17.0) or Prism (version 7.0) with appropriate methods as indicated in the figure legends. No statistical methods were used to pre-determine sample size or to randomize. For all analyses, the tests were two-tailed and a P-value < 0.05 was considered statistically significant. See detailed statistical description for all figures in Supplementary Table 1.

**Reporting summary**. Further information on research design is available in the Nature Research Reporting Summary linked to this article.

## Data availability
The datasets generated during and/or analysed during the current study are available from the corresponding author upon reasonable request. The source data are provided as a Source Data file with the paper.

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

## Acknowledgements

This project was supported by grants from the National Natural Science Foundation of China (81521062, 81630098, 81973298, and 81671282) and the Young Elite Scientist Sponsorship Program by CAST (2018QNRC001).

## Author contributions

Z.C. and Yi Wang conceived this project and wrote the manuscript. B.C., Yi Wang, and C.L.X. designed the experiments and analyzed the acquired data. H.M.C., L.Y.X., and W.K.L. contributed to identification of transgenic animals and cell counting. J.L.Z. and T.T.H. contributed to immunohistochemistry. B.C., L.Y.C., and J.L.Z contributed to surgery experiments. B.C., Yi Wang, W.K.L., and Ying Wang contributed to the behavioral experiments. Ying Wang and B.C. contributed to the monosynaptic tracing experiments. Yi Wang contributed to the in vivo single-unit recordings. P.D. contributed to the fiber photometry experiments. C.L.X. contributed to the in vitro electrophysiology. Y.G., S.H.Z., S.W., Y.D.Z., W.W.H., and S.M.D. contributed to discussion of the study.

## Competing interests

The authors declare no competing interests.
