## [Peer Review File · Nature Communications]

Reviewers' Comments:

Reviewer #1:

Remarks to the Author:

Overall:

This manuscript by Chen and colleagues is thorough and very interesting. The work builds on nearly half a century of preclinical research on the role of the substantia nigra pars reticulata in the control of seizures, and identifies a pathway from the SNpr to the PF thalamus that can be targeted for the control of seizures. This is a novel and interesting finding and will be of interest to both the epilepsy field and a broader neuroscience audience. While I have comments on some points in the background and discussion, and some methodological, statistical and control issues that should be addressed, overall, I find this to be a highly compelling manuscript.

1. The work of the late Dr. Karen Gale, who originally described the anti-seizure effects of nigral inhibition in the early 1980s, is not cited at all in this manuscript. Dr. Gale's name is synonymous with the substantia nigra within the field of epilepsy, and this oversight should be corrected.
2. The authors describe the SN-PF pathway as "previously unknown" which is surprising given this pathway was reported in 1979 by Beckstead, Nauta and colleagues (*Brain Res*, 1979). There is also no discussion of the role of the indirect SN-nRT-PF pathway originally described by Tsumori et al. (*Brain Res*, 2000). This is of particular interest as the authors found no effect of RT activation/inhibition in the current study.
3. There is no discussion of subregion within the SNpr that is involved. Work from the Redgrave and Moshe groups have both shown that not all regions of the SNpr are equivalent in their impact on seizure activity. Indeed, these groups have reported rostrocaudal differences, and have even suggested that the pars lateralis of the SN may in fact be the critical site. Along these lines there is also no discussion of prior work showing that activation of SNpr (pharmacologically) can increase seizure severity (see for example, Sperber et al., *Brain Res*, 1989).
4. Nail-Boucherie and colleagues (*Epilepsia*, 2005), reported that an excitatory projection from superior colliculus to PF is also involved in the control of seizures (the GAERS model of absence epilepsy). Given that the SC is a primary target of SNpr outflow, there appears to be convergence in the circuit. This may merit some discussion.
5. In 1987, Garant and Gale reported that ascending projections from the SNpr to the thalamus were not necessary for the anticonvulsant action of nigral inhibition against maximal electroshock seizures. While the seizures in the MES model are quite different from those in the kindling model or the intrahippocampal KA model, this may also merit some discussion.
6. Methodological Question – for the single unit analysis, it is not described how excitation, inhibition and no response were defined, nor is it clear the number of trials (seizures) recorded for each unit. Did the proportion of neurons displaying an excitatory response exceed that to be expected by chance?
7. Single Unit Analysis - If multiple seizures were recorded within subject, did the response change across repeated stimulation? Bonhaus et al., (*J Neurosci*, 1986) previously reported striking differences in the engagement of SNpr neurons in naïve and kindled animals. This should be clarified. Note that this same question applies to the photometry data.
8. Photometry – it has become more or less standard in the field to include imaging at a calcium-insensitive excitation wavelength for GCaMP (or imaging in GFP controls) to ensure that responses are not due to issues such as movement artifact. Indeed, even small deflections in the fiber optic cable can produce large amplitude responses during photometry. Can the authors provide evidence that the photometry signal they report is biologically relevant and not due to movement artifact?
9. The use of parametric statistics for the kindling progression is slightly concerning, as these data are inherently non-normal (they are categorical). Perhaps the authors might consider using an approach

like the Aligned Rank Transform for nonparametric ANOVA to address this (see Wobbrock et al for details).

10. Why do the authors break out Stage 2, Stage 4 and Fully Kindled, as compared to any of the other kindling stages in terms of number of stimulations to reach these thresholds?

11. For Figs 1 and 2, there is appreciable variability in the response to photostimulation within the treated groups, it would be particularly interesting to know how the optic placement and expression of the animals varied in this case. More generally, it is unclear if/how optic placement and opsin expression was verified for each subject.

12. Fig 3 – DREADD experiment in chronic TLE – the manuscript would be strengthened by including appropriate controls for Fig 3D-H (i.e., DREADD negative animals treated with CNO) as the authors acknowledge that CNO can be associated with off target effects.

13. Fig 4 - (1) what effect does BIC in PF have on seizures in the absence of light stimulation? (2) how do the authors account for potential antidromic activation of SNpr-PF terminals, and subsequent activation of other nigral targets?

14. Clarification for Figure S5 – for Panels C-I, how many mice were used for these experiments (I am assuming the individual data points represent cells, not animals?)

15. The finding that the neurons that receive input from SNr in the PF thalamus are local circuit is quite interesting – and somewhat surprising given that others (see: Bentivoglio et al., Exp Brain Res 1991; Arcelli et al., Brain Res Bull, 1997) have argued that interneurons are sparse or absent in the intralaminar thalamus of rodents, a profile that differs from that in the cat and primate. The author should discuss this point – are they certain these neurons didn't project elsewhere?

16. Fig S7 would benefit from DREADD-negative controls.

17. Minor comment: The axons of transfected neurons have been reported to remain photoexcitable even when severed from their parent somata 35, 36. This line on Pg 14 is out of context.

Reviewer #2:

Remarks to the Author:

The authors present a commendable enormous amount of work to decipher the SNr components that may control seizures in TLE.

They use a combination of techniques to propose that a GABAergic circuit involving the SNr inhibition of PF neurons is central to the control of seizure propagation.

The role of the SNr in TLE has been proposed a long time ago; authors using various protocols such as DBS, lesions, and injections of chemical. Note that the authors forgot to mention the seminal work of Karen Gale. The concept is not novel, but the detailed mechanism was not identified.

I have a major concern. The authors claim that "These findings provide a better understanding of the pathological network changes and the precise spatiotemporal control of epilepsy." However, 90% of the "TLE" experiments are done in control animals, which are kindled. This type of acute model can be used to propose testable hypotheses to chronic models. However, only one set of experiments is performed in a chronic model of TLE. As a result, from kindled animals, it is not possible to claim "a better understanding of the pathological network changes" because there are no major network alterations in the progressive kindling model. Likewise, it is not possible to conclude that we get a better understanding of "the precise spatiotemporal control of epilepsy". In TLE (both in patients and chronic models), there are major anatomical and functional alterations in SNr and thalamic nuclei. The circuits are reorganized, and we do not know whether results obtained in control animals can be generalized to chronic model.

There is nothing wrong with doing a study using the kindling model, but the impact of the conclusions is limited, by the very nature of the approach. My evaluation is that relying mostly on kindled normal animals to draw important conclusions on TLE is not enough for Nature Comm. As it stands, the study belongs more to a specialized journal. My evaluation would be entirely different if the authors were to

replicate their main findings in the chronic model. They have tried to verify one aspect (but there are major issues there, which are discussed below), so they have the possibility to do it.

Major

1. A major issue is the interpretation of most experimental results. The authors nicely show that a transition activation/inactivation of neurons/axons produce the expected effects in the recorded cells. But these experiments were performed under anesthesia with urethane that directly affects GABAergic transmission, thus acting as a confounding factor. In addition, in the kindling model, the optogenetic manipulation occurs for each stimulation. We do not know how cells will react to a build-up of activation (or CI loading). The authors need to show that the activation/inactivation do work recording units in awake animals (there could be ceiling effects, or reversal, or post inhibitory rebound, which is typical for some PV cells).
2. Characterization of seizures. Fig 1G and others. I do not understand. You indicated that you filtered the signal at 100Hz. Thus, you cannot see any frequency above 50 Hz at best, or 25 Hz. How did you perform the signal analysis and FFT? In addition, FFT is not appropriate for this type of dynamical signal, wavelet analysis is much better. In addition, you cannot calculate a power spectrum on a signal that is not in steady-state. Perhaps, you can look at seizure onset and offset. You may also consider using the coastline index to characterize seizures.
3. In the acute KA model. What were the criteria to distinguish FS from GS? The data is confusing. If the latencies to GS and SE are 30s and 35s, respectively, how can you have 6 GS on average? In addition, Fig 3C shows that in the CNO condition (bottom panel), the seizure seems very different. Please show the detailed LFP in the three conditions for GS and SE.
4. The latter leads to a major issue. Is it possible that CNO treatment and the heavy opto stimulations done during kindling change brain states globally? Targeting hub regions, such as SNr or thalamic nuclei may alter global brain dynamics. As a result, the effects may be due to a global alteration versus local one. This should be discussed at least.
5. Chronic model. This is the model that should be used throughout. First, I advise performing a morphological study to determine the amount of cell loss (in particular PV cells) in SNr and downstream regions. Second, I do not understand the definition used by the authors for FSs. Are they flurries of spikes? Please check the literature for what is accepted as a definition of a TLE seizure. The CNO protocol involves a daily injection in the animal. A control group is required with saline injection. Finally, why are the properties of FSs and GSs in HM3 and HM4 animals so different? They should be the same?
6. To convince us that the work is pertinent to TLE, you need to perform the expts described in Figs 4-6 in the chronic model.
7. Fig S5. I am not able to evaluate the in vitro part. Critical details are missing in the method section. How the different parameters were measured is not indicated. You need to provide how many cells were recorded from how many slices from how many animals.

Minor.

1. Fig S1C and elsewhere. With n=3 mice and few neurons, how did you assess inter-animal difference assessed? What were the criteria to decide that cells were excited/inhibited? What was the repartition per mouse?
2. Chronic model. Note that ref 28 is not appropriate as the intra-hippocampal injection came much later. Note also that CNO is not inert. Some papers showed that it does have a physiological impact.
3. Fig 3H1, the two panels are identical.
4. Fig S4G, why measuring the duration of the motor phase in this experiment and not for the other ones?

Reviewer #3:

Remarks to the Author:

The manuscript reports a new projection from SNr PV neurons to GABAergic cells in Pf thalamus and suggests that this projection is involved in TLE seizure control. The existence of GABAergic cells in Pf and the relevance of these in controlling TLE seizure severity is novel, interesting, and surprising given how small the proportion of GABA cells is in Pf thalamus (only 5%). Also, it is interesting that the inhibitory output from SNr PV neurons to the thalamic PF, but not to the RT, RE or VM, is involved in the TLE seizure control. However, I have some concerns regarding data interpretation and clarity that need to be addressed to support the main conclusions.

Major Concerns:

1. The authors claim that SNr PV cells control TLE seizures via a new "disinhibitory" projection from SNrPV cells to GABAergic cells in Pf, rather than via the direct projection of SNr to glutamatergic cells. "Quantity analysis showed the number of SNr PV neurons targeting PF glutamatergic neurons was much lower than that targeting PF GABAergic neurons (Fig. S6)." However, SNrPV stimulation inhibits a non-negligible population of Pf GLU cells as well (see Fig 5B). In fact, SNr stimulation inhibits the same number of GLU and GABA cells in Pf according to Fig 5B. How do the authors explain the inconsistencies between conclusions from Fig S6 and conclusions from Fig 5B (see also Fig S9)?

In other words, in light of the results shown in Fig. 5B and Figs. S5 and S6, the overall working model is unclear. The proportion of GABA vs GLU cells in Pf is only 5% in the Pf nucleus (see Fig S5A); Moreover less than 20% of these GABA Pf cells receive a functional input from SNr (Fig S5, patching results: only 3 GABA cells out of 17 receive SNr inputs). Could this low number be due to sub-optimal recording conditions? What is the rationale for using 4-AP and TTX when recording evoked IPSCs from SNr to Pf thalamus?

Can the authors speculate on how the SNr projection onto just 1% of Pf nucleus is more important for seizure control than the direct targeting of GLU Pf cells by SNr (which would represent a higher number of cells according to Fig 5B, although inconsistent with Fig S6)?

In Fig S7, the authors nicely show that the chemogenetic activation of PF GABAergic neurons blocks pro-epileptic effects of optogenetic activation of SNr-PF GABAergic projections. However, I am not convinced that this result can be interpreted as if the pro-epileptic effect of SNr activation was due 100% to GABA Pf cells and did not involve SNr projections to GLU Pf cells (because see Fig 5B, there are GLU Pf cells that seem to be inhibited by SNr). Thus, the fact that the pro-epileptic effect of SNr activation was reduced in the Fig S7 experiment could be due to a combined effect of a direct inhibition of Pf GLU cells by SNr AND to inhibition of Pf GLU cells by chemogenetic activation of GABA Pf cells.

The authors do claim that GLU Pf neurons are important for SNr-mediated modulation of TLE seizures (Fig S9 and last part of the discussion), but it is unclear how the authors reconcile the direct SNr effects on GABA vs GLU Pf cells.

Fig. 5: The authors conclude that the anti-epileptic effect of optogenetic activation of PF GABAergic neurons was reversed by intra-PF application of the GABA receptor antagonist bicuculline. However, bicuculline could affect both glutamatergic and GABAergic cells in PF, both of which receive GABAergic inputs from SNr. This caveat should be addressed in the text because it affects data interpretation.

In sum, it would significantly strengthen the manuscript if the authors could propose a working model / diagram based on their results. The authors should edit the text to make it very clear what would one would expect from activation or inhibition of SNrPV cells on seizures with and without existence of GABAergic Pf cells (even if these are present in relatively small numbers). How does SNr PV projection compare between GABAergic and glutamatergic Pf cell population? This is key for speculating about the relevance of the "new" circuit. In other words, the presence of GABAergic cells in Pf is novel in this manuscript, so the authors should clarify with a concluding/speculative diagram how the presence of these cells and the fact they receive projections from SNr would actually affects the effect of SNr on TLE seizures.

2. Given that the authors claim that the output of Pf cells is important for controlling seizures (which is the major claim of the study), I recommend showing the firing of these cells during seizures in the main figures rather than in the Supplemental figure 9.

3. Given that the existence of GABAergic cells in Pf thalamus is novel (because the Pf neurons are thought to be only glutamatergic), it is important to characterize the properties of these cells. Fig S5 panel C shows an unnaturally large current injection of -500 pA and the traces are not representative of the average results (e.g firing). I recommend finding more representative voltage traces in response to positive and negative current pulses.

4) The manuscript contains numerous errors and lacks clarity.

a) Page 3, Introduction: "Basal ganglia circuits are closely involved in the modulation, propagation, and cessation of several types of epileptic seizures, including TLE 7, 8."

Please note that TLE is not an epileptic seizure. TLE is a form of epilepsy. This sentence needs to be rewritten. Suggestion: "Basal ganglia circuits are closely involved in the modulation, propagation, and cessation of seizures in different types of epilepsies, including TLE 7, 8."

-Also, the references 7 and 8 do not encompass the broad claim about basal ganglia controlling several types of seizures. Basal ganglia have also been shown to control absence type epileptic seizures (see work from the Charpier group on how SNr projections control thalamocortical oscillations in epilepsy, e.g., PMID:17251435). The authors cite the references correctly in the discussion but not in the introduction.

b) Page 3, Introduction, The reference cited in this sentence is incorrect:

"The substantia nigra pars reticulata (SNr), a region that mainly contains GABAergic neurons, controls the activity of both corticothalamic and limbic networks through its primary GABAergic output, acting like the "choke point" of basal ganglia 9".

Reference 9 corresponds to: "Paxinos G. The rat nervous system. (ed^(eds). Fourth edition. edn. Elsevier/Academic Press, (2015)" and is not the right reference for the notion of choke points. For the notion of choke points, the authors should cite relevant manuscripts (e.g the reference 2 from this manuscript which is the one that focuses on the notion of choke points).

c) Intro page 4: What neurons do the authors refer to by "these neurons" in the following section:

"We identified a previously unknown nigra-parafascicular disinhibitory circuit for regulation of seizure in TLE. We found that selective activation of these neurons amplifies seizure activities, whereas

inhibition of SNr PV neurons alleviated the severity of epileptic seizures”.

Do they mean the following: “We found that selective activation of SNr PV neurons amplifies seizure activities, whereas their inhibition alleviated the severity of epileptic seizures” ?

d) In the same paragraph, the authors write:

“.[...] the SNr PV neurons sent long-range axons to the GABAergic neurons in posterior parafascicular nucleus (PF), and optogenetic manipulation of this disinhibitory nigra-parafascicular circuit bidirectionally modulated seizures in TLE.”

This statement could be interpreted as if SNr PV neurons do not project to glutamatergic neurons in Pf that are known to be the major cell type in Pf. However, SNr PV projections to Pf do not seem to be cell-type specific (according to the text) because SNr PV cells seem to be projecting to Pf glutamatergic cells as well (e.g Fig 5B), suggesting that the direct SNr->GLU Pf pathway needs to be considered when interpreting the results from Fig S7.

e) The authors use interchangeably different expressions to describe the model: “KA-induced chronic epileptic model” and “hippocampal kindling model” (e.g. in the titles of Figures 1, 2 vs Figure 3. To avoid confusion, I suggest to use the same wording.

f) The legend of the Fig. S1G which should be key for showing that SNr stimulation actually controls TLE seizures needs to be clarified. The authors need to explain in this legend what they mean by “Blue-Imm.” vs “Blue Pre” and “Blue Delay”. The authors should explain in the legend what the “different photostimulation timing” means in the following sentence “The effect of optogenetic activation of SNr GABAergic neurons with different photostimulation timing on the development of seizure stage (G-1).”

I recommend adding a diagram showing the timing of photostimulation.

g) Fig S3:

-VTN (ventral thalamic nucleus) is mentioned in the legend, but I can't find VTN in the figure panels. Indicate in the panels, or remove from the legend.

Also, Po is indicated in panels, but the abbreviation is not defined in the legend.

-Panel C: the authors claim in the legend and in the main text that SNr projects to RT but I can't see any projections from SNr to RT in the panels. If the authors want to report this projection, I suggest adding a zoom on the RT to show presence of SNr fibers/synaptic boutons.

h) Fig S1: legend: In the sentence: “...ipsilateral SNr from 3 wildtype mice”: the authors should clarify ipsilateral to what?

i) There are many grammatical and typing mistakes throughout the text. It would be too long to list all of them.

E.g., “These results indicate that activation of SNr GABAergic neurons is sufficient to amplifies seizure activities in hippocampal kindling model”.

Reviewer #4:

Remarks to the Author:

In this study Chen et al analyzed the involvement of the substantia nigra pars reticulata (SNr) in temporal lobe epilepsy (TLE) by employing the kindling and the kainic acid animal models. They report

that optogenetic or chemogenetic activation of SNr parvalbumin+ (PV) interneurons “amplifies” seizure activities while their inhibition alleviates them. They also state that seizure severity was bidirectionally regulated by optogenetic manipulation of SNr PV fibers projecting to the parafascicular nucleus (PF), and found with electrophysiology combined with rabies virus-assisted circuit-mapping that SNr PV neurons directly project to and functionally inhibit posterior PF GABAergic neurons. They conclude that their findings reveal “a previously unknown long-range SNr-PF disinhibitory circuit that modulate seizures in TLE, and that inactivation of this circuit can alleviate epileptic seizures.

This is a complex study that includes a variety of experimental approaches providing a huge amount of data. The topic is relatively unexplored and the findings original and consistent with the conclusion that the SNr-PF circuit modulates seizures. However, at this stage, this study is characterized by a rather superficial analysis of the results, which may be due to the amount of different experiments performed. Perhaps the authors should focus on some specific experiments and provide full, in depth information on them.

Some specific critiques are as follows.

1. In the main body of the paper it should be specified whether the “CA3 kindling” was acute or the classic chronic kindling. This point is of paramount importance to follow what written in the Results (p. 5), namely “Immediately after the initiation of CA3 kindling, about 55% of GABAergic neurons in the ipsilateral SNr increased their firing rate (peak firing rate was 17.06 ± 1.01 Hz; out of 38 recorded neurons from 3 wildtype mice, 21 neurons were excited, 4 inhibited, 13 no response; no inter-animal statistical difference was detected, Chi-square test, $p=0.8927$, Fig. S1B and F1C).” In fact, it is obscure to me whether these changes occurred during the kindling stimulation or after several days of kindling.
2. The same critique applies to what stated in p. 6: In the hippocampal kindling model, we applied 30 s photostimulation in the ipsilateral SNr immediately after hippocampal kindling...” More specific information on the timing of optogenetic stimulation in relation with the kindling procedure must be given.
3. I have some concerns with the “negative” data obtained by stimulating SNr NOS positive neurons (see p. 8), which according to these authors is the second largest population of GABAergic cells in the SNr. Did they check for expression? How come, if effectively activated by optogenetic procedures these interneurons did not influence the “kindling” process?? In fact, were they optogenetically excited??
4. As remarked above, with regard to the kindling experiments, also the chemogenetic procedures applied to kainic acid treated mice (pp. 10 and 11) lack experimental details.
5. There is actually some debate in the literature regarding the role of CNO in chemogenetics. Some studies have shown that CNO does not cross the blood-brain barrier but that is instead converted to clozapine, which acts as a D1R agonist. Since it is known that the activation of D1R modulates the release of GABA from SN terminals, it would be interesting to see whether a similar effect on chronic seizures in the KA model is also observed in the PVCre + CNO group (Figure 3).
6. Page 6: It is stated that “Representative peri-event histograms confirmed that blue-light stimulation (473 nm, 20 Hz, 10 ms, 5 mW, 10 s on-off cycle) excited 15 out of 19 SNr GABAergic neurons from 3 Vgat-ChR2-eYFP mice, suggesting that SNr GABAergic neurons can be functionally activated by blue-light stimulation. Yellow-light stimulation (589 nm, continuous light, 5 mW, 10s on-off cycle), serving as the control stimulation, had no effect on the same neurons (Fig. S1F). It is unclear why a 20 Hz

blue-light stimulation was used to activate SNr GABAergic neurons whereas continuous yellow light stimulation was used for control experiments. A similar comment would apply to the results shown in figure 1 G-1 if the same procedure was used.

7. Page 9: "The number of stimulations needed to reach stage 2 was not affected here (only has a tendency, $U=19$, $P=0.1740$)". I would suggest removing "only has a tendency" since the p value of 0.17 is not close to significance.

8. Page 29: It is unclear whether the status epilepticus induced with local administration of KA was pharmacologically stopped or allowed to self-terminate. Status epilepticus of different durations could lead to differences in seizure occurrence and neuropathology during the chronic period. Please specify.

9. Page 30: It is stated that "Spontaneous seizure events were defined as regular spike clusters with a duration of ≥ 10 s, spike frequency of ≥ 2 Hz and amplitude at least three times of the baseline EEG, accompanying behavioral tonic-clonic GSs". Does this mean that non-convulsive seizures that were only visible on the EEG were not considered?

The paper should also be carefully edited. A few examples

- p.2, in the Introduction "...remote brain regions, extending from cortical to subcortical limbic structures and other remote structures."
- p.3 in the Introduction: "...targeting at the SNr..."
- p. 5 in the Results: "in the previous study."
- p.19: "Furthermore, optogenetic activation of PF glutamatergic neurons was sufficient to accelerate seizure development in hippocampal kindling model".

Reviewers' comments:

Reviewer #1 (Remarks to the Author):

Overall:

This manuscript by Chen and colleagues is thorough and very interesting. The work builds on nearly half a century of preclinical research on the role of the substantia nigra pars reticulata in the control of seizures, and identifies a pathway from the SNpr to the PF thalamus that can be targeted for the control of seizures. This is a novel and interesting finding and will be of interest to both the epilepsy field and a broader neuroscience audience. While I have comments on some points in the background and discussion, and some methodological, statistical and control issues that should be addressed, overall, I find this to be a highly compelling manuscript.

Reply: Thank you very much for reviewing our work and propounding valuable comments, which are very helpful for us to improve the quality of our paper significantly. Below are our point-by-point responses.

1. The work of the late Dr. Karen Gale, who originally described the anti-seizure effects of nigral inhibition in the early 1980s, is not cited at all in this manuscript. Dr. Gale's name is synonymous with the substantia nigra within the field of epilepsy, and this oversight should be corrected.

Reply: Thank you very much for this comment. We have included the work of Dr. Karen Gale in our revised manuscript.

Page 3-4 in revised manuscript:

--"Experimental studies have also reported structural and functional changes of SNr neurons among different types of epileptic models^{14, 15, 16, 17, 18}. In addition to those findings, lesion¹⁹, pharmacological interference^{20, 21, 22, 23} or deep brain stimulation^{24, 25} targeting the SNr can regulate the process of epileptic seizures, suggesting that the SNr plays a key role in seizure control."--

Page 22 in revised manuscript:

--"Interestingly, SNr GABAergic neurons control absence epilepsy through the downstream VM thalamocortical circuit⁹, while nigrothalamic/nigrostriatal projections do not appear to contribute to the electroshock convulsions⁵¹, suggesting that basal ganglia play a role in various seizure disorders through distinct downstream circuits."--

New added references:

18. Gale K, Iadarola MJ. GABAergic denervation of rat substantia nigra: functional and pharmacological properties. *Brain research* 183, 217-223 (1980).
19. Garant DS, Gale K. Lesions of substantia nigra protect against experimentally induced seizures. *Brain research* 273, 156-161 (1983).
23. Iadarola MJ, Gale K. Substantia nigra: site of anticonvulsant activity mediated by gamma-aminobutyric acid. *Science* 218, 1237-1240 (1982).
24. Sperber EF, Wurlpel JN, Zhao DY, Moshe SL. Evidence for the involvement of nigral GABA_A receptors in seizures of adult rats. *Brain research* 480, 378-382 (1989).
51. Garant DS, Gale K. Substantia nigra-mediated anticonvulsant actions: role of nigral output pathways. *Experimental neurology* 97, 143-159 (1987).

2. The authors describe the SN-PF pathway as "previously unknown" which is surprising given this pathway was reported in 1979 by Beckstead, Nauta and colleagues (Brain Res, 1979). There is also no discussion of the role of the indirect SN-nRT-PF pathway originally described by Tsumori et al. (Brain

Res, 2000). This is of particular interest as the authors found no effect of RT activation/inhibition in the current study.

Reply: Thank you very much for this comment. According, we have deleted the “previously unknown” words and added some discussions in the revised manuscript as following:

Page 21 in revised manuscript:

--“Surprisingly, although there is an indirect pathway from the SNr to the PF via the RT⁴⁵, we found that SNr-RT PV projections had no effect on seizure propagation of TLE. These results indicated that the SNr PV projections were circuit-specific regulators of TLE.”—

3. There is no discussion of subregion within the SNpr that is involved. Work from the Redgrave and Moshe groups have both shown that not all regions of the SNpr are equivalent in their impact on seizure activity. Indeed, these groups have reported rostrocaudal differences, and have even suggested that the pars lateralis of the SN may in fact be the critical site. Along these lines there is also no discussion of prior work showing that activation of SNpr (pharmacologically) can increase seizure severity (see for example, Sperber et al., Brain Res, 1989).

Reply: Thank you very much for this insightful comment. Indeed, not all subregions of the SNr may be equivalent in their impact on seizure activity, as we can see appreciable variability in the response to optogenetic modulation in **Fig.1 and 2**. Here, we did not further analysis how the fiber location within the SNr would have a correlation with the treated effects, because apart from fiber location, there are other factors (the amount of viral expression, area size of illumination, et.al.) that would affect seizure-modulating effects. We do agree with the reviewer that precise location of SNr subregion is an important subject for future precise seizure control, as previous studies have indicated that there are rostro-caudal differences of SNr in seizure control. Accordingly, we have cited corresponding papers and added some discussions in the revised manuscript as following:

Page 3-4 in revised manuscript:

--“In addition to those findings, lesion¹⁹, pharmacological interference^{20, 21, 22, 23, 24} or deep brain stimulation^{25, 26} targeting the SNr can regulate the process of epileptic seizures, suggesting that the SNr plays a key role in seizure control.”--

Page 21 in revised manuscript:

--“It should be noted that not all regions of the SNr may be equivalent in their impact on seizure activity, as we can see appreciable variability in the response to optogenetic modulation in Fig.1 and 2. Indeed, previous studies have reported that there is rostro-caudal differences of SNr in seizure control^{46, 47, 48}.”--

New added references:

24. Sperber EF, Wurlpel JN, Zhao DY, Moshe SL. Evidence for the involvement of nigral GABAA receptors in seizures of adult rats. Brain research 480, 378-382 (1989).
46. Moshe SL, Garant DS, Sperber EF, Veliskova J, Kubova H, Brown LL. Ontogeny and topography of seizure regulation by the substantia nigra. Brain & development 17 Suppl, 61-72 (1995).
47. Veliskova J, Loscher W, Moshe SL. Regional and age specific effects of zolpidem microinfusions in the substantia nigra on seizures. Epilepsy research 30, 107-114 (1998).
48. Shehab S, Simkins M, Dean P, Redgrave P. Regional distribution of the anticonvulsant and behavioural effects of muscimol injected into the substantia nigra of rats. The European journal of neuroscience 8, 749-757 (1996).

4. Nail-Boucherie and colleagues (Epilepsia, 2005), reported that an excitatory projection from superior colliculus to PF is also involved in the control of seizures (the GAERS model of absence

epilepsy). Given that the SC is a primary target of SNpr outflow, there appears to be convergence in the circuit. This may merit some discussion.

Reply: Thank you very much for this comment. Accordingly, we have added some discussions in the revised manuscript as following:

Page 21-22 in revised manuscript:

--“As a common issue in optogenetic and chemogenetic modulation, targeting hub regions, such as the SNr or PF here, may alter global brain dynamics. Therefore, apart from SNr-PF circuit, other potential global alterations may also account for the factors for the regulator seizure in TLE. For example, the superior colliculus, a primary target of SNr output, sends glutamatergic projections to the PF, which was previously reported to be involved in the control of absence seizures⁵⁰. It is possible that the superior colliculus may also have a potential role in seizure of TLE.”--

New added reference:

50. Nail-Boucherie K, Le-Pham BT, Gobaille S, Maitre M, Aunis D, Depaulis A. Evidence for a role of the parafascicular nucleus of the thalamus in the control of epileptic seizures by the superior colliculus. *Epilepsia* 46, 141-145 (2005).

5. In 1987, Garant and Gale reported that ascending projections from the SNpr to the thalamus were not necessary for the anticonvulsant action of nigral inhibition against maximal electroshock seizures. While the seizures in the MES model are quite different from those in the kindling model or the intrahippocampal KA model, this may also merit some discussion.

Reply: Thank you very much for this comment. Accordingly, we have added some discussions in the revised manuscript as following:

Page 22 in revised manuscript:

--“Interestingly, SNr GABAergic neurons control absence epilepsy through the downstream VM thalamocortical circuit⁹, while nigrothalamic/nigrostriatal projections do not appear to contribute to the electroshock convulsions⁵¹, suggesting that basal ganglia play a role in various seizure disorders through distinct downstream circuits.”--

New added reference:

51. Garant DS, Gale K. Substantia nigra-mediated anticonvulsant actions: role of nigral output pathways. *Experimental neurology* 97, 143-159 (1987).

6. Methodological Question – for the single unit analysis, it is not described how excitation, inhibition and no response were defined, nor is it clear the number of trials (seizures) recorded for each unit. Did the proportion of neurons displaying an excitatory response exceed that to be expected by chance?

Reply: Thank you very much for this comment. As previous study¹, the criteria used to define an “excited” or “inhibited” neuronal response was as follows: firing rates were considered to be significantly different if they were >2 SDs of baseline averages. Briefly, the average firing rate during each 10-s-duration bin was calculated in 1-min baseline period. Then, the average firing rate of the seizure period or photostimulation was calculated and compared with that of the baseline period to test whether firing rate was >2 SDs greater or less than the baseline average. In single-unit recording experiments with optogenetic modulation, we found blue-light stimulation excited putative PV neurons (10 out of 22 neurons from 3 *PV-ChR2^{SNr}* mice), while yellow-light stimulation, as a control light, had no effect on the same neurons (**Fig. 1E**), suggesting that the proportion of neurons displaying an excitatory response may not be expected by chance. To make it more clear, we have added detailed information in the revised method as following:

Page 33-34 in revised manuscript:

--“The criteria used to define an “excited” or “inhibited” neuronal response were as follows: firing rates were considered to be significantly different if they were >2 SDs greater or less than baseline averages⁶⁶. Briefly, the average firing rate during each 10-s-duration bin was calculated in 1-min baseline period. Then, the average firing rate of the seizure period or photostimulation was calculated and compared with that of the baseline period to test whether firing rate was >2 SDs greater or less than the baseline average.”--

Reference:

1. Toyoda I, Fujita S, Thamattoor AK, Buckmaster PS. Unit Activity of Hippocampal Interneurons before Spontaneous Seizures in an Animal Model of Temporal Lobe Epilepsy. *Journal of Neuroscience* 35, 6600-6618 (2015).

7. Single Unit Analysis - If multiple seizures were recorded within subject, did the response change across repeated stimulation? Bonhaus et al., (J Neurosci, 1986) previously reported striking differences in the engagement of SNpr neurons in naïve and kindled animals. This should be clarified. Note that this same question applies to the photometry data.

Reply: Thank you very much for this important comment. For single-unit analysis, we did single-unit recording in anesthesia mice, so we did not measure the response change across repeated stimulations-induced seizure development. While, for fiber photometry experiment, we recorded the Ca²⁺ response change across repeated stimulation, we found that calcium signal of SNr PV neurons gradually increases during kindling acquisition (**Fig. S2a**). Combined with previous report from Bonhaus et al., this data suggested that basal firing rate of SNr GABAergic neurons and their response to seizure significantly increased after acquisition of generalized seizure. Thus, we have added this data into **Fig. S2a** as following:

Revised Fig. S2a

8. Photometry – it has become more or less standard in the field to include imaging at a calcium-insensitive excitation wavelength for GCaMP (or imaging in GFP controls) to ensure that responses are not due to issues such as movement artifact. Indeed, even small deflections in the fiber optic cable can produce large amplitude responses during photometry. Can the authors provide evidence that the photometry signal they report is biologically relevant and not due to movement artifact?

Reply: Thank you very much for this important suggestion. To test whether the increased calcium signal was due to movement artifact, we injected AAV-eYFP control virus into the SNr of *PV-cre* mice (*PV-eYFP^{SNr}* mice), and recording calcium response during kindling-induced seizure. We found that there is no change of calcium signal in *PV-eYFP^{SNr}* mice, suggesting that calcium signal of SNr PV neurons was caused seizure activities, but not movement artifact. Therefore, we have included this data in the **Fig. S2b**.

9. The use of parametric statistics for the kindling progression is slightly concerning, as these data are inherently non-normal (they are categorical). Perhaps the authors might consider using an approach like the Aligned Rank Transform for nonparametric ANOVA to address this (see Wobbrock et al for details).

Reply: Thank you very much for this comment. Accordingly, we have used the Aligned Rank Transform for nonparametric ANOVA for the statistical test for the development of seizure stage during kindling progression in the revised manuscript as previous study¹.

Reference:

1. Wobbrock, J.O., L. Findlater, D. Gergle, and J.J. Higgins, The Aligned Rank Transform for Nonparametric Factorial Analyses Using Only ANOVA Procedures. 29th Annual Chi Conference on Human Factors in Computing Systems, 2011: p. 143-146.

10. Why do the authors break out Stage 2, Stage 4 and Fully Kindled, as compared to any of the other kindling stages in terms of number of stimulations to reach these thresholds?

Reply: Thank you very much for this comment. The severity of kindling-induced behavioral seizures was scored according to Racine scale. Usually, seizure stages 1-3 indicate focal seizures (FSs), stages 4-5 indicate generalized seizures (GSs) and three consecutive stage 5 seizures indicate fully kindled state. Thus, we used Stage 2, Stage 4 and Fully Kindled to represent typical FS, GS and fully kindled states, respectively.

11. For Figs 1 and 2, there is appreciable variability appreciable variability within the treated groups, it would be particularly interesting to know how the optic placement and expression of the animals varied in this case. More generally, it is unclear if/how optic placement and opsin expression was verified for each subject.

Reply: Thank you very much for these insightful comments.

Fluorescent expression of optogenetic or chemogenetic virus was examined for all mice with Olympus FV-1000 imaging system. If the optical fiber was located above (within 0.5 mm) the targeted region with correct viral expression, the corresponding mouse was taken into statistical analysis, since previous study indicated that surface positioning of the optical fiber allows effective stimulation with ~500 µm depth of cortical cortex. (Cardin, J.A., M. Carlen, K. Meletis, et al., Targeted optogenetic stimulation and recording of neurons in vivo using cell-type-specific expression of Channelrhodopsin-2. Nature Protocols, 2010. 5(2): p. 247-254.)

We have also noticed there is appreciable variability in optogenetic modulation experiment in Fig. 1 and 2 variability. Please see all the raw data in the “**source data**” file. We did not further analysis how the fiber location within the SNr would have a correlation with the treated effects, since apart from fiber location, there are other factors (the amount of viral expression, area size of illumination, et.al.) that would affect seizure-modulating effects. We do agree with the reviewer that precise location of SNr subregion is an important subject for future precise seizure control, as previous studies have indicated that there are rostro-caudal differences of SNr in seizure control. Accordingly, we have added corresponding discussion in the revised manuscript and more detailed description for method as following:

Page 36 in revised manuscript:

--“Fluorescent expression of optogenetic or chemogenetic virus was examined for all mice. If the optical fiber was located above (within 0.5 mm) the targeted region with correct viral expression, the corresponding mouse was taken into statistical analysis”--

Page 21 in revised manuscript:

--“It should be noted that not all regions of the SNr may be equivalent in their impact on seizure activity, as we can see appreciable variability in the response to optogenetic modulation in Fig.1 and 2. Indeed, previous studies have reported that there is rostro-caudal differences of SNr in seizure control^{46, 47, 48}.”--

New added references:

46. Moshe SL, Garant DS, Sperber EF, Veliskova J, Kubova H, Brown LL. Ontogeny and topography of seizure regulation by the substantia nigra. *Brain & development* 17 Suppl, 61-72 (1995).
47. Veliskova J, Loscher W, Moshe SL. Regional and age specific effects of zolpidem microinfusions in the substantia nigra on seizures. *Epilepsy research* 30, 107-114 (1998).
48. Shehab S, Simkins M, Dean P, Redgrave P. Regional distribution of the anticonvulsant and behavioural effects of muscimol injected into the substantia nigra of rats. *The European journal of neuroscience* 8, 749-757 (1996).

12. Fig 3 – DREADD experiment in chronic TLE – the manuscript would be strengthened by including appropriate controls for Fig 3D-H (i.e., DREADD negative animals treated with CNO) as the authors acknowledge that CNO can be associated with off target effects.

Reply: Thank you very much for this constructive suggestion. To test whether the CNO itself would have any seizure-modifying effect in chronic epileptic mice, we injected AAV-ef1a-DIO-mCherry control virus into the SNr of *PV-cre* mice (*PV-mCherry^{SNr}* mice). We found that CNO treatment did not change the frequency and seizure duration of both FS and GS in *PV-mCherry^{SNr}* mice, suggesting anti-seizure effect of CNO in *PV-hM4Di^{SNr}* mice may not be associated with off target effects of CNO. Thus, we have added this result in the revised **Fig. 3G** as following:

Revised Fig. 3G

Page 11 in revised manuscript:

--“CNO itself did not change the frequency and seizure duration of both FS and GS in *PV-mCherrySNr* mice (Fig. 3G), suggesting anti-seizure effect of CNO in *PV-hM4Di^{SNr}* mice may not be associated with off target effects of CNO.”--

13. Fig 4 - (1) what effect does BIC in PF have on seizures in the absence of light stimulation? (2) how do the authors account for potential antidromic activation of SNpr-PF terminals, and subsequent activation of other nigral targets?

Reply: Thank you very much for these insightful comments.

(1) We found that bicuculline injected into the PF alone had no obvious effect on kindling acquisition (Please see **Figure for reviewer 1** below). It can be caused by the non-specific effect of pharmacological modulation on PF neuron. As it was also referred by the third reviewer, bicuculline could affect both glutamatergic and GABAergic cells in PF, both of which receive GABAergic inputs from the SNr or even other regions. This caveat may complicate data interpretation. To avoid confusing, we did not include this data in the manuscript.

(2) To test whether there is a potential antidromic activation of SNr soma when activating SNpr-PF terminals, we performed c-fos immunohistochemical experiment in the SNr. We found that there is no c-fos positive

neuron after long-term photostimulation of SNpr-PF terminals. As a positive control for c-fos antibody, we found there is large amounts of c-fos positive staining in the motor cortex after seizure activities (Please see **Figure for reviewer 2** below). Meanwhile, in our study, we found that optogenetic inhibition of SNpr-PF terminals, which may not produce potential antidromic effects on SNr soma, retarded seizure spread in kindling model (**Fig. 4**). Thus, these data suggested that PF might be the main downstream brain region that involved in seizure modulation of SNr PV neurons.

Figure for reviewer 1

Figure for reviewer 2

14. Clarification for Figure S5 – for Panels C-I, how many mice were used for these experiments (I am assuming the individual data points represent cells, not animals?)

Reply: Thanks very much. Indeed, the individual data points in panels C-I of original Fig. S5 represent cells. All these cells are from 8 slices of 6 mice. We have corrected this mistake in the revised manuscript as following:

Page 8 in revised supplementary information:

--“The number of cells (from 8 slices of 6 mice) used in each group is indicated in figure.”--

15. The finding that the neurons that receive input from SNr in the PF thalamus are local circuit is quite interesting – and somewhat surprising given that others (see: Bentivoglio et al., Exp Brain Res 1991; Arcelli et al., Brain Res Bull, 1997) have argued that interneurons are sparse or absent in the intralaminar thalamus of rodents, a profile that differs from that in the cat and primate. The author should discuss this point – are they certain these neurons didn’t project elsewhere?

Reply: Thank you very much for this important comment. In our study, we found that the expression of Arch-eYFP or ChR2-mCherry was localized to the PF in *Vgat::Arch* or *Vgat::ChR2* mice, respectively (**Fig. 5 b and f**), suggesting that GABAergic neurons are local, but not projecting neurons. Meanwhile, the anti-epileptic effect of optogenetic activation of PF GABAergic neurons was reversed by intra-PF application of the GABA_A receptor antagonist bicuculline (**Fig. 5h**), suggesting there is a local GABAergic microcircuit within the PF regulating the seizure propagation.

To further verify that PF GABAergic neurons are local, we checked the Arch-eYFP expression in whole brain slices after injecting very little virus (50 nl) in the PF of *Vgat-cre* mice to prevent viral spillover and found no projecting terminals outside the PF. The typical pictures of viral expression are showed below (**Figure for reviewer 3**). Thus, all above data indicated that GABAergic neurons in the PF are local, but not projecting neurons.

As previous studies suggest that interneurons are sparse or even absent in the intralaminar thalamus of rodents, in the present study we first reveal the function of PF GABAergic neuron in epilepsy. Thus, we emphasized this novelty in the discussion of revised manuscript.

Figure for reviewer 3

16. Fig S7 would benefit from DREADD-negative controls.

Reply: Thanks, we have included this data and corresponding descriptions in the revised **Fig. S9D** in revised manuscript as following:

Revised Fig. S9D

17. Minor comment: The axons of transfected neurons have been reported to remain photoexcitable even when severed from their parent somata^{35, 36}. This line on Pg 14 is out of context.

Reply: Thanks, we have deleted this sentence in our revised manuscript.

Reviewer #2 (Remarks to the Author):

The authors present a commendable enormous amount of work to decipher the SNr components that may control seizures in TLE. They use a combination of techniques to propose that a GABAergic circuit involving the SNr inhibition of PF neurons is central to the control of seizure propagation. The role of the SNr in TLE has been proposed a long time ago; authors using various protocols such as DBS, lesions, and injections of chemical. Note that the authors forgot to mention the seminal work of Karen Gale. The concept is not novel, but the detailed mechanism was not identified.

I have a major concern. The authors claim that "These findings provide a better understanding of the pathological network changes and the precise spatiotemporal control of epilepsy." However, 90% of the "TLE" experiments are done in control animals, which are kindled. This type of acute model can be used to propose testable hypotheses to chronic models. However, only one set of experiments is performed in a chronic model of TLE. As a result, from kindled animals, it is not possible to claim "a better understanding of the pathological network changes" because there are no major network alterations in the progressive kindling model. Likewise, it is not possible to conclude that we get a better understanding of "the precise spatiotemporal control of epilepsy". In TLE (both in patients and chronic models), there are major anatomical and functional alterations in SNr and thalamic nuclei. The circuits are reorganized, and we do not know whether results obtained in control animals can be generalized to chronic model.

There is nothing wrong with doing a study using the kindling model, but the impact of the conclusions is limited, by the very nature of the approach. My evaluation is that relying mostly on kindled normal animals to draw important conclusions on TLE is not enough for Nature Comm. As it stands, the study belongs more to a specialized journal. My evaluation would be entirely different if the authors were to replicate their main findings in the chronic model. They have tried to verify one aspect (but there are major issues there, which are discussed below), so they have the possibility to do it.

Reply: Thank you very much for reviewing our work and propounding valuable comments, which are helpful for us to improve the quality of our paper greatly. Notably, we have included the work of Dr. Karen Gale in our revised manuscript and have verified main findings (Fig.4-6) in chronic KA TLE model. Below are our point-by- point responses to the comments.

Major

1. A major issue is the interpretation of most experimental results. The authors nicely show that a transition activation/inactivation of neurons/axons produce the expected effects in the recorded cells. But these experiments were performed under anesthesia with urethane that directly affects GABAergic transmission, thus acting as a confounding factor. In addition, in the kindling model, the optogenetic manipulation occurs for each stimulation. We do not know how cells will react to a build-up of activation (or CI loading). The authors need to show that the activation/inactivation do work recording units in awake animals (there could be ceiling effects, or reversal, or post inhibitory rebound, which is typical for some PV cells).

Reply: Thank you very much for these important comments. We agree with the reviewer that anesthesia may affect GABAergic transmission and thus acting as a confounding factor for the data interpretation. Accordingly, we have performed additional unit recording experiments in awake mice.

First, we were aimed to tested whether optogenetic stimulation, using the similar parameter in kindling model, were able to reliably activate SNr neuron in awake mice. We verified that there is the increased firing of the SNr GABAergic neurons reliably in response to blue-light stimulation (473 nm, 20 Hz, 10 ms, 5 mW, "30s- on, 10s-off" cycle) in awake *Vgat-ChR2-eYFP* mice (**Fig. S1G**).

Furthermore, in KA-induced chronic awake *PV-ChR2^{SNr-PF}* mice, we found that optogenetic activation of SNr-PF PV projections can inhibit PF GABAergic neurons (2/4) and activated a part of PF glutamatergic neurons (10/24 from 3 mice) reliably during the whole photostimulation period (**Fig. S7C-D**). Interestingly, the modulating effect was much stronger in the initial seconds than that in later stage during the whole photostimulation period (**Fig. S7D**), suggesting that there might be accumulative CI- loading in postsynaptic

neurons that weaken optogenetic manipulation. Nevertheless, optogenetic activation of SNr-PF PV projections can still reliably inhibit PF GABAergic neurons during whole photostimulation period. After photostimulation, we did not detect any post inhibitory rebound firing. Thus, we have added these new data in the revised manuscript as following:

Revised Fig. S1G

Revised Fig. S7C-E

2. Characterization of seizures. Fig 1G and others. I do not understand. You indicated that you filtered the signal at 100Hz. Thus, you cannot see any frequency above 50 Hz at best, or 25 Hz. How did you perform the signal analysis and FFT? In addition, FFT is not appropriate for this type of dynamical signal, wavelet analysis is much better. In addition, you cannot calculate a power spectrum on a signal that is not in steady-state. Perhaps, you can look at seizure onset and offset. You may also consider using the coastline index to characterize seizures.

Reply: Thank you very much for this valuable comment. Actually, in our study, raw EEG signals were sampled at 1000 Hz and recorded with band-pass filters spanning DC to 200 Hz. Spectral analysis was conducted using the fast Fourier transform (FFT), which provided the total power between 0 and 100 Hz. We agree with the reviewer that FFT may not be appropriate for this type of dynamical signal, thus we used wavelet analysis to calculate the power spectrum of EEG during seizures (**revised Figs.1G and 2E**).

Meanwhile, as suggested by the reviewer, we also used coastline index to characterize seizures. We found that photo-activation of SNr PV neurons significantly increased the coastline index of EEG during seizures (**revised Fig. 1G**), while photo-inhibition of SNr PV neurons significantly decreased the coastline index of EEG during seizures (**revised Fig. 2E**). Thus, we have added this data in the revised manuscript as following:

Revised Fig.1G

Revised Fig.2E

Page 31 in revised manuscript:

--“Briefly, raw EEG signals were sampled at 1000 Hz and recorded with band-pass filters spanning DC to 200 Hz. Spectral analysis was conducted using the fast Fourier transform (FFT) or wavelet transform, which provided the total power between 0 and 100 Hz. The coastline index was determined as the sum of the absolute difference between successive points.”--

3. In the acute KA model. What were the criteria to distinguish FS from GS? The data is confusing. If the latencies to GS and SE are 30s and 35s, respectively, how can you have 6 GS on average? In addition, Fig 3C shows that in the CNO condition (bottom panel), the seizure seems very different. Please show the detailed LFP in the three conditions for GS and SE.

Reply: Thanks very much for this comment. In acute KA model, behavior seizure stage accompanied by EEG activities are used to distinguish FS from GS. According to Racine scale, seizure stages 1-3 indicate FSs, and stages 4-5 are GSs. The seizure stage was scored by an investigator blinded to the group allocation. In this acute seizure model, intra-hippocampal KA injection induced high amplitude spike-waves in the EEG and gradually elicited FS, GS and SE within tens of minutes. Seizure events were defined as regular spike clusters with a duration of ≥ 10 s, spike frequency of ≥ 2 Hz and amplitude at least three times of the baseline EEG. Once the mice were in a SE condition, they would still have several GSs during 1.5-h observation period. To make it more clear, we have showed the detailed LFP in three conditions and indicated the timing where GS onset (red arrowheads) in the revised **Fig. 3C**. Meanwhile, we have added detailed description in the revised method.

Revised Fig. 3C

Page 30 in revised manuscript:

--“KA injection induced high amplitude spike-waves in the EEG and gradually elicited FS, GS and SE within tens of minutes. The seizure stage was scored by an investigator blinded to the group allocation during 1.5h observation period.”--

4. The latter leads to a major issue. Is it possible that CNO treatment and the heavy opto stimulations done during kindling change brain states globally? Targeting hub regions, such as SNr or thalamic nuclei may alter global brain dynamics. As a result, the effects may be due to a global alteration versus local one. This should be discussed at least.

Reply: Thank you very much for this insightful comment. Accordingly, we have added some discussion in the revised manuscript as following:

Page 21-22 in revised manuscript:

--“As a common issue in optogenetic and chemogenetic modulation, targeting hub regions, such as the SNr or PF here, may alter global brain dynamics. Therefore, apart from SNr-PF circuit, other potential global alterations may also account for the factors for the regulator seizure in TLE. For example, the superior colliculus, a primary target of SNr output, sends glutamatergic projections to the PF, which was previously reported to be involved in the control of absence seizures⁵⁰. It is possible that the superior colliculus may also have a potential role in seizure of TLE.”--

New added reference:

50. Nail-Boucherie K, Le-Pham BT, Gobaille S, Maitre M, Aunis D, Depaulis A. Evidence for a role of the parafascicular nucleus of the thalamus in the control of epileptic seizures by the superior colliculus. *Epilepsia* 46, 141-145 (2005).

5. Chronic model. This is the model that should be used throughout. First, I advise performing a morphological study to determine the amount of cell loss (in particular PV cells) in SNr and downstream regions. Second, I do not understand the definition used by the authors for FSs. Are they flurries of spikes? Please check the literature for what is accepted as a definition of a TLE seizure. The CNO protocol involves a daily injection in the animal. A control group is required with saline injection. Finally, why are the properties of FSs and GSs in HM3 and HM4 animals so different? They should be the same?

Reply: Thank you very much for those constructive and valuable comments.

First, we performed a morphological study to determine whether the amount of PV cells in SNr are changed in chronic epileptic model. Immunohistochemistry data showed that there is a minor decrease in the number of SNr PV neurons in KA-induced chronic awake mice (**Fig. S7A, B**). This is consistent with previous finding that there is neural loss in the SNr after animals have experienced epilepticus status^{1,2}.

Second, in KA-induced chronic TLE model, spontaneous seizure events were defined as regular spike clusters with a duration of ≥ 10 s, spike frequency of ≥ 2 Hz and amplitude at least three times of the baseline EEG according to previous study³. In the present model, most of seizures we recorded were characterized by bursts of high-voltage sharp waves (revised **Fig. 3E**, upper panel) associated with any obvious behavioral alterations in the parallel video recordings. As previously reported⁴, those seizures only occurred in the ipsilateral hippocampus and usually were interpreted as non-convulsive focal seizures (FSs). In addition to FSs, mice also exhibited infrequent tonic-clonic generalized seizures (GSs) that were associated with typical paroxysmal EEG activity with obvious post-seizure depression (revised **Fig. 3E**, lower panel). To make it more clear, we have showed typical FS and GS in the revised **Fig. 3E** and added detailed description in the revised method.

Third, as also suggested by the first reviewer, to test whether the daily injection of CNO would have any seizure-modifying effect in chronic epileptic mice, we injected AAV-ef1a-DIO-mCherry control virus into the SNr of *PV-cre* mice (*PV-mCherry*^{SNr} mice). We found that CNO treatment did not change the frequency and seizure duration of both FS and GS in *PV-mCherry*^{SNr} mice, suggesting anti-seizure effect of CNO in *PV-hM4Di*^{SNr} mice may not be associated with off target effects of CNO or daily injection manipulation. Thus, we have added this result in the revised **Fig. 3G**.

Finally, we have also noticed that the properties of FSs and GSs in HM3 and HM4 animals have a big variety. The precise reason is still unknown. We speculate this can be caused by the methodology of chronic KA model in our study. We stereotaxically injected KA into right dorsal hippocampus to induce SE in anesthetized mice. The mice were allowed to recovery from anesthetized without SE termination. This may lead to different SE duration in different mice, which may be an important contributor to the big variety among different mice. Thus, to make a fair comparison, we use self-control in chronic KA experiment.

Revised Fig. S7A, B

Revised Fig. 3D-G

Page 30 in revised manuscript:

--“First, we stereotaxically injected KA into right dorsal hippocampus to induce SE in anesthetized mice. The mice were allowed to self-terminate and recovery from anesthesia state.”--

Page 30-31 in revised manuscript:

--“Most of seizures we recorded were characterized by bursts of high-voltage sharp waves and are not associated with any obvious behavioral alterations in the parallel video recordings, which usually were interpreted as non-convulsive FSs as previous study⁶⁵. In addition to FSs, mice also exhibited infrequent

tonic-clonic GSs that were associated with typical paroxysmal EEG activity with obvious post-seizure depression.”--

References:

1. Lindvall, O., M. Ingvar, and F.H. Gage, Short-Term Status Epilepticus in Rats Causes Specific Behavioral Impairments Related to Substantia-Nigra Necrosis. *Experimental Brain Research*, 1986. 64(1): p. 143-148.
2. Schmidtkastner, R., C. Heim, and K.H. Sontag, Damage of Substantia-Nigra Pars Reticulata during Pilocarpine-Induced Status Epilepticus in the Rat - Immunohistochemical Study of Neurons, Astrocytes and Serum-Protein Extravasation. *Experimental Brain Research*, 1991. 86(1): p. 125-140.
3. Bragin, A., A. Azizyan, J. Almajano, C.L. Wilson, and J. Engel, Analysis of chronic seizure onsets after intrahippocampal kainic acid injection in freely moving rats. *Epilepsia*, 2005. 46(10): p. 1592-1598.
4. Riban, V., V. Bouillere, B.T. Pham-Le, et al., Evolution of hippocampal epileptic activity during the development of hippocampal sclerosis in a mouse model of temporal lobe epilepsy. *Neuroscience*, 2002. 112(1): p. 101-111.

6. To convince us that the work is pertinent to TLE, you need to perform the exps described in Figs 4-6 in the chronic model.

Response: Thank you very much for this constructive comment. Accordingly, we have performed additional experiment with KA-induced chronic epileptic model to verify the main finding in Figs. 4-6. In KA-induced chronic awake mice, although there is a minor reduce in the number of SNr PV neurons (**Fig. S7A, B**), we found that optogenetic activation of SNr-PF PV projections can also inhibit 2/4 putative GABAergic neurons and activated a part of putative glutamatergic neurons (10/24 from 3 mice) reliably during the whole photostimulation period (**Fig. S7C-E**). Furthermore, chemogenetic inactivation of SNr-PF PV projections by intra-PF injection of CNO, or chemogenetic activation of PF GABAergic neurons, similarly alleviated the severity of spontaneous seizure in KA-induced chronic epileptic model (**Fig. S7F, G**). Thus, all these above data suggested that SNr-PF neural circuit is involved in seizure control in KA-induced chronic epileptic model.

Revised Fig. S6

Figure S7 | SNr-PF circuit is involved in seizure control in KA-induced chronic epileptic model. (A, B) Representative immunohistochemistry images (A) and statistical data (B) showing the number of PV neurons in the SNr from WT and KA mice. Bar, 300 μ m. (C) Scheme of experiment for *in vivo* single-unit recordings in the PF of awake *PV-ChR2^{SNr-PF}* chronic epileptic mice. (D) Representative peri-event raster histograms showing the firing rate of PF GABAergic and glutamatergic neurons in response to photo-activation of SNr-PF PV projections; (E) Quantification of the number of PF GABAergic and glutamatergic neurons in response to photo-activation of SNr-PF PV projections from 3 *PV-ChR2^{SNr-PF}* mice. (F) Effects of chemogenetic inhibition of SNr-PF PV projection in *PV-hM4Di^{SNr-PF}* mice on the number and duration of seizures in KA-induced chronic epileptic model. Wilcoxon matched-pairs test, * $P < 0.05$ compared to Pre; # $P < 0.05$ compared to Post. (G) Effects of chemogenetic activation of PF GABAergic neurons in *PV-hM3Dq^{PF}* mice on the number and duration of seizures in KA-induced chronic epileptic model. Wilcoxon matched-pairs test, * $P < 0.05$ compared to Pre; # $P < 0.05$ compared to Post. The number of neurons and mice used in each group is indicated in figure. All the data are presented as mean \pm S.E.M.. Source data are provided as a Source Data file.

7. Fig S5. I am not able to evaluate the in vitro part. Critical details are missing in the method section. How the different parameters were measured is not indicated. You need to provide how many cells were recorded from how many slices from how many animals.

Reply: Thanks very much. We have added more detailed method for measure different parameters in electrophysiological experiment. The individual data points in panels C-I of original Fig. S5 represent cells. All these cells are from 8 slices of 6 mice. Thus, we have added those detailed descriptions in the revised manuscript as following:

Page 34-35 in revised manuscript:

--“To record optical evoked GABAergic postsynaptic currents, recording pipettes were routinely filled with a solution containing the following (in mM): 140 CsCl, 5 NaCl, 10 HEPES, 0.2 EGTA, 2 Mg-ATP, 0.3 Na₃GTP, 5 QX314, 10 Na₂-phosphocreatine, pH 7.2. Neurons were held at -70 mV. The inhibitory postsynaptic currents (IPSCs) were isolated by addition of 6-cyano-7-nitroquinoxaline-2,3-dione (CNQX; 20 μ M) and (2R)-amino-5-phosphonovaleric acid (D-AP5; 20 μ M) to the ACSF to block excitatory transmission mediated by AMPA/kainite and N-methyl-D-aspartate (NMDA) receptors, respectively. Experiments were performed in the presence of the sodium channel blocker tetrodotoxin (TTX, 1 μ M) and the potassium channel blocker 4-aminopyridine (4-AP, 100 μ M) to isolate the monosynaptic current while shining blue light on the surface of brain slices using an optic fiber connected to a blue laser power source (IKECOOL-Laser 473 nm). Light pulse (1 Hz, 10 ms, 10 pulses, 2 mW) was controlled with EPC10 patch-clamp amplifier (HEKA Instruments). To record the action potential firing properties of glutamatergic and GABAergic neurons in the PF, a series of hyperpolarizing and depolarizing current steps (-100 pA- +100 pA, 100 pA each a step, 500 ms) were applied to determine firing properties of PF neurons. A series of 300-ms depolarizing current pulses (increased in 5-pA steps) were applied to measure the threshold, amplitude, and the half-wave width of the action potentials. The pipette solution contained the following (in mM): 140 K-gluconate, 5 NaCl, 2 Mg-ATP, 0.2 EGTA, and 10 HEPES.”--

Page 7 in revised supplementary information:

--“The number of cells (from 8 slices of 6 mice) used in each group is indicated in figure.”--

Minor.

1. Fig S1C and elsewhere. With n=3 mice and few neurons, how did you assess inter-animal difference assessed? What were the criteria to decide that cells were excited/inhibited? What was the repartition per mouse?

Reply: Thank you very much for these comments.

The neuronal response repartition per mouse in Fig. S1C was showed in the below table. We used Chi-square test and found there is no inter-animal statistical differences (Chi-square=1.11, df=4, p=0.8927).

Meanwhile, as previous study¹, the criteria used to define an “excited” or “inhibited” neuronal response were as follows: firing rates were considered to be significantly different if they were >2 SDs of baseline averages. Briefly, the average firing rate during each 10-s-duration bin was calculated in 1-min baseline period. Then, the average firing rate of the seizure period or photostimulation was calculated and compared with that of the baseline period to test whether firing rate was >2 SDs greater or less than the baseline average.

To make it more clear, we have included raw data in the “**source data file**” and added the criteria of “excited/inhibited” in the revised method as following:

Table for neuronal response repartition per mouse in Fig. S1c

Mouse No.	Increase	Decrease	No change
Mouse 1	7	1	6
Mouse 2	11	2	5
Mouse 3	3	1	2

Page 33-34 in revised manuscript:

--“The criteria used to define an “excited” or “inhibited” neuronal response were as follows: firing rates were considered to be significantly different if they were >2 SDs greater or less than baseline averages⁶⁶. Briefly, the average firing rate during each 10-s-duration bin was calculated in 1-min baseline period. Then, the average firing rate of the seizure period or photostimulation was calculated and compared with that of the baseline period to test whether firing rate was >2 SDs greater or less than the baseline average.”--

Reference:

1. Toyoda I, Fujita S, Thamattoor AK, Buckmaster PS. Unit Activity of Hippocampal Interneurons before Spontaneous Seizures in an Animal Model of Temporal Lobe Epilepsy. *Journal of Neuroscience* 35, 6600-6618 (2015).

2. Chronic model. Note that ref 28 is not appropriate as the intra-hippocampal injection came much later. Note also that CNO is not inert. Some papers showed that it does have a physiological impact.

Reply: Thank you very much for this comment. Accordingly, we have revised original ref28. Meanwhile, to test whether the CNO itself would have any seizure-modifying effect in chronic epileptic mice via physiological impacts, we injected AAV-ef1a-DIO-mCherry control virus into the SNr of *PV-cre* mice (*PV-mCherry^{SNr}* mice). We found that CNO treatment did no change the frequency and seizure duration of both FS and GS in *PV-mCherry^{SNr}* mice, suggesting anti-seizure effect of CNO in *PV-hM4Di^{SNr}* mice may be due to its chemogenetic modulation, but not be associated with physiological impacts of CNO itself, Thus, we have added this result in the revised Fig. 3G.

Revised Fig. 3G

Revised reference:

34. Davenport CJ, Brown WJ, Babb TL. Sprouting of Gabaergic and Mossy Fiber Axons in Dentate Gyrus Following Intrahippocampal Kainate in the Rat. *Experimental neurology* 109, 180-190 (1990).

3. Fig 3H1, the two panels are identical.

Reply: Thanks. We have corrected this careless mistake in revised Fig. 3I as following:

Revised Fig. 3I

4. Fig S4G, why measuring the duration of the motor phase in this experiment and not for the other ones?

Reply: Thank you for this comment. We found that photo-inhibiting PV projections of the SNr-RT, SNr-RE or SNr-VM had no effect on kindling acquisition. As a positive control for effectiveness of optogenetic modulation, we tested only one neural pathway (SNr-VM) to debase the workload. We found that the inhibition of SNr-VM circuit increased the clonic motor phase in seizure stage 3 of the kindling seizures, which is consistent with previous studies that the VM is an important site for motor function.

Reviewer #3 (Remarks to the Author):

The manuscript reports a new projection from SNr PV neurons to GABAergic cells in Pf thalamus and suggests that this projection is involved in TLE seizure control. The existence of GABAergic cells in Pf and the relevance of these in controlling TLE seizure severity is novel, interesting, and surprising given how small the proportion of GABA cells is in Pf thalamus (only 5%). Also, it is interesting that the inhibitory output from SNr PV neurons to the thalamic PF, but not to the RT, RE or VM, is involved in the TLE seizure control. However, I have some concerns regarding data interpretation and clarity that need to be addressed to support the main conclusions.

Reply: Thank you very much for reviewing our work and propounding valuable comments, which are helpful for us to improve the quality of our paper significantly. Below are our point-by-point responses to the comments.

Major Concerns:

1. The authors claim that SNr PV cells control TLE seizures via a new “disinhibitory” projection from SNr PV cells to GABAergic cells in Pf, rather than via the direct projection of SNr to glutamatergic cells. “Quantity analysis showed the number of SNr PV neurons targeting PF glutamatergic neurons was much lower than that targeting PF GABAergic neurons (Fig. S6).” However, SNr PV stimulation inhibits a non-negligible population of Pf GLU cells as well (see Fig 5B). In fact, SNr stimulation inhibits the same number of GLU and GABA cells in Pf according to Fig 5B. How do the authors explain the inconsistencies between conclusions from Fig S6 and conclusions from Fig 5B (see also Fig S9)?

In other words, in light of the results shown in Fig. 5B and Figs. S5 and S6, the overall working model is unclear. The proportion of GABA vs GLU cells in Pf is only 5% in the Pf nucleus (see Fig S5A); Moreover less than 20% of these GABA Pf cells receive a functional input from SNr (Fig S5, patching results: only 3 GABA cells out of 17 receive SNr inputs). Could this low number be due to sub-optimal recording conditions? What is the rationale for using 4-AP and TTX when recording evoked IPSCs from SNr to Pf thalamus?

Can the authors speculate on how the SNr projection onto just 1% of Pf nucleus is more important for seizure control than the direct targeting of GLU Pf cells by SNr (which would represent a higher number of cells according to Fig 5B, although inconsistent with Fig S6)?

Reply: Thank you very much for all these important comments.

We have also noticed that there seems to be a discrepancy between structural connection and functional connection. Indeed, quantity analysis of viral tracing data showed the number of SNr PV neurons targeting PF glutamatergic neurons was much lower than that targeting PF GABAergic neurons. While, electrophysiological data, including extracellular recording and patch recording, was only used to verify there is a functional input from SNr PV neuron. The success rate of functional connection is extremely low, especially in patch data (only 3 GABA cells out of 17 receive SNr inputs), the possible reason may be: (1) SNr PV neurons may project to a small part of PF GABAergic neurons. In our electrophysiological experiment, it is possible that not all the recorded PF GABAergic neurons actually receive projections from the SNr. (2) The viral-mediated ChR2 expression at the presynaptic part may be low efficient, and the light level is not able to induce GABAergic responses in postsynaptic PF GABAergic neuron¹. Similar low efficient synaptic transmission in electrophysiological data has been described in many previous reports²⁻⁴. (3) As reviewer referred, this can also be due to sub-optimal recording conditions. To isolate monosynaptic current, experiments were performed in the presence of TTX and 4-AP to block indirect circuit, similar as previous studies^{5,6}. Together, our anatomical and functional evidence at least indicate that a portion of SNr PV neurons directly innervate the PF GABAergic neurons.

Thus, to make it more clear, we have added some explanation in the revised manuscript as following:

Page 15 in revised manuscript:

--“The low percentage of PF GABAergic neurons receiving projections from SNr may relate to the following three reasons: (1) SNr PV neurons may project to a small part of the posterior PF GABAergic neurons. (2)

The efficacy of virus-mediated ChR2 expression at the presynaptic part is low, and the light level is not able to induce GABAergic responses in postsynaptic PF GABAergic neurons. (3) Sub-optimal recording conditions; to isolate monosynaptic current, experiments were performed in the presence of TTX and 4-AP to block indirect circuit.”--

References:

1. Cardin, J.A., M. Carlen, K. Meletis, et al., Targeted optogenetic stimulation and recording of neurons in vivo using cell-type-specific expression of Channelrhodopsin-2. *Nature Protocols*, 2010. 5(2): p. 247-254.
2. Zhou, M., Z.H. Liu, M.D. Melin, et al., A central amygdala to zona incerta projection is required for acquisition and remote recall of conditioned fear memory. *Nature Neuroscience*, 2018. 21(11): p. 1515-+.
3. Rodriguez, E., K. Sakurai, J.N. Xu, et al., A craniofacial-specific monosynaptic circuit enables heightened affective pain. *Nature Neuroscience*, 2017. 20(12): p. 1734-+.
4. Knowland, D., V. Lilascharoen, C.P. Pacia, et al., Distinct Ventral Pallidal Neural Populations Mediate Separate Symptoms of Depression. *Cell*, 2017. 170(2): p. 284-+.
5. Owen, S.F., J.D. Berke, and A.C. Kreitzer, Fast-Spiking Interneurons Supply Feedforward Control of Bursting, Calcium, and Plasticity for Efficient Learning. *Cell*, 2018. 172(4): p. 683-+.
6. Besnard, A., Y. Gao, M.T. Kim, et al., Dorsolateral septum somatostatin interneurons gate mobility to calibrate context-specific behavioral fear responses. *Nature Neuroscience*, 2019. 22(3): p. 436-+.

In Fig S7, the authors nicely show that the chemogenetic activation of PF GABAergic neurons blocks pro-epileptic effects of optogenetic activation of SNr-PF GABAergic projections. However, I am not convinced that this result can be interpreted as if the pro-epileptic effect of SNr activation was due 100% to GABA Pf cells and did not involve SNr projections to GLU Pf cells (because see Fig 5B, there are GLU Pf cells that seem to be inhibited by SNr). Thus, the fact that the pro-epileptic effect of SNr activation was reduced in the Fig S7 experiment could be due to a combined effect of a direct inhibition of Pf GLU cells by SNr AND to inhibition of Pf GLU cells by chemogenetic activation of GABA Pf cells. The authors do claim that GLU Pf neurons are important for SNr-mediated modulation of TLE seizures (Fig S9 and last part of the discussion), but it is unclear how the authors reconcile the direct SNr effects on GABA vs GLU Pf cells.

Fig. 5: The authors conclude that the anti-epileptic effect of optogenetic activation of PF GABAergic neurons was reversed by intra-PF application of the GABAA receptor antagonist bicuculline. However, bicuculline could affect both glutamatergic and GABAergic cells in PF, both of which receive GABAergic inputs from SNr. This caveat should be addressed in the text because it affects data interpretation.

In sum, it would significantly strengthen the manuscript if the authors could propose a working model / diagram based on their results. The authors should edit the text to make it very clear what would one would expect from activation or inhibition of SNr PV cells on seizures with and without existence of GABAergic Pf cells (even if these are present in relatively small numbers). How does SNr PV projection compare between GABAergic and glutamatergic Pf cell population? This is key for speculating about the relevance of the “new” circuit. In other words, the presence of GABAergic cells in Pf is novel in this manuscript, so the authors should clarify with a concluding/speculative diagram how the presence of these cells and the fact they receive projections from SNr would actually affects the effect of SNr on TLE seizures.

Reply: Thank you very much for all these important and constructive comments. We totally agree with the reviewer that the chemogenetic activation of PF GABAergic neurons can indirectly inhibit PF glutamatergic neurons, which also blocks pro-epileptic effects of optogenetic activation of SNr-PF GABAergic projections. Indeed, *in vivo* single-unit recordings showed that photo-activation of SNr-PF PV axons reliably and quickly inhibited the firing rate of 6/12 putative GABAergic neurons recorded in the PF. While the firing change of PF glutamatergic neurons in response to photo-activation of SNr-PF PV axons was heterogeneous: 5/26 neurons were decreased, 6/26 neurons were increased and 15/26 neurons had no change from 6 *PV-ChR2^{SNr-PF}* mice (**Fig. 5B**). Similarly, we also verified this phenomenon in KA-induced chronic awake mice (**Fig. S6C-**

E). These data suggested that there is both direct and indirect neural connection between SNr PV neuron and PF glutamatergic neurons. Further, we found that during kindling-induced hippocampal seizures, about 54.5% of GABAergic neurons in the PF decreased their firing rate during seizures (1/21 neuron was excited, 11/21 neurons were inhibited, 9/21 neurons were not changed). Whereas, 50% of glutamatergic neurons in the ipsilateral PF increased their firing rate during seizures (16/33 neurons were excited, 1/33 neuron was inhibited, 15/33 neurons had no response, **Fig. S9A-S9C**). Optogenetic activation of PF GABAergic neurons inhibited surrounding glutamatergic neurons locally in the *Vgat-ChR2^{PF}* mice (Fig. S9D), and direct optogenetic inhibition of PF glutamatergic neurons produced an anti-epileptic effect on seizure development. These data indicated that indirect neural connection between SNr PV neuron and PF glutamatergic neurons may contribute more to the seizure. Although our data cannot support the seizure modulation effect of SNr PV neuron is 100% due to the GABAergic neuron-mediated indirect pathway, it at least demonstrated that PF GABAergic neuron and surrounding PF glutamatergic was involved in seizure modulation. Thus, as reviewer suggested, we have added a working model in the revised **Fig. S11** and carefully edited the text to make it clear how SNr PV neurons modulate PF neurons and seizure of TLE.

In addition, we agree with the reviewer that bicuculline is not cell type specific, as it could affect both glutamatergic and GABAergic cells in the PF. Thus, we have indicated this caveat in the revised manuscript.

Revised Fig. S11

Figure S11 | Summary of a disinhibitory nigra-parafascicular neural circuit in seizure in temporal lobe epilepsy.

Page 14 in revised manuscript:

--“The firing change of PF glutamatergic neurons in response to photo-activation of SNr-PF PV axons was heterogeneous: 5/26 neurons were decreased, 6/26 neurons were increased and 15/26 neurons had no change from 6 *PV-ChR2^{SNr-PF}* mice (no inter-animal statistical difference, *Chi-square* test, $p=0.5875$, Fig. 5B). This data suggested that there might be both direct and indirect neural connection between SNr PV neuron and PF glutamatergic neurons.”--

Page 17 in revised manuscript:

--“Whereas, 50% of glutamatergic neurons in the ipsilateral PF increased their firing rate during seizures (16/33 neurons were excited, 1/33 neuron was inhibited, 15/33 neurons had no response, Fig. 7A-C), suggesting PF glutamatergic neurons may be indirectly disinhibited by SNr PV neuron and thus involved in seizure.”--

Page 19 in revised manuscript:

--“suggesting PF glutamatergic neurons were a part of the circuit regulating seizure activities of TLE in a bidirectional manner (see summary diagram in Fig. S11)”--

Page 20 in revised manuscript:

--“suggesting PF glutamatergic neurons were a part of the circuit regulating seizure activities of TLE in a bidirectional manner (see summary diagram in Fig. S11)”--

2. Given that the authors claim that the output of Pf cells is important for controlling seizures (which is the major claim of the study), I recommend showing the firing of these cells during seizures in the main figures rather than in the Supplemental figure 9.

Reply: Thank you very much for this comment. We have moved this part of data to the main figure (Please see in new **Fig. 7**).

3. Given that the existence of GABAergic cells in Pf thalamus is novel (because the Pf neurons are thought to be only glutamatergic), it is important to characterize the properties of these cells. Fig S5 panel C shows an unnaturally large current injection of -500 pA and the traces are not representative of the average results (e.g firing). I recommend finding more representative voltage traces in response to positive and negative current pulses.

Reply: Thank you very much for this comment. We have added more representative voltage traces with -100 pA injection of hyperpolarizing current in the revised **Fig. S6** as following:

Revised Fig. S6C

4) The manuscript contains numerous errors and lacks clarity.

a) Page 3, Introduction: “Basal ganglia circuits are closely involved in the modulation, propagation, and cessation of several types of epileptic seizures, including TLE 7, 8.” Please note that TLE is not an epileptic seizure. TLE is a form of epilepsy. This sentence needs to be rewritten. Suggestion: “Basal ganglia circuits are closely involved in the modulation, propagation, and cessation of seizures in different types of epilepsies, including TLE 7, 8.”

-Also, the references 7 and 8 do not encompass the broad claim about basal ganglia controlling several types of seizures. Basal ganglia have also been shown to control absence type epileptic seizures (see work from the Charpier group on how SNr projections control thalamocortical oscillations in epilepsy, e.g., PMID:17251435). The authors cite the references correctly in the discussion but not in the introduction.

Reply: Thanks. We have revised these sentences as following:

Page 3 in revised manuscript:

--“Basal ganglia circuits are closely involved in the modulation, propagation, and cessation of seizure in different types of epilepsies, including TLE ^{7, 8, 9}.”--

b) Page 3, Introduction, The reference cited in this sentence is incorrect:

“The substantia nigra pars reticulata (SNr), a region that mainly contains GABAergic neurons, controls the activity of both corticothalamic and limbic networks through its primary GABAergic output, acting like the “choke point” of basal ganglia ⁹”. Reference 9 corresponds to: “Paxinos G. The rat nervous system. (ed[^](eds). Fourth edition. edn. Elsevier/Academic Press, (2015)” and is not the right reference for the notion of choke points. For the notion of choke points, the authors should cite

relevant manuscripts (e.g the reference 2 from this manuscript which is the one that focuses on the notion of choke points).

Reply: Thanks. We have changed references in this sentence as following:

Page 3 in revised manuscript:

--“The substantia nigra pars reticulata (SNr), a region that mainly contains GABAergic neurons, controls the activity of both corticothalamic and limbic networks through its primary GABAergic output, acting like the “choke point” of basal ganglia^{10, 11}--

Revised references:

10. Alexander GE, Crutcher MD. Functional Architecture of Basal Ganglia Circuits - Neural Substrates of Parallel Processing. Trends in neurosciences 13, 266-271 (1990).

11. Parent A, Hazrati LN. Functional-Anatomy of the Basal Ganglia .1. The Cortico-Basal Ganglia-Thalamo-Cortical Loop. Brain Research Reviews 20, 91-127 (1995).

c) Intro page 4: What neurons do the authors refer to by “these neurons” in the following section:

“We identified a previously unknown nigra-parafascicular disinhibitory circuit for regulation of seizure in TLE. We found that selective activation of these neurons amplifies seizure activities, whereas inhibition of SNr PV neurons alleviated the severity of epileptic seizures”. Do they mean the following: “We found that selective activation of SNr PV neurons amplifies seizure activities, whereas their inhibition alleviated the severity of epileptic seizures”?

Reply: Thanks. We have revised this sentence as following:

Page 4 in revised manuscript:

--“We found that selective activation of SNr PV neurons amplifies seizure activities, whereas their inhibition alleviated the severity of epileptic seizures.”--

d) In the same paragraph, the authors write:

“[...] the SNr PV neurons sent long-range axons to the GABAergic neurons in posterior parafascicular nucleus (PF), and optogenetic manipulation of this disinhibitory nigra-parafascicular circuit bidirectionally modulated seizures in TLE.” This statement could be interpreted as if SNr PV neurons do not project to glutamatergic neurons in Pf that are known to be the major cell type in Pf. However, SNr PV projections to Pf do not seem to be cell-type specific (according to the text) because SNr PV cells seem to be projecting to Pf glutamatergic cells as well (e.g Fig 5B), suggesting that the direct SNr->GLU Pf pathway needs to be considered when interpreting the results from Fig S7.

Reply: Thanks. To make it more clear, we have revised this sentence as following:

Page 4 in revised manuscript:

--“Subsequent *in vivo* and *in vitro* electrophysiology combined with rabies virus-assisted circuit-mapping revealed that the SNr PV neurons form structural and functional connection with neurons in the parafascicular nucleus (PF), among which nigra-parafascicular disinhibitory circuit is involved in bidirectional modulation of seizures in TLE.”--

e) The authors use interchangeably different expressions to describe the model: “KA-induced chronic epileptic model” and “hippocampal kindling model” (e.g. in the titles of Figures 1, 2 vs Figure 3. To avoid confusion, I suggest to use the same wording.

Reply: Thanks. Actually, the data in **Fig.1 and 2** is from hippocampal kindling model, while the data in **Fig. 3** is from kainic acid (KA)-induced spontaneous seizure model. To avoid confusion, we revised the title of **Fig. 3** as following:

Page 47 in revised manuscript:

--“Figure 3 | SNr PV neurons bidirectionally modulate the severity of epileptic seizures in kainic acid (KA)-induced spontaneous seizure model.”--

f) The legend of the Fig. S1G which should be key for showing that SNr stimulation actually controls TLE seizures needs to be clarified. The authors need to explain in this legend what they mean by “Blue-Imm.” vs “Blue Pre” and “Blue Delay”. The authors should explain in the legend what the “different photostimulation timing” means in the following sentence “The effect of optogenetic activation of SNr GABAergic neurons with different photostimulation timing on the development of seizure stage (G-1).” I recommend adding a diagram showing the timing of photostimulation.

Reply: Thanks very much for this suggestion. We have added description in the figure legend and further added diagram for the timing of photostimulation in the revised **Fig. S1H** as following:

Revised Fig. S1H

Page 2 in revised supplementary information:

--“Yellow group means yellow light stimulation immediately after each kindling stimulation, Blue-Imm group means blue light stimulation immediately after each kindling stimulation, Blue-Pre group means blue light stimulation before each kindling stimulation, while Blue-Delay group means blue light stimulation after each kindling stimulation with a 10s delay.”--

g) Fig S3:

-VTN (ventral thalamic nucleus) is mentioned in the legend, but I can't find VTN in the figure panels. Indicate in the panels, or remove from the legend. Also, Po is indicated in panels, but the abbreviation is not defined in the legend.

-Panel C: the authors claim in the legend and in the main text that SNr projects to RT but I can't see any projections from SNr to RT in the panels. If the authors want to report this projection, I suggest adding a zoom on the RT to show presence of SNr fibers/synaptic boutons.

Reply: Thanks. We have defined abbreviation for Po and deleted VTN in the legend. Meanwhile, we showed a zoom on the RT in the revised **Fig. S4C**. Indeed, there is only a few fibers from SNr in the RT at -1.00 mm slice, while there are much more fibers from SNr at anterior RT (**Fig. S5A**).

Page 5 in revised supplementary information:

--“Po, posterior thalamic nuclear group.”--

Revised Fig. S4B

h) Fig S1: legend: In the sentence: “...ipsilateral SNr from 3 wildtype mice”: the authors should clarify ipsilateral to what?

Reply: Thanks. We have revised description in the figure legend as following:

Page 1 in revised supplementary information:

--“Statistic of responses of recorded SNr GABAergic neurons ipsilateral to kindling site during kindling-induced seizures.”--

i) There are many grammatical and typing mistakes throughout the text. It would be too long to list all of them.

E.g., “These results indicate that activation of SNr GABAergic neurons is sufficient to amplifies seizure activities in hippocampal kindling model”.

Reply: Thanks. The English of manuscript have been re-checked by a native English speaker. All the revisions in the main manuscript are highlighted by red color. The example sentence has been revised as following:

Page 7 in revised manuscript:

--“These results indicate that activation of SNr GABAergic neurons is sufficient to amplify seizure activities in hippocampal kindling model.”--

Reviewer #4 (Remarks to the Author):

In this study Chen et al analyzed the involvement of the substantia nigra pars reticulata (SNr) in temporal lobe epilepsy (TLE) by employing the kindling and the kainic acid animal models. They report that optogenetic or chemogenetic activation of SNr parvalbumin+ (PV) interneurons “amplifies” seizures activities while their inhibition alleviates them. They also state that seizure severity was bidirectionally regulated by optogenetic manipulation of SNr PV fibers projecting to the parafascicular nucleus (PF), and found with electrophysiology combined with rabies virus-assisted circuit-mapping that SNr PV neurons directly project to and functionally inhibit posterior PF GABAergic neurons. They conclude that their findings reveal “a previously unknown long-range SNr-PF disinhibitory circuit that modulate seizures in TLE, and that inactivation of this circuit can alleviate epileptic seizures.

This is a complex study that includes a variety of experimental approaches providing a huge amount of data. The topic is relatively unexplored and the findings original and consistent with the conclusion that the SNr-PF circuit modulates seizures. However, at this stage, this study is characterized by a rather superficial analysis of the results, which may be due to the amount of different experiments performed. Perhaps the authors should focus on some specific experiments and provide full, in depth information on them.

Reply: Thank you very much for reviewing our work and propounding valuable comments, which are helpful for us to improve the quality of our paper significantly. Below are our point-by-point responses to the comments.

Some specific critiques are as follows.

1. In the main body of the paper it should be specified whether the “CA3 kindling” was acute or the classic chronic kindling. This point is of paramount importance to follow what written in the Results (p. 5), namely “Immediately after the initiation of CA3 kindling, about 55% of GABAergic neurons in the ipsilateral SNr increased their firing rate (peak firing rate was 17.06 ± 1.01 Hz; out of 38 recorded neurons from 3 wildtype mice, 21 neurons were excited, 4 inhibited, 13 no response; no inter-animal statistical difference was detected, Chi-square test, $p=0.8927$, Fig. S1B and F1C).” In fact, it is obscure to me whether these changes occurred during the kindling stimulation or after several days of kindling.

Reply: Thank you very much for this comment. In the present study, we used rapid hippocampal kindling model in all optogenetic modulation and fiber photometry experiments. Mice are received 10 kindling stimulations daily and they usually get to fully kindled state within three days. While, in single unit recording experiment, we test firing response of SNr PV neuron to kindling-induced acute seizure in urethane-anesthetized mice, otherwise there would be amounts of electrophysiological artifacts due to seizure movements in awake mice. To make it more clear, we have added detailed description in revised manuscript as following:

Page 5 in revised manuscript:

--“Immediately after the initiation of CA3 kindling, about 55% of GABAergic neurons in the ipsilateral SNr increased their firing rate during kindling-induced acute seizure (peak firing rate was 17.06 ± 1.01 Hz; out of 38 recorded neurons from 3 wildtype mice, 21 neurons were excited, 4 inhibited, 13 no response; no inter-animal statistical difference was detected, Chi-square test, $p=0.8927$, Fig. S1B and F1C).”--

Page 5-6 in revised manuscript:

--“We detected an increase of calcium response (average signal peak $\Delta F/F$: $2.02 \pm 0.15\%$ for focal seizures and $3.24 \pm 0.54\%$ for generalized seizures, Fig. 1B) in the ipsilateral SNr during hippocampal kindling-induced seizure (monophasic square-wave pulses, 20 Hz, 1 ms/pulse, 40 pulses). Calcium signals of SNr PV neurons gradually increased along with the development of seizure severity during three-days kindling acquisition (Fig. S2).”--

2. The same critique applies to what stated in p. 6: In the hippocampal kindling model, we applied 30 s photostimulation in the ipsilateral SNr immediately after hippocampal kindling...” More specific information on the timing of optogenetic stimulation in relation with the kindling procedure must be given.

Reply: Thanks very much for this comment. We have added detailed description in the figure legend and further added diagram for the timing of photostimulation in the revised **Fig. S1H** suggested by the third reviewer.

Page 6-7 in revised manuscript:

--In the hippocampal kindling model, we applied 30 s photostimulation in the ipsilateral SNr immediately after each hippocampal kindling stimulation (similar to the closed-loop stimulation pattern²⁹, Fig. S1E)--

Page 2 in revised supplementary information:

--“Yellow group means yellow light stimulation immediately after each kindling stimulation, Blue-Imm group means blue light stimulation immediately after each kindling stimulation, Blue-Pre group means blue light stimulation before each kindling stimulation, while Blue-Delay group means blue light stimulation after each kindling stimulation with a 10s delay.”--

Revised Fig. S1H

3. I have some concerns with the “negative” data obtained by stimulating SNr NOS positive neurons (see p. 8), which according to these authors is the second largest population of GABAergic cells in the SNr. Did they check for expression? How come, if effectively activated by optogenetic procedures these interneurons did not influence the “kindling” process?? In fact, were they optogenetically excited??

Reply: Thank you very much for this insightful comment. We have checked NOS expression in immunohistochemical experiment and found that there is a small amount of NOS positive neurons in the SNr (revised **Fig. S3A**). Meanwhile, after light stimulation in the SNr of *NOS-ChR2^{SNr}* mice, our preliminary data showed there are a lot of NOS positive neurons are c-fos positive, suggesting that NOS neurons can be optogenetically excited. Interestingly, we found that optogenetic activation of SNr NOS positive neurons in *NOS-ChR2^{SNr}* mice had no effects on seizure propagation in kindling model (**Fig. S3B,C**). The potential reasons may: (1) the NOS neuron and PV neuron have different downstream targeting regions that are differently involved in seizure circuits; (2) the number of NOS neurons is much less than that of PV neurons, which make it less effective to be involved in seizure control. Thus, according to reviewer’s comment, we have added this data into the revised **Fig. S3**.

Revised Fig.3

4. As remarked above, with regard to the kindling experiments, also the chemogenetic procedures applied to kainic acid treated mice (pp. 10 and 11) lack experimental details.

Reply: Thanks. We have added experimental details for chemogenetic modulation in the revised manuscript as following:

Page 32 in revised manuscript:

--“Then, the mice were injected (i.p., 1 mg/kg) with CNO daily for 7 days (0.5 h before EEG recording each day) to test the effect of the chemogenetic modulation on chronic spontaneous seizures.”--

5. There is actually some debate in the literature regarding the role of CNO in chemogenetics. Some studies have shown that CNO does not cross the blood-brain barrier but that is instead converted to clozapine, which acts as a DIR agonist. Since it is known that the activation of DIR modulates the release of GABA from SN terminals, it would be interesting to see whether a similar effect on chronic seizures in the KA model is also observed in the PVCre + CNO group (Figure 3).

Reply: Thank you very much for this constructive suggestion. To test whether the CNO itself would have any seizure-modifying effect in chronic epileptic mice, we injected AAV-ef1a-DIO-mCherry control virus into the SNr of *PV-cre* mice (*PV-mCherry^{SNr}* mice). We found that CNO treatment did no change the frequency and seizure duration of both FS and GS in *PV-mCherry^{SNr}* mice, suggesting anti-seizure effect of CNO in *PV-hM4Di^{SNr}* mice be due to its chemogenetic modulation, but not be associated with physiological impacts of CNO itself. Thus, we have added this result in the revised **Fig. 3G** as following:

Revised Fig. 3G

Page 11 in revised manuscript:

--“CNO itself did no change the frequency and seizure duration of both FS and GS in *PV-mCherry^{SNr}* mice (Fig. 3G), suggesting anti-seizure effect of CNO in *PV-hM4Di^{SNr}* mice may not be associated with off target effects of CNO.”--

6. Page 6: It is stated that “Representative peri-event histograms confirmed that blue-light stimulation (473 nm, 20 Hz, 10 ms, 5 mW, 10 s on-off cycle) excited 15 out of 19 SNr GABAergic neurons from 3 *Vgat-ChR2-eYFP* mice, suggesting that SNr GABAergic neurons can be functionally activated by blue-light stimulation. Yellow-light stimulation (589 nm, continuous light, 5 mW, 10s on-off cycle), serving as the control stimulation, had no effect on the same neurons (Fig. S1F). It is unclear why a 20 Hz blue-light stimulation was used to activate SNr GABAergic neurons whereas continuous yellow light stimulation was used for control experiments. A similar comment would apply to the results shown in figure 1 G-1 if the same procedure was used.

Reply: Thank you very much for this comment. In fact, in optogenetic activation experiment, we used 20-Hz blue light stimulation (473 nm, 20 Hz, 10 ms, 5 mW) to test whether the ChR2-expressing neuron is functional, while yellow-light stimulation (589 nm, 20 Hz, 10 ms, 5 mW) serves as control light. This is the same situation in kindling model of Fig.1. However, in optogenetic inhibition experiment (Fig. 2), we used continuous yellow light (589 nm, continuous light, 5 mW) to test whether the ArchT-expressing neuron is functionally inhibited. Because continuous yellow light illumination allows to maintain the membrane hyperpolarization of proton pump expressing neurons for long time periods and avoids rebound excitation^{1,2}. To make it more clear, we have corrected description in the revised manuscript as following:

Page 6 in revised manuscript:

--“Representative peri-event histograms confirmed that blue-light stimulation (473 nm, 20 Hz, 10 ms, 5 mW, 10 s on-off cycle) excited 15 out of 19 SNr GABAergic neurons from 3 *Vgat-ChR2-eYFP* mice, suggesting that SNr GABAergic neurons can be functionally activated by blue-light stimulation. Yellow-light stimulation (589 nm, 20 Hz, 10 ms, 5 mW, 10 s on-off cycle), serving as the control stimulation, had no effect on the same neurons (Fig. S1F).”--

Page 9 in revised manuscript:

--“*In vivo* single-unit recordings confirmed that yellow-light stimulation (589 nm, continuous light, 5 mW, 10s on-off cycle), but not blue-light stimulation (473 nm, continuous light, 5 mW, 10 s on-off cycle, the control light), inhibited the firing rate of putative PV neurons (7 out of 16 neurons from 3 *PV-Arch*^{SNr} mice, Fig. 2C).”--

References:

1. Yizhar, O., L.E. Fenno, T.J. Davidson, M. Mogri, and K. Deisseroth, Optogenetics in neural systems. *Neuron*, 2011. 71(1): p. 9-34.
2. Kokaia, M., M. Andersson, and M. Ledri, An optogenetic approach in epilepsy. *Neuropharmacology*, 2013. 69: p. 89-95.

7. Page 9: “The number of stimulations needed to reach stage 2 was not affected here (only has a tendency, $U=19$, $P=0.1740$). I would suggest removing “only has a tendency” since the p value of 0.17 is not close to significance.

Reply: Thanks. We have deleted “only has a tendency” in this sentence as following:

Page 9 in revised manuscript:

--“The number of stimulations needed to reach stage 2 was not affected here ($U=19$, $P=0.1740$).”--

8. Page 29: It is unclear whether the status epilepticus induced with local administration of KA was pharmacologically stopped or allowed to self-terminate. Status epilepticus of different durations could lead to differences in seizure occurrence and neuropathology during the chronic period. Please specify.

Reply: Thank you very much for this important comment. We stereotaxically injected KA into right dorsal hippocampus to induce SE in anesthetized mice. The mice were allowed to recovery from anesthesia state and allowed to self-terminate. We totally agree with the reviewer that status epilepticus of different durations could lead to differences in seizure occurrence and neuropathology during the chronic period. In our study, we have noticed that the properties of FSs and GSs in HM3 and HM4 animals have a big variety. We speculate this can be caused by the methodology of chronic KA model in our study. Thus, to make a fair comparison, we use self-control in chronic KA experiment. According to reviewer's comment, we have added some description in the revised manuscript as following:

Page 31 in revised manuscript:

--"First, we stereotaxically injected KA into right dorsal hippocampus to induce SE in anesthetized mice. The mice were allowed to self-terminate and recovery from anesthesia state."--

9. Page 30: It is stated that "Spontaneous seizure events were defined as regular spike clusters with a duration of ≥ 10 s, spike frequency of ≥ 2 Hz and amplitude at least three times of the baseline EEG, accompanying behavioral tonic-clonic GSs". Does this mean that non-convulsive seizures that were only visible on the EEG were not considered?

Reply: Thank you very much for this important comment. In KA-induced chronic TLE model, spontaneous seizure events were defined as regular spike clusters with a duration of ≥ 10 s, spike frequency of ≥ 2 Hz and amplitude at least three times of the baseline EEG according to previous study¹. In the present model, most of seizures we recorded were characterized by bursts of high-voltage sharp waves (revised **Fig. 3E**, upper panel) associated with any obvious behavioral alterations in the parallel video recordings. As previously reported², those seizures only occurred in the ipsilateral hippocampus and usually were interpreted as non-convulsive focal seizures (FSs). In addition to FSs, mice also exhibited infrequent tonic-clonic generalized seizures (GSs) that were associated with typical paroxysmal EEG activity with obvious post-seizure depression (revised **Fig. 3E**, lower panel). To make it more clear, we have showed typical FS and GS in the revised **Fig. 3E** and added detailed description in the revised method.

Revised Fig. 3E

Page 30-31 in revised manuscript:

--"Most of seizures we recorded were characterized by bursts of high-voltage sharp waves and are not associated with any obvious behavioral alterations in the parallel video recordings, which usually were interpreted as non-convulsive FSs as previous study⁶⁵. In addition to FSs, mice also exhibited infrequent tonic-clonic GSs that were associated with typical paroxysmal EEG activity with obvious post-seizure depression."--

References:

1. Bragin, A., A. Azizyan, J. Almajano, C.L. Wilson, and J. Engel, Analysis of chronic seizure onsets after intrahippocampal kainic acid injection in freely moving rats. *Epilepsia*, 2005. 46(10): p. 1592-1598.

2. Riban, V., V. Bouilleret, B.T. Pham-Le, et al., Evolution of hippocampal epileptic activity during the development of hippocampal sclerosis in a mouse model of temporal lobe epilepsy. *Neuroscience*, 2002. 112(1): p. 101-111.

The paper should also be carefully edited. A few examples

- p.2, in the Introduction “...remote brain regions, extending from cortical to subcortical limbic structures and other remote structures.”

- p.3 in the Introduction: “...targeting at the SNr...”

- p. 5 in the Results: “in the previous study.”

- p.19: “Furthermore, optogenetic activation of PF glutamatergic neurons was sufficient to accelerate seizure development in hippocampal kindling model”.

Reply: Thank you very much. The English of manuscript have been re-checked by a native English speaker. All the revisions in the main manuscript are highlighted by red color. The listed examples have been revised as following:

Page 3 in revised manuscript:

--“Structural and metabolic imaging from both clinical and experimental studies demonstrate that abnormal pathological changes in TLE are associated with not only the neighboring epileptogenic structures but also with remote brain regions, extending from subcortical limbic structures to cortical and other remote structures^{5,6}”--

Page 3-4 in revised manuscript:

--“In addition to those findings, lesion¹⁹, pharmacological interference^{20, 21, 22, 23, 24} or deep brain stimulation^{25, 26} targeting the SNr can regulate the process of epileptic seizures, suggesting that the SNr plays a key role in seizure control.”--

Page 5 in revised manuscript:

--“The putative GABAergic neurons, which constitute the majority of SNr neurons, are characterized by high firing rates and narrow spike waveforms (Fig. S1A), as described in previous study³¹”--

Page 19 in revised manuscript:

--“Furthermore, glutamatergic neurons were sufficient to accelerate seizure development in hippocampal kindling model when they were optogenetically activated”--

Reviewers' Comments:

Reviewer #1:

Remarks to the Author:

The authors have compellingly and thoroughly addressed all my concerns. This is an interesting and solid set of findings.

Reviewer #2:

Remarks to the Author:

The authors did a commendable large amount of work to address all my concerns, except one, which can be fixed.

The remaining major issue is about the chronic epilepsy model. I think that what the authors call seizures, are not seizures. They rely on reference #65. However, Riban et al call these events paroxysmal discharges, not seizures. In their Fig. 4A, we see a burst of spikes, and Fig. 4B shows something like a subclinical seizure (FS), which looks like the generalized one shown in Fig. 5. This is the definition we use in the clinic. The FS looks similar to a GS. Spiking activity is OK, but it should display clear dynamics (usually around 10-20 Hz, and not 2 Hz).

For example, in Fig 3I, 20-40 seizures/h, this is really extreme for a focal model. This is more like continuous status epilepticus. My guess is that the authors misclassified bursts of spikes as FS (as top of Fig 3E).

They should ask a clinician, or an EEG expert, and redo the analysis of spontaneous seizures.

Discussion points

- 1) Some sentence parts do not make sense. Please have the English corrected. I have highlighted some in the annotated paper.
- 2) You do not study seizure propagation, which would require multi-site recordings. You study kindling progression. Please replace it throughout.
- 3) The discussion should include a clear statement that kindling was done in control animals. This allowed you to unravel mechanisms, some of which you checked in a more relevant chronic model.
- 4) Please make it clear that the circuit you describe is only modulatory. The effects you obtain are small. You do not abort or abolish seizures in the KA model, you decrease their occurrence (provided that the major point is clearly addressed).
- 5) You have used singly housed animals. This produces stress in animals and worsens the epilepsy phenotype (Mazouze et al., eNeuro, 2019). This caveat should be discussed.

Reviewer #3:

Remarks to the Author:

The authors addressed my concerns and accordingly revised the manuscript.

Minor concern: the authors should clarify what they mean by "indirect circuit" lines 363-364: "experiments were performed in the presence of TTX and 4-AP to block indirect circuit."

Reviewer #4:

Remarks to the Author:

No further comment but to proof read the paper carefully. There are still some "typos". For instance in

the Abstract, lines 36 & 37: "Collectively, our results revealed a long-range SNr-PF disinhibitory circuit, participates in regulating seizure in TLE, and that inactivation of this circuit can alleviate severity of epileptic seizures" a "that" is missing.

Reviewers' comments:

Reviewer #1 (Remarks to the Author):

The authors have compellingly and thoroughly addressed all my concerns. This is an interesting and solid set of findings.

Response: We truly appreciate your consideration on our work and important comment.

Reviewer #2 (Remarks to the Author):

The authors did a commendable large amount of work to address all my concerns, except one, which can be fixed. The remaining major issue is about the chronic epilepsy model. I think that what the authors call seizures, are not seizures. They rely on reference #65. However, Riban et al call these events paroxysmal discharges, not seizures. In their Fig. 4A, we see a burst of spikes, and Fig. 4B shows something like a subclinical seizure (FS), which looks like the generalized one shown in Fig. 5. This is the definition we use in the clinic. The FS looks similar to a GS. Spiking activity is OK, but it should display clear dynamics (usually around 10-20 Hz, and not 2 Hz). For example, in Fig 3I, 20-40 seizures/h, this is really extreme for a focal model. This is more like continuous status epilepticus. My guess is that the authors misclassified bursts of spikes as FS (as top of Fig 3E). They should ask a clinician, or an EEG expert, and redo the analysis of spontaneous seizures.

Response: We truly appreciate the reviewer for his or her consideration on our work and insightful comments. According to reviewer's suggestion, we have asked clinicians and EEG experts (Dr. Shuang Wang and Dr. Yi Guo) to carefully review our EEG data. They agree with the reviewer that the FS defined in our manuscript is exactly paroxysmal discharges defined in previous studies¹⁻³. Generally, such paroxysmal discharges are not associated with any obvious behavioral abnormalities². In our study, paroxysmal events were defined as non-convulsive focal seizures (FS) when they lasted more than 10 s and had average spike frequency ≥ 2 Hz according to previous study⁴. Therefore, to make it more clear, we have included more detail descriptions about the EEG analysis and definition as following:

Page 31-32 in the revised manuscript:

--"Most of discharges we recorded were characterized by bursts of high-voltage sharp waves and are not associated with any obvious behavioral alterations in the parallel video recordings, which usually were interpreted as paroxysmal discharges as previous studies^{68, 69}. Such paroxysmal discharges were defined as non-convulsive focal seizures (FS) when they lasted more than 10 s and had average spike frequency ≥ 2 Hz⁷⁰. In addition to paroxysmal discharges, mice also exhibited infrequent tonic-clonic GSs that were associated with typical paroxysmal EEG activity with obvious post-seizure depression."--

References:

1. Riban V, Bouilleret V, Pham-Le BT, Fritschy JM, Marescaux C, Depaulis A. Evolution of hippocampal epileptic activity during the development of hippocampal sclerosis in a mouse model of temporal lobe epilepsy. *Neuroscience* 112, 101-111 (2002).
2. Arabadzisz D, Antal K, Parpan F, Emri Z, Fritschy JM. Epileptogenesis and chronic seizures in a mouse model of temporal lobe epilepsy are associated with distinct EEG patterns and selective neurochemical alterations in the contralateral hippocampus. *Experimental neurology* 194, 76-90 (2005).
3. Groticke I, Hoffmann K, Loscher W. Behavioral alterations in a mouse model of temporal lobe epilepsy induced by intrahippocampal injection of kainate. *Experimental neurology* 213, 71-83 (2008).
4. Bragin A, Azizyan A, Almajano J, Wilson CL, Engel J. Analysis of chronic seizure onsets after intrahippocampal kainic acid injection in freely moving rats. *Epilepsia* 46, 1592-1598 (2005).

Discussion points

1) Some sentence parts do not make sense. Please have the English corrected. I have highlighted some in the annotated paper.

Response: Thank you very much for this comment. Accordingly, we have proofed our manuscript carefully. All the revisions are highlighted by red color.

2) You do not study seizure propagation, which would require multi-site recordings. You study kindling progression. Please replace it throughout.

Response: Thank you very much for this important suggestion. Accordingly, we have used “kindling progression” instead of “seizure propagation” in our revised manuscript.

3) The discussion should include a clear statement that kindling was done in control animals. This allowed you to unravel mechanisms, some of which you checked in a more relevant chronic model.

Response: Thank you very much for this important suggestion. To make it more clear, we have indicated “kindling model” in some result descriptions. Some representative examples are listed as following:

Page 20 line 19 in the revised manuscript:

--“Using optogenetics, we provided direct evidence that activation of SNr GABAergic neurons including PV neurons promoted kindling progression, while inhibition of these neurons alleviated the severity of epileptic seizures in hippocampal kindling model.”--

Page 21 line 12 in the revised manuscript:

--“We also investigated the underlying mechanism of SNr related circuitry involved in seizure of kindling model.”--

4) Please make it clear that the circuit you describe is only modulatory. The effects you obtain are small. You do not abort or abolish seizures in the KA model, you decrease their occurrence (provided that the major point is clearly addressed).

Response: Thank you very much for this important comment. Accordingly, we have indicated this point in our revised manuscript as following:

Page 22 line 20 in the revised manuscript:

--“It should be noted that the circuit we revealed here is only modulatory, as intervening this circuit did not abort or abolish seizures in both kindling and KA models.”--

5) You have used singly housed animals. This produces stress in animals and worsens the epilepsy phenotype (Mazouze et al., eNeuro, 2019). This caveat should be discussed.

Response: Thank you very much for this insightful comment. Accordingly, we have added this caveat in our description and cited this paper in our revised manuscript as following:

Page 25 line 13 in the revised manuscript:

--“The mice were individually housed after surgery for a better recovery and meanwhile reducing the failure rate of electrodes implantation, although singly housed animals showed stress and worse epilepsy phenotype⁶⁴.”--

Reviewer #3 (Remarks to the Author):

The authors addressed my concerns and accordingly revised the manuscript. Minor concern: the authors should clarify what they mean by "indirect circuit" lines 363-364: "experiments were performed in the presence of TTX and 4-AP to block indirect circuit."

Response: We truly appreciate your consideration on our work and important comment. Accordingly, we have added corresponding description to clarify "using TTX and 4-AP to block indirect circuit" in our experiment as following:

Page 15 line 9 in the revised manuscript:

--"Experiments performed in the presence of TTX and 4-AP, which were used to block action potential-dependent synaptic transmission in indirect circuit ^{41, 42}, demonstrated the direct monosynaptic input from the SNr to the PF."--

Reviewer #4 (Remarks to the Author):

No further comment but to proof read the paper carefully. There are still some "typos". For instance in the Abstract, lines 36 & 37: "Collectively, our results revealed a long-range SNr-PF disinhibitory circuit, participates in regulating seizure in TLE, and that inactivation of this circuit can alleviate severity of epileptic seizures" a "that" is missing.

Response: Thank you very much for your consideration on our work and important comments. Accordingly, we have proofed our manuscript carefully. All the revisions are highlighted by red color. The example sentence was revised as following:

Page 2 line 11 in the revised manuscript:

--“Collectively, our results revealed that a long-range SNr-PF disinhibitory circuit participates in regulating seizure in TLE, and that inactivation of this circuit can alleviate severity of epileptic seizures.”--

Reviewers' Comments:

Reviewer #2:

Remarks to the Author:

No more issues

REVIEWERS' COMMENTS:

Reviewer #2 (Remarks to the Author):

No more issues

Response: We truly appreciate your consideration on our work and important comment.